# LATENT CAUSAL INVARIANT MODEL

## ABSTRACT

Current supervised learning can learn spurious correlation during the data-fitting process, imposing issues regarding interpretability, out-of-distribution (OOD) generalization, and robustness. To avoid spurious correlation, we propose a **La**tent **C**ausal **I**nvariance **M**odel (LaCIM) which pursues *causal prediction*. Specifically, we introduce latent variables that are separated into (a) output-causative factors and (b) others that are spuriously correlated to the output via confounders, to model the underlying causal factors. We further assume the generating mechanisms from latent space to observed data to be *causally invariant*. We give the identifiable claim of such invariance, particularly the disentanglement of output-causative factors from others, as a theoretical guarantee for precise inference and avoiding spurious correlation. We propose a Variational-Bayesian-based method for estimation and to optimize over the latent space for prediction. The utility of our approach is verified by improved interpretability, prediction power on various OOD scenarios (including healthcare) and robustness on security.

## 1 INTRODUCTION

Current data-driven deep learning models, revolutionary in various tasks though, heavily rely on *i.i.d* data to exploit all types of correlations to fit data well. Among such correlations, there can be spurious ones corresponding to biases (*e.g.*, selection or confounding bias due to coincidence of the presence of the third factor) inherited from the data provided. Such data-dependent spurious correlations can erode the *(i)* interpretability of decision-making, *(ii)* ability of out-of-distribution (OOD) generalization, *i.e.*, extrapolation from observed to new environments, which is crucial especially in safety-critical tasks such as healthcare, and *(iii)* robustness to small perturbation (Goodfellow et al., 2014).

Recently, there is a Renaissance of causality in machine learning, expected to pursue causal prediction (Schölkopf, 2019). The so-called "causality" is pioneered by Judea Pearl (Pearl, 2009), as a mathematical formulation of this metaphysical concept grasped in the human mind. The incorporation of a *priori* about cause and effect endows the model with the ability to identify the causal structure which entails not only the data but also the underlying process of how they are generated. For causal prediction, the old-school methods (Peters et al., 2016; Bühlmann, 2018) causally related the output label $Y$ to the *observed input* $X$, which however is NOT conceptually reasonable in scenarios with sensory-level observed data (*e.g. modeling pixels as causal factors of $Y$ does not make much sense*).

For such applications, we rather adopt the manner in Bengio et al. (2013); Biederman (1987) to relate the causal factors of $Y$ to unobserved abstractions denoted by $S$, *i.e.*, $Y \leftarrow f_y(S, \varepsilon_y)$ via mechanism $f_y$. We further assume existence of additional latent components denoted by $Z$, that together with $S$ generates the input $X$ via mechanism $f_x$ as $X \leftarrow f_x(S, Z, \varepsilon_x)$. Taking image classification as an example, the $S$ and $Z$ respectively refer to object-related abstractions (*e.g.*, contour, texture, color) and contextual information (*e.g.*, light, view). Such an assumption is similarly adopted in the literature of nonlinear Independent Components Analysis (ICA) (Hyvarinen and Morioka, 2016; Hyvärinen et al., 2019; Khemakhem, Kingma and Hyvärinen, 2020; Teshima et al., 2020) and latent generative models (Suter et al., 2019), which are *however* without separation of output ($y$)-causative factors (*a.k.a*, $S$) and other correlating factors (*a.k.a*, $Z$) that can both be learned in data-fitting process.

We encapsulate these assumptions into a novel causal model, namely **La**tent **C**ausal **I**nvariance **M**odel (LaCIM) as illustrated in Fig. 1, in which we assume the structural equations $f_x$ (associated with $S, Z \to X$), $f_y$ (associated with $S \to Y$) to be the **C**ausal **I**nvariant **Me**chanisms (CIMe) that hold under any circumstances with P$(S, Z)$ allowed to be varied across domains. The incorporation of these

*priories* can explain the spurious correlation embedded in the back-door path from $Z$ to $Y$ (contextual information to the class label in image classification). To avoid learning spurious correlations, our goal is to identify the intrinsic CIMe $f_x, f_y$. Specifically, we first prove the identifiability (*i.e.*, the possibility to be precisely inferred up to an equivalence relation) of the CIMe. Notably, far beyond the scope in existing literature (Khemakhem, Kingma and Hyvärinen, 2020), our results can implicitly, and are the *first* to disentangle the output-causative factors (*a.k.a*, $S$) from others (*a.k.a*, $Z$) for prediction, to ensure the isolation of undesired spurious correlation. Guaranteed by such, we propose to estimate the CIMe by extending the Variational Auto-encoder (VAE) (Kingma and Welling, 2014) to the supervised scenario. For OOD prediction, we propose to optimize over latent space under the identified CIMe. To verify the correctness of our identifiability claim, we conduct a simulation experiment. We further demonstrate the utility of our LaCIM via high explainable learned semantic features, improved prediction power on various OOD scenarios (including tasks with confounding and selection bias, healthcare), and robustness on security.

We summarize our contribution as follows: **(i) Methodologically**, we propose in section 4.1 a latent causal model in which only a subset of latent components are causally related to the output, to avoid spurious correlation and benefit OOD generalization; **(ii) Theoretically**, we prove the identifiability (in theorem 4.3) of CIMe $f_x, f_y$ from latent variables to observed data, which disentangles output-causative factors from others; **(iii) Algorithmically**, guided by the identifiability, we in section 4.3 reformulate Variational Bayesian method to estimate CIMe during training and optimize over latent space during the test; **(iv) Experimentally**, LaCIM outperforms others in terms of prediction power on OOD tasks and interpretability in section 5.2, and robustness to tiny perturbation in section 5.3.

## 2 RELATED WORK

The invariance/causal learning proposes to learn the assumed invariance for transferring. For the invariance learning methods in Krueger et al. (2020) and Schölkopf (2019), the "invariance" can refer to stable correlation rather than causation, which lacks the interpretability and impedes its generalization to a broader set of domains. For causal learning, Peters et al. (2016); Bühlmann (2018); Kuang et al. (2018); Heinze-Deml and Meinshausen (2017) assume causal factors as observed input, which is inappropriate for sensory-level observational data. In contrast, our LaCIM introduces latent components as causal factors of the input; *more importantly*, we explicitly separate them into the output-causative features and others, to avoid spurious correlation. Further, we provide the identifiability claim of causal invariant mechanisms. In independent and concurrent works, Teshima et al. (2020) and Ilse et al. (2020) also explore latent variables in causal relation. As comparisons, Teshima et al. (2020) did not differentiate $S$ from $Z$; and Ilse et al. (2020) proposed to augment intervened data, which can be intractable in real cases.

Other works which are conceptually related to us, as a non-exhaustive review, include (i) transfer learning which also leverages invariance in the context of domain adaptation (Schölkopf et al., 2011; Zhang et al., 2013; Gong et al., 2016) or domain generalization (Li et al., 2018; Shankar et al., 2018); and (ii) causal inference (Pearl, 2009; Peters et al., 2017) which proposes a structural causal model to incorporate intervention via "do-calculus" for cause-effect reasoning and counterfactual learning; (iii) latent generative model which also assumes generation from latent space to observed data (Kingma and Welling, 2014; Suter et al., 2019) *but* aims at learning generator in the unsupervised scenario.

## 3 PRELIMINARIES

**Problem Setup & Notation** Let $X, Y$ respectively denote the input and output variables. The training data $\{\mathcal{D}^e\}_{e \in \mathcal{E}_{\text{train}}}$ are collected from the set of multiple environments $\mathcal{E}_{\text{train}}$, where each domain $e$ is associated with a distribution $P^e(X, Y)$ over $\mathcal{X} \times \mathcal{Y}$ and $\mathcal{D}^e = \{x_i^e, y_i^e, d^e\}_{i \in [n_e]} \overset{i.i.d}{\sim} P^e$ with $[k] := \{1, ..., k\}$ for any $k \in \mathbb{Z}^+$. The $d^e \in \{0, 1\}^m$ denotes the one-hot encoded domain index for $e$, where $1 \le m := |\mathcal{E}_{\text{train}}| \le n := \sum_{e \in \mathcal{E}_{\text{train}}} n_e$. Our goal is to learn a model $f : \mathcal{X} \mapsto \mathcal{Y}$ that learns output ($y$)-causative factors for prediction and performs well on the set of all environments $\mathcal{E} \supset \mathcal{E}_{\text{train}}$, which is aligned with existing OOD generalization works (Arjovsky et al., 2019; Krueger et al., 2020). We use respectively upper, lower case letter and Cursive letter to denote the random variable, the instance and the space, *e.g.*, $a$ is an instance in the space $\mathcal{A}$ of random variable $A$. The $[f]_\mathcal{A}$ denotes the $f$ restricted on dimensions of $\mathcal{A}$. The Sobolev space $W^{k,p}(\mathcal{A})$ contains all $f$ such that $\int_\mathcal{A} |\partial_A f^\alpha|_{A=a}|^p da < \infty, \forall \alpha \le k$.

**Structural Causal Model.** The structural causal model (SCM) is defined as the causal graph assigned with structural equations. The causal graph encodes the assumptions in missing arrows in a directed acylic graph (DAG): $G = (V, E)$ with $V, E$ respectively denoting the node set and the edge set. The $Pa(k)$ denotes the set of parent nodes of $V_k$ for each $V_k \in V$ and the $X \to Y \in E$ indicates the causal effect of $X$ on $Y$. The structural equations $\{V_k \leftarrow f_k(Pa(k), \varepsilon_k)\}_{V_k \in V}$, quantify the causal effects shown in the causal graph $G$. By assuming independence among exogenous variables $\{\varepsilon_k\}_k$, the Causal Markov Condition states that $P(\{V_k = v_k\}_{V_k \in V}) = \Pi_k P(V_k = v_k | Pa(k) = pa(k))$. A back-door path from $V_a$ to $V_b$ is defined as a path that ends with an arrow pointing to $V_a$ (Pearl, 2009).

## 4 METHODOLOGY

We build our causal model associated with **C**ausal **I**nvariant **Me**chanism (CIMe, *i.e.*, $f_x, f_y$) and a **priori** about the generating process in section 4.1, followed by our identifiability result for CIMe in section 4.2. Finially, we introduce our learning method to estimate CIMe in section 4.3.

### 4.1 LATENT CAUSAL INVARIANCE MODEL

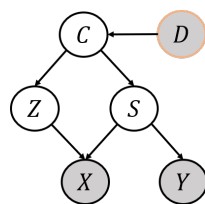

We introduce latent variables to model the abstractions/concepts that play as causal factors that generate the observed variables $(X, Y)$, which is more reasonable than assuming the $X$ as the direct cause of $Y$ in scenarios with sensory-level data. We explicitly separate the latent variables into two parts: the $S$ and $Z$ that respectively denote the $y$ (output)-causative and $y$-non-causative factors, as shown by the arrow $S \to Y$ in Fig. 1. Besides, the $X$ and $Y$ are respectively generated by $S, Z$ and $S$, via structural equations (with noise) $f_x, f_y$, which are denoted as **C**ausal **I**nvariant **Me**chanisms (CIMe) that hold across all domains. The output $Y$ denotes the label generated by human knowledge, *e.g.*, the semantic shape, the contour to discern the object, etc. Hence, we assume the $Y$ as the outcome/effect of these high-level abstractions (Biederman, 1987) rather than the cause (detailed comparison with $Y \to S$ is left in supplementary 7.7.1). We call the model associated with the causal

Figure 1: The DAG for LaCIM. The variables marked by white (gray) color represent the unobserved (observed) variables. Each arrow represents the causal effect from the variable it points from on the one it points to. The $C$ denotes the confounder of $S, Z$, which are the causal factors of $X, Y$. The $D$ denotes the domain index, which varies across domains and characterizes the distribution of $C$.

graph in Fig. 1 as **La**tent **C**ausal **I**nvariance **M**odel (LaCIM), with formal definition given in Def. 4.1.

As an illustration, we consider the image classification in which $X, Y$ denote the image and the class label. Instead of $X$, *i.e.*, the pixels, it is more reasonable to assume the causal factors (of $X, Y$) as latent concepts $(S, Z)$ that can denote light, angle, the shape of the object to generate $X$ following the physical mechanisms. Among these concepts, only the ones that are causally related to the object, *i.e.*, $S$ (*e.g.*, shape) are causal factors of the object label, *i.e.*, $Y$. Following the physical or natural law, the mechanisms $S, Z \to X, S \to Y$ invariantly hold across domains. The $\mathcal{S} := \mathbb{R}^{q_s}, \mathcal{Z} := \mathbb{R}^{q_z}$ denote the space of $S, Z$, with $P^e(S, Z)$ (that characterizes the correlation between $S$ and $Z$) varying across $\mathcal{E}$ (*e.g.*, the object is more associated with a specific scene than others).

We assume that the $y$-non-causative factor (*i.e.*, $Z$) is associated with (but not causally related to) $S, Y$ through the confounder $C$, which is allowed to take a specific value for each sample unit. Therefore, the back-door path $Z \leftarrow C \to S \to Y$ induces the correlation between $Z$ and $Y$ in each single domain. Rather than invariant causation, this correlation is data-dependent and can vary across domains, which is known as *"spurious correlation"*. In real applications, this spurious correlation corresponds to the bias inherited from data, *e.g.* the contextual information in object classification. This domain-specific $S$-$Z$ correlation, can be explained by the source variable $D$, which takes a specific and fixed value for each domain and functions the prior of distribution of the confounder $C$, as illustrated in Fig. 1. This source variable $D$ can refer to attributes/parameters that characterize the distribution of $S, Z$ in each domain. When such attributes are unobserved, we use the domain index as a substitute. Consider the cat/dog classification task as an illustration, the animal in each image is either associated with the snow or grass. The $S, Z$ respectively denote the concepts of animals and scenes. The $D$ denotes the sampler, which can be described by the proportions of scenes associated with the cat and those associated with the dog. The $D$ generates the $C$ that denotes the (time, weather) to go outside and

collect samples. Since each sampler may have a fixed pattern (*e.g.* gets used to going out in the sunny morning (or in the snowy evening)), the data he/she collects, may have sample selection bias (*e.g.* with dogs (cats) more associated with grass (snow) in the sunny morning (or snowy evening) ). In this regard, the scene concepts $Z$ can be correlated with the animal concepts $S$, and also the label $Y$.

**Definition 4.1** (LaCIM). *The **La**tent **C**ausal **I**nvariance **M**odel (LaCIM) for $e \in \mathcal{E}$ is defined as a SCM characterized by (i) the **causal graph**, i.e., the $G = (V, E)$ with $V = \{C, S, Z, X, Y\}$ and $E = \{C \rightarrow S, C \rightarrow Z, Z \rightarrow X, S \rightarrow X, S \rightarrow Y\}$; and (ii) **structural equations** with causal mechanisms $\{f_c, f_z, f_s, f_x, f_y\}$ embodying the quantitative causal information: $c \leftarrow f_c(d^e, \varepsilon_c), z \leftarrow f_z(c, \varepsilon_z), s \leftarrow f_s(c, \varepsilon_s); x \leftarrow f_x(s, z, \varepsilon_x); y \leftarrow f_y(s, \varepsilon_y)$, in which $\{\varepsilon_c, \varepsilon_z, \varepsilon_s, \varepsilon_x, \varepsilon_y\}$ are independent exogenous variables that induce $p_{f_c}(c|d^e), p_{f_z}(z|c), p_{f_s}(s|c), p_{f_x}(x|s, z), p_{f_y}(y|s)$. The CIMe $f_x, f_y$ are assumed to be invariant across $\mathcal{E}$. We call the environment-dependent parts: $P^e(S, Z)$ and $P^e(S, Z|X)$ as $S, Z$-prior and $S, Z$-inference in the following.*

**Remark 1.** *We denote LaCIM-$d_s$ and LaCIM-d as two versions of LaCIM, with the source variable $d_s$ with practical meaning (e.g. attributes or parameters of $P(S, Z)$) observed or not. The observation of $d_s$ can be possible in some applications (e.g., age, gender that characterizes population in medical diagnosis). As for the LaCIM-d with $d_s$ unobserved, we use domain index $D$ as a substitute.*

Denote $\mathcal{C}$ as the space of $C$. We assume that the $\mathcal{C}$ is finite union of disjoint sets $\{\mathcal{C}_r\}_{r=1}^R$, *i.e.* $\mathcal{C} := \cup_{r=1}^R \mathcal{C}_r$, such that for any $c_{r,i} \neq c_{r,j} \in \mathcal{C}_r$, it holds that $p(s, z|c_{r,i}) = p(s, z|c_{r,j})$ for any $(s, z)$. Returning to the cat/dog classification example, the $\mathcal{C}$ denotes the range of time to collect samples, *i.e.*, $00 : 00\text{-}24 : 00$. The $\mathcal{C}$ can be divided into several time periods $\mathcal{C}_1, ..., \mathcal{C}_R$, such that the proportion of concepts of (animal,scene) given any $c$ in the same period is unchanged, *e.g.*, the dog often comes up on the grass in the morning. Further, since $p(x, y|s, z) = p(x|s, z)p(y|s)$ is invariant, we have for each $\mathcal{C}_r$ that $p(x, y|c_{r,i}) = \int p(x, y|s, z)p(s, z|c_{r,i})dsdz = \int p(x, y|s, z)p(s, z|c_{r,j})dsdz = p(x, y|c_{r,j})$ for any $(x, y)$. That is, the $\{p(x, y|c_r\}_{c_r \in \mathcal{C}_r}$ for each $(x, y)$ collapse to a single point, namely $p(x, y|c_r)$. In this regard, we have $p^e(x, y) := p(x, y|d^e) = \sum_{r=1}^R p(x, y|c_r)p(c_r|d^e)$. Besides, we assume the *Additive Noise Model* (ANM) for $X$, *i.e.*, $f_x(s, z, \varepsilon_x) = \hat{f}_x(s, z) + \varepsilon_x$ (we replace $\hat{f}_x$ with $f_x$ without loss of generality), which has been widely adopted to identify the causal factors (Janzing et al., 2009; Peters et al., 2014; Khemakhem, Kingma and Hyvärinen, 2020). We need to identify the CIMe (*i.e.*, $f_x, f_y$), guaranteed by the identifiability that ensures the learning method to distinguish $S$ from $Z$ to avoid spurious correlation, as presented in section 4.2. Traditionally speaking, the identifiability means the parameter giving rise to the observational distribution $p_{\theta^\star}(x, y|d^e)$ can be uniquely determined, *i.e.*, $p_\theta(x, y|d^e) = p_{\tilde{\theta}}(x, y|d^e) \implies \theta = \tilde{\theta}$. Instead of strict uniqueness, we rather identify an equivalent class of $\theta^\star$ (in Def. 4.2) that suffices to disentangle the $y$-causative features $S$ from $Z$ to avoid learning spurious correlation. To achieve this goal, we first narrow our interest in case when $p(s, z|c)$ is exponential family in Eq. (1), in which we can respectively identify the $S, Z$ up to linear and point-wise transformations given by theorem 4.3; then we generalize to any $p(s, z|c)$ as long as it belongs to Sobolev space, as explained in theorem 4.4.

A reformulated VAE is proposed to learn the CIMe practically. For generalization, note that the gap between two environments in terms of prediction given $x$, *i.e.*, $\left| \mathbb{E}_{p^{e_2}}[Y|X = x] - \mathbb{E}_{p^{e_1}}[Y|X = x] \right| = \int_{\mathcal{S}} |p^{e_2}(s|x) - p^{e_1}(s|x)| p_{f_y}(y|s)ds$, is mainly due to the inconsistency of $S, Z$-inference, *i.e.*, $p^e(s, z|x) \neq p^{e'}(s, z|x)$ for $e' \neq e$ (for details please refer to theorem 7.1 in supplement 7.1). Therefore, one cannot directly apply the trained $\{p^e(s, z|x), p^e(y|x)\}_{e \in \mathcal{E}_{\text{train}}}$ to the inference model of new environment, *i.e.* $p^{e'}(s, z|x), p^{e'}(y|x)$ for $e' \notin \mathcal{E}_{\text{train}}$. To solve this problem and generalize to new environment, we note that since $p_{f_x}(x|s, z)$ and $p_{f_y}(y|s)$ are shared among all environments, we propose to inference $s, z$ that give rise to the test sample $x$ via maximizing the identified $p_{f_x}(x|s, z)$, as a pseudo-likelihood of $x$ given $(s, z)$, rather than using $S, Z$-inference model which is inconsistent among environments. Then, we feed estimated $s$ into invariant predictor $p_{f_y}(y|s)$ for prediction.

### 4.2 IDENTIFIABILITY OF CAUSAL INVARIANT MECHANISMS

We present the identifiability claim about the CIMe $f_x, f_y$, which implicitly distinguishes the $y$-causative factors (*a.k.a*, $S$) from others (*a.k.a*, $Z$) for prediction, to provide a theoretical guarantee for avoiding spurious correlations. Notably, the $S$ and $Z$ play "asymmetric roles" in terms of generating process, as reflected in additional generating flow from $S$ to $Y$. This "information intersection" property of $S$, *i.e.*, $f_y^{-1}(\bar{y}) = [f_x^{-1}]_S(\bar{x})$ for any $(\bar{x}, \bar{y}) \in f_x(\mathcal{S}, \mathcal{Z}) \times f_y(\mathcal{S})$ if $y = f_y(s) + \varepsilon_y$, is exploited to disentangle $S$ from $Z$. Such a disentanglement analysis, is crucial to causal prediction

but lacked in existing literature about identifiability, such as those identifying the discrete latent confounders (Janzing, Sgouritsa, Stegle, Peters and Schölkopf, 2012; Sgouritsa et al., 2013); or those relying on ANM assumption (Janzing, Peters, Mooij and Schölkopf, 2012); linear ICA (Eriksson and Koivunen, 2003); (Khemakhem, Kingma and Hyvärinen, 2020; Khemakhem, Monti, Kingma and Hyvärinen, 2020; Teshima et al., 2020) (Please refer to supplement 7.6 for more broad reviews). Besides, our analysis extends the scope of Khemakhem, Kingma and Hyvärinen (2020) to categorical $Y$ and general forms of $\mathrm{P}(S, Z|C = c)$ that belongs to Sobolev space, in theorem 4.4. Note that our analysis does NOT require observing the original source variable $d_s$.

We first narrow our interest to a family class of LaCIM denoted as $\mathcal{P}_{\exp}$ in which any $p \in \mathcal{P}_{\exp}$ satisfies that (i) the $S, Z$ belong to the exponential family; and that (ii) the $Y$ is generated from the ANM. We will show later that $\mathcal{P}_{\exp}$ can approximate any $\mathrm{P}(S, Z|c) \in W^{r,2}(\mathcal{S} \times \mathcal{Z})$ for some $r \geq 2$:

$$\mathcal{P}_{\exp} = \left\{ \text{LaCIM with } y = f_y(s) + \varepsilon_y, p(s, z|c) := p_{\mathbf{T}^z, \mathbf{\Gamma}^z_c}(z|c) p_{\mathbf{T}^s, \mathbf{\Gamma}^s_c}(s|c) \right\}$$

$$\text{with } p_{\mathbf{T}^t, \mathbf{\Gamma}^t_c}(t) := \prod_{i=1}^{q_t} \exp\left( \sum_{j=1}^{k_t} T^t_{i,j}(t_i) \Gamma^t_{c,i,j} + B_i(t_i) - A^t_{c,i} \right) \text{ for } t = s, z, \text{ and } e \in \mathcal{E}, \tag{1}$$

where $\{T^t_{i,j}(t_i)\}, \{\Gamma^t_{c,i,j}\}$ denote the sufficient statistics and natural parameters, $\{B_i\}, \{A^t_{c,i}\}$ denote the base measures and normalizing constants to ensure the integral of distribution equals to 1. Let $\mathbf{T}^t(t) := [\mathbf{T}^t_1(t_1), ..., \mathbf{T}^t_{q_t}(t_{q_t})] \in \mathbb{R}^{k_t \times q_t}$ $(\mathbf{T}^t_i(t_i) := [T^t_{i,1}(t_i), ..., T^t_{i,k_t}(t_i)], \forall i \in [q_t])$, $\mathbf{\Gamma}^t_c := [\mathbf{\Gamma}^t_{c,1}, ..., \mathbf{\Gamma}^t_{c,q_t}] \in \mathbb{R}^{k_t \times q_t}$ $(\mathbf{\Gamma}^t_{c,i} := [\Gamma^t_{c,i,1}, ..., \Gamma^t_{c,i,k_t}], \forall i \in [q_t])$. We define the $\sim_p$-identifiability for $\theta := \{f_x, f_y, \mathbf{T}^s, \mathbf{T}^z\}$ as:

**Definition 4.2** ($\sim_p$-identifiability). *We define a binary relation on the parameter space of $\mathcal{X} \times \mathcal{Y}$: $\theta \sim_p \tilde{\theta}$ if there exist two sets of permutation matrices and vectors, $(M_s, a_s)$ and $(M_z, a_z)$ for $s$ and $z$ respectively, such that for any $(x, y) \in \mathcal{X} \times \mathcal{Y}$,*

$$\mathbf{T}^s([f_x^{-1}]_{\mathcal{S}}(x)) = M_s \tilde{\mathbf{T}}^s([\tilde{f}_x^{-1}]_{\mathcal{S}}(x)) + a_s, \ \mathbf{T}^z([f_x^{-1}]_{\mathcal{Z}}(x)) = M_z \tilde{\mathbf{T}}^z([\tilde{f}_x^{-1}]_{\mathcal{Z}}(x)) + a_z,$$

$$p_{f_y}(y|[f_x^{-1}]_{\mathcal{S}}(x)) = p_{\tilde{f}_y}(y|[\tilde{f}_x^{-1}]_{\mathcal{S}}(x)),$$

*We say that $\theta$ is $\sim_p$-identifiable, if for any $\tilde{\theta}$, $p^e_\theta(x, y) = p^e_{\tilde{\theta}}(x, y) \ \forall e \in \mathcal{E}_{\text{train}}$, implies $\theta \sim_p \tilde{\theta}$.*

It can be shown that $\sim_p$ satisfies the reflective property ($\theta \sim_p \theta$), the symmetric property (if $\theta \sim_p \tilde{\theta}$ then $\tilde{\theta} \sim_p \theta$), and the transitive property (if $\theta_1 \sim_p \theta_2$ and $\theta_2 \sim_p \theta_3$, then $\theta_1 \sim_p \theta_3$), and hence is an equivalence relation (details in supplement 7.2). This definition states that the $S, Z$ can be identified up to permutation and point-wise transformation, which is sufficient for disentanglement of $S$ and identifying the predicting mechanism $p_{f_y}(y|[f_x^{-1}]_{\mathcal{S}}(x))$. Specifically, the definition regarding $f_x$ implies the separation of $S$ and $Z$ unless the extreme case when $S$ can be represented by $Z$, *i.e.*, there exists a function $h : \mathcal{S} \to \mathcal{Z}$ such that $[f_x^{-1}]_{\mathcal{S}}(x) = h([f_x^{-1}]_{\mathcal{Z}}(x))$. This definition is inspired by but beyond the scope of unsupervised scenario considered in nonlinear ICA (Hyvärinen et al., 2019; Khemakhem, Kingma and Hyvärinen, 2020) to further distinguish of $S$ from $Z$. Besides, the $p_{f_y}(y|[f_x^{-1}]_{\mathcal{S}}(x)) = p_{\tilde{f}_y}(y|[\tilde{f}_x^{-1}]_{\mathcal{S}}(x))$ further guarantees the identifiability of prediction: predict using $f_y(s)$ with $s$ obtained from $f_x$. The following theorem presents the $\sim_p$-identifiability for $\mathcal{P}_{\exp}$:

**Theorem 4.3** ($\sim_p$-identifiability). *For $\theta$ in the LaCIM $p^e_\theta(x, y) \in \mathcal{P}_{\exp}$ for any $e \in \mathcal{E}_{\text{train}}$, we assume that **i)** CIMe satisfies that $f_x$, $f'_x$ and $f''_x$ are continuous and that $f_x, f_y$ are bijective; **ii)** the $T^t_{i,j}$ are twice differentiable for any $t = s, z, i \in [q_t], j \in [k_t]$; **iii)** the exogenous variables satisfy that the characteristic functions of $\varepsilon_x, \varepsilon_y$ are almost everywhere nonzero. Under the diversity condition on $A := [P_{d^{e_1}}^\top, ..., P_{d^{e_m}}^\top]^\top \in \mathbb{R}^{m \times R}$ with $P_{d^e} := [p(c_1|d^e), ..., p(c_R|d^e)]$ that the $A$ and $\left[ [\mathbf{\Gamma}^{t=s,z}_{c_2} - \mathbf{\Gamma}^{t=s,z}_{c_1}]^\top, ..., [\mathbf{\Gamma}^{t=s,z}_{c_R} - \mathbf{\Gamma}^{t=s,z}_{c_1}]^\top \right]^\top$ have full column rank for both $t = s$ and $t = z$, we have that the $\theta := \{f_x, f_y, \mathbf{T}^s, \mathbf{T}^z\}$ are $\sim_p$ identifiable.*

The bijectivity of $f_x$ and $f_y$ have been widely assumed in Janzing et al. (2009); Peters et al. (2014; 2017); Khemakhem, Kingma and Hyvärinen (2020); Teshima et al. (2020) as a basic condition for identifiability. It naturally holds for $f_x$ to be bijective since the latent components $S, Z$, as high-level abstractions which can be viewed as embeddings in auto-encoder (Kramer, 1991), lies in lower-dimensional space compared with input $X$ which is supposed to have more variations, *i.e.*, $(q_s + q_z < q_x)$. For categorical $Y$, the $f_y$ which generates the classification result, *i.e.*, $p(y = k|s) = [f_y]_k(s)/(\sum_k [f_y]_k(s))$, will be shown later to be identifiable.

The diversity condition implies that **i)** $m \geq R \geq \max(k_z * q_z, k_s * q_s) + 1$; and that **ii)** different environments are variant enough in terms of $S$-$Z$ correlation (which is also assumed in Arjovsky et al. (2019)), as a necessary for the invariant one to be identified. As noted in the formulation, a larger $m$ would be easier to satisfy the condition, which agrees with the intuition that more environments can provide more complementary information for the identification of the invariant mechanisms.

**Remark 2.** *The dimensions of the ground-truth $S, Z$ are unknown, making the check about whether $m$ is large enough impossible. Besides, in some real applications, the training environments are passively observed and may not satisfy the condition. However, we empirically find the improvement of LaCIM in terms of both OOD prediction and interpretability, if the multiple environments provided are diverse enough. Besides, a training environment can be the mixture of many sub-environments, which motivates to splitting the data according to their source ID or clustering results (Teney et al., 2020) to obtain more environments, making the condition easier to satisfy.*

**Extension to the general form of LaCIM.** We generalize the identifiable result in theorem 4.3 to any LaCIM as long as its $P(S, Z | C = c) \in W^{r,2}(\mathcal{S} \times \mathcal{Z})$ (for some $r \geq 2$) and categorical $Y$, in the following theorem. This is accomplished by showing that any such LaCIM can be approximated by a sequence of distributions in $\mathcal{P}_{\exp}$, motivated by the facts in Barron and Sheu (1991) that the exponential family is dense in the set of distributions with bounded support, and in Maddison et al. (2016) that the continuous variable with multinomial logit model can be approximated by a series of distributions with *i.i.d* Gumbel noise as the temperature converges to infinity.

**Theorem 4.4** (Asymptotic $\sim_p$-identifiability). *Consider a LaCIM satisfying that $p_{f_x}(x|s,z)$ and $p_{f_y}(y|s)$ are smooth w.r.t $s, z$ and $s$ respectively. For each $e$ and $c \in \mathcal{C}$, suppose $P^e(S, Z | C = c) \in W^{r,2}(\mathcal{S} \times \mathcal{Z})$ for some $r \geq 2$, we have that $P$ is asymptotically $\sim_p$-identifiable defined as: $\forall \, \epsilon > 0$, $\exists \sim_p$-identifiable $\tilde{P}_\theta \in \mathcal{P}_{\exp}$, s.t. $d_{\mathrm{Pok}}(p^e(x,y), \tilde{p}_\theta^e(x,y)) < \epsilon, \forall e \in \mathcal{E}_{\mathrm{train}}, (x,y) \in \mathcal{X} \times \mathcal{Y}$ [1].*

## 4.3 Causal Supervised Variational Auto-Encoder

Guided by identifiability, we first provide the training method to learn $f_x, f_y$ by reformulating VAE in a supervised scenario, followed by optimization over latent space for inference and test.

**Training.** To learn the CIMe and $p_{f_x}(x|s,z), p_{f_y}(y|s)$ for invariant prediction, we implement the generative model to fit $\{p^e(x,y)\}_{e \in \mathcal{E}_{\mathrm{train}}}$, which has been guaranteed by theorem 4.3, 4.4 to be able to identify the ground-truth predicting mechanism. Specifically, we reformulate the objective of VAE, as a generative model proposed in (Kingma and Welling, 2014), in supervised scenario. For unsupervised learning, the VAE introduces the variational distribution $q_\psi$ parameterized by $\psi$ to approximate the intractable posterior by maximizing the following **E**vidence **L**ower **B**ound (ELBO): $-\mathcal{L}_{\phi,\psi} = \mathbb{E}_{p(x)}\left[\mathbb{E}_{q_\psi(z|x)} \log \frac{p_\phi(x,z)}{q_\psi(z|x)}\right]$, as a tractable surrogate of maximum likelihood $\mathbb{E}_{p(x)} \log p_\phi(x)$. Specifically, the ELBO is less than and equal to $\mathbb{E}_{p(x)}\left[\log p_\phi(x)\right]$ and the equality can only be achieved when $q_\psi(z|x) = p_\phi(z|x)$. Therefore, maximizing the ELBO over $p_\phi$ and $q_\psi$ will drive **(i)** $q_\psi(z|x)$ to learn $p_\phi(z|x)$; **(ii)** $p_\phi$ to learn the ground-truth model $p$ (including $p_\phi(x|z)$ to learn $p(x|z)$).

In our supervised scenario, we introduce the variational distribution $q_\psi^e(s, z | x, y)$ and the corresponding ELBO for any $e$ is $-\mathcal{L}_{\phi,\psi}^e = \mathbb{E}_{p^e(x,y)}\left[\mathbb{E}_{q_\psi^e(s,z|x,y)} \log \frac{p_\phi^e(x,y,s,z)}{q_\psi^e(s,z|x,y)}\right]$. Similarly, minimizing $\mathcal{L}_{\phi,\psi}^e$ can drive $p_\phi(x|s,z), p_\phi(y|s)$ to learn the CIMe (*i.e.* $p_{f_x}(x|s,z), p_{f_y}(y|s)$), and also $q_\psi^e(s, z | x, y)$ to learn $p_\phi^e(s, z|x, y)$. In other words, the $q_\psi$ can inherit the properties of $p_\phi$. As $p_\phi^e(s, z|x, y) = \frac{p_\phi^e(s,z|x)p_\phi(y|s)}{p_\phi^e(y|x)}$ for our DAG in Fig. 1, we can similarly reparameterize $q_\psi^e(s, z|x, y)$ as $\frac{q_\psi^e(s,z|x)q_\psi(y|s)}{q_\psi^e(y|x)}$. According to Causal Markov Condition, we have that $p_\phi^e(x, y, s, z) = p_\phi(x|s, z)p_\phi^e(s, z)p_\phi(y|s)$. Substituting the above reparameterizations into the ELBO with $q_\psi(y|s)$ replaced by $p_\phi(y|s)$, the $\mathcal{L}_{\phi,\psi}^e$ can be rewritten as:

$$\mathcal{L}_{\phi,\psi}^e = \mathbb{E}_{p^e(x,y)}\left[-\log q_\psi^e(y|x) - \mathbb{E}_{q_\psi^e(s,z|x)} \frac{p_\phi(y|s)}{q_\psi^e(y|x)} \log \frac{p_\phi(x|s,z)p_\phi^e(s,z)}{q_\psi^e(s,z|x)}\right], \quad (2)$$

where $q_\psi^e(y|x) = \int_{\mathcal{S}} q_\psi^e(s|x)p_\phi(y|s)ds$. The overall loss function is: $\mathcal{L}_{\phi,\psi} = \sum_{e \in \mathcal{E}_{\mathrm{train}}} \mathcal{L}_{\phi,\psi}^e$. The training datasets $\{\mathcal{D}^e\}_{e \in \mathcal{E}_{\mathrm{train}}}$ are applied to optimize prior model $p_\phi^e(s, z)$, inference

---

[1] The $d_{\mathrm{Pok}}$ denotes the Pokorov distance and $\lim_{n \to \infty} d_{\mathrm{Pok}}(\mu_n, \mu) \to 0 \iff \mu_n \xrightarrow{d} \mu$.

Table 1: MCC of identified latent variables. Average over 20 times for each data.

| | Data #1 | | Data #2 | | Data #3 | | Data #4 | | Data #5 | | Average | |
|---|---|---|---|---|---|---|---|---|---|---|---|---|
| | $Z$ | $S$ | $Z$ | $S$ | $Z$ | $S$ | $Z$ | $S$ | $Z$ | $S$ | $Z$ | $S$ |
| pool-LaCIM | 0.28 | 0.58 | 0.38 | 0.66 | 0.34 | 0.77 | 0.34 | 0.79 | 0.36 | 0.75 | 0.34 | 0.71 |
| LaCIM-$d_s$ (**Ours**, $m = 5$) | **0.81** | 0.86 | **0.81** | 0.87 | **0.85** | 0.87 | 0.73 | 0.78 | 0.86 | **0.87** | **0.82** | 0.85 |
| LaCIM-$d$ (**Ours**, $m = 3$) | 0.62 | 0.88 | 0.63 | 0.83 | 0.70 | 0.78 | 0.69 | 0.81 | 0.71 | 0.84 | 0.67 | 0.83 |
| LaCIM-$d$ (**Ours**, $m = 5$) | 0.64 | 0.82 | 0.75 | 0.80 | 0.76 | 0.83 | 0.79 | **0.90** | 0.75 | 0.85 | 0.74 ↑ | 0.84 ↑ |
| LaCIM-$d$ (**Ours**, $m = 7$) | 0.70 | **0.89** | **0.81** | **0.90** | 0.82 | **0.88** | **0.84** | 0.83 | **0.90** | 0.85 | 0.81 ↑ | **0.87** ↑ |

model $q_\psi^e(s, z|x)$ and generative models $p_\phi(x|s, z), p_\phi(y|s)$ in Eq. (2). The generative models $p_\phi(x|s, z), p_\phi(y|s)$ are shared among all environments, while the $p_\phi^e(s, z), q_\psi^e(s, z|x)$ are respectively $p_\phi(s, z|d_s^e), q_\psi(s, z|x, d_s^e)$ and $p_\phi(s, z|d^e), q_\psi(s, z|x, d^e)$ for LaCIM-$d_s$ and LaCIM-$d$.

**Inference & Test.** When $d_s^{e'}$ can be acquired during test for $e' \in \mathcal{E}_{\text{test}}$, we can predict $y$ as $\arg\max_y p_\phi(y|x, d_s^{e'}) = \int q_\psi(s|x, d_s^{e'}) p_\phi(y|s) ds$. Otherwise, for LaCIM-$d$ with $d_s$ unobserved, we first optimize $s, z$ via $(s^\star, z^\star) := \arg\max_{s,z} \log p_\phi(x|s, z)$ and predict $y$ as $\arg\max_y q_\psi(y|s^\star)$. Specifically, we adopt the strategy for optimization in Schott et al. (2018) that we first sample initial points and select the one with the maximum $\log p_\phi(x|s, z)$, then we optimize for 50 iterations using Adam. The implementation details and optimization effect are shown in supplement 7.9.

## 5 EXPERIMENTS

We evaluate LaCIM on (I) synthetic data to verify the identifiability in theorem 4.3; (II) OOD challenges: object classification with sample selection bias (Non-I.I.D. Image dataset with Contexts (NICO)); Hand-Writing Recognition with confounding bias (Colored MNIST (CMNIST)); prediction of Alzheimer's Disease (Alzheimer's Disease Neuroimaging Initiative (ADNI www.loni.ucla.edu/ADNI); (III) Robustness on detecting images with small perturbation (FaceForensics++).

### 5.1 SIMULATION

To verify the identifiability claim and effectiveness of our learning method, we implement LaCIM on synthetic data. The data generating process is provided in Supplement 7.8. The domain index $D \in \mathbb{R}^m$ is denoted as a one-hot encoded vector with $m = 5$. To verify the utility of training on multiple domains ($m > 1$), we also conduct LaCIM by pooling data from all $m$ domains together, namely pool-LaCIM for comparison. We randomly generate $m = 5$ datasets and run 20 times for each. We compute the metric mean correlation coefficient (MCC) adopted in Khemakhem, Kingma and Hyvärinen (2020) to measure the goodness of identifiability under permutation by introducing cost optimization to assign each learned component to the source component. This measurement is aligned with the goal of $\sim_p$-identifiability, which allows us to distinguish $S$ from $Z$. Table 5.1 shows the superiority of our LaCIM-$d$, LaCIM-$d_s$ over pool-LaCIM in terms of the CIMe relating to $S, Z$ under permutation, by means of multiple diverse experiments. Besides, we consider LaCIM-$d$ on $m = 3, 5, 7$ with the same total number of samples. It yields that more environments can perform better; and that even $m = 3$ still performs much better than pool-LaCIM. To illustrate the learning effect, we visualize the learned $Z$ in Fig. 7.8, with $S$ left in supplement 7.8 due to space limit.

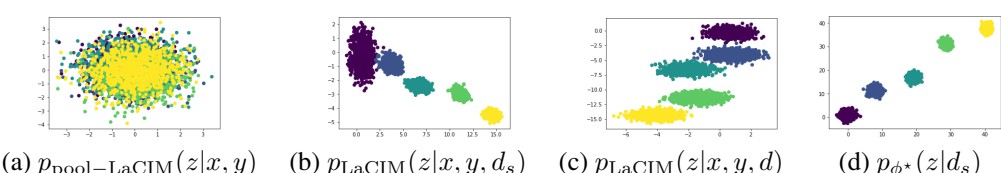

(a) $p_{\text{pool}-\text{LaCIM}}(z|x, y)$    (b) $p_{\text{LaCIM}}(z|x, y, d_s)$    (c) $p_{\text{LaCIM}}(z|x, y, d)$    (d) $p_{\phi^\star}(z|d_s)$

Figure 2: Visualization of $Z$. From left to right are: estimated posterior by pool-LaCIM, LaCIM-$d_s$, LaCIM-$d$ and the ground-truth. As shown, the LaCIM-$d_s$ (Fig.(b)) and LaCIM-$d$ (Fig.(c)) can identify the ground truth distribution of $Z$ (*i.e.*, $p_{\phi^\star}(z|d_s)$) up to permutation and point-wise transformation, which validates the claim in theorem 4.3.

## 5.2 REAL-WORLD OOD CHALLENGE

We present our LaCIM's results on three OOD tasks, with different environments associated with different values of $d_s$. We implement both versions of LaCIM, *i.e.*, LaCIM-$d_s$ and LaCIM-$d$, with task-dependent definition of $d_s$. In CMNIST, the $d_s$ (digit color) is a fully observed confounder, and LaCIM-$d_s$ in this case is the ceiling of LaCIM-$d$ under the same implementation. In NICO and ADNI, the LaCIM-$d$ even outperform LaCIM-$d_s$, when the source variables are only partially observed.

**Dataset.** We describe the datasets as follows (the $X$ denote image; the $Y$ denote label):

*NICO*: we evaluate the cat/dog classification in "Animal" dataset in NICO, a benchmark for non-i.i.d problem in He et al. (2019). Each animal is associated with "grass","snow" contexts with different proportions, denoted as $d_s \in \mathbb{R}^4$ (cat,dog in grass,snow). We set $m = 8$ and $m = 14$. The $C, Z, S$ respectively denote the (time,whether) of sampling, the context and semantic shape of cat/dog.

*CMNIST*: We relabel the digits 0-4 and 5-9 as $y = 0$ and $y = 1$, based on MNIST. Then we color $p^e (1 - p^e)$ of images with $y = 0$ ($y = 1$) as green and color others as red. We set $m = 2$ with $p^{e_1} = 0.9, p^{e_2} = 0.8$. The $d_s^e$ is $p^e$ to describe the intensity of spursiou correlation caused by color. We do not flip $y$ with 25% like Arjovsky et al. (2019) [2], since doing so will cause the digit correlated rather than causally related to the label, which is beyond our scope. The $Z, S$ respectively represent the color and number. The $C$ can also denote (time,whether) for which the painter draws the number and color, *e.g.*, the painter tends to draw red 0 more often than green 1 in the sunny morning.

*ADNI.* The data are obtained from the ADNI databaset, the $\mathcal{Y} := \{0, 1, 2\}$ with 0,1,2 respectively denoting AD, Mild Cognitive Impairment (MCI) and Normal Control (NC). The $X$ is Magnetic resonance imaging (sMRI). We set $m = 2$. We consider two types of $d_s$: Age and TAU (a biomarker Humpel and Hochstrasser (2011)). The $S$ ($Z$) denote the disease-related (-unrelated) brain regions. The $C$ denotes the hormone level that can affect the brain structure development.

**Compared Baselines.** We compare with (i) Cross-Entropy (CE) from $X \rightarrow Y$ (CE $X \rightarrow Y$), (ii) domain-adversarial neural network (DANN) for domain adaptation Ganin et al. (2016), (iii) Maximum Mean Discrepancy with Adversarial Auto-Encoder (MMD-AAE) for domain generalization Li et al. (2018), (iv) Domain Invariant Variational Autoencoders (DIVA) Ilse et al. (2019) (v) Selecting Data Augmentation (SDA) Ilse et al. (2020), (vi) Invariant Risk Mnimization (IRM) Arjovsky et al. (2019), (vii) CE $(X, d_s) \rightarrow Y$, (viii) VAE with causal graph $C \rightarrow V \rightarrow \{X, Y\}$ with $V$ mixing $S, Z$ and we call it sVAE for simplicity. We only implement SDA on CMNIST, since the intervened-data generation of SDA requires explicitly extracting the $S, Z$, which is intractable in ADNI and NICO. For fair comparison, we keep the model capacity (numer of parameters) in the same level.

**Implementation Details.** For each domain $e$, we implement the reparameterization with $\rho_s^e, \rho_z^e$: $s', z' = \rho_s^e(s), \rho_z^e(z)$, to transform the $p^e(s, z)$ into isotropic Gaussian; then the generative models are correspondingly modified as $\{p_\phi(x|(\rho_s^e)^{-1}(s), (\rho_z^e)^{-1}(z)), p_\phi(y|(\rho_s^e)^{-1}(s))\}$ according to rule of change of variables. The optimized parameters are $\{\{q_\psi^e(s, z|x)\}_e, p_\phi(x|s, z), p_\phi(y|s), \{\rho_{t=s,z}^e\}_e\}$, with the encoder $q_\psi^e(s, z|x)$ being sequentially composed of: *i)* the sequential of Conv-BN-ReLU-MaxPool blocks that shared among $\mathcal{E}_{\text{train}}$, followed by *ii)* the sequential of ReLU-FC for the mean and log-variance of $S, Z$ that are specific to $e$. The structure of $\rho_{t=s,z}^e$ is FC-ReLU-FC. The decoder $p_\phi(x|s, z)$ is the sequential of upsampling, several TConv-BN-ReLU blocks and Sigmoid. The predictor $p_\phi(y|s)$ is sequential of FC→BN→ReLU blocks, followed by Softmax (or Sigmoid) for classification. The network structure and the output channel size for CMNIST, NICO and ADNI are introduced in supplement 7.11, 7.12, 7.13, Tab. 13, 14. We implement SGD as optimizer: with learning rate (lr) 0.5 and weight decay (wd) $1e$-5 for CMNIST; lr 0.01 with decaying $0.2\times$ every 60 epochs, wd $5e$-5 for NICO and ADNI (wd is $2e$-4). The batch-size are set to 256, 30 and 4 for CMNIST, NICO, ADNI. The "FC", "BN" stand for Fully-Connected, Batch-Normalization.

**Results.** We report accuracy over three runs for each method. As shown in Tab. 2 [3] our LaCIM-$d$ performs comparable and better than others on all applications, except the 99.3 achieved by SDA on CMNIST, which is comparable to the result on the original MNIST. This is because during training,

---

[2]We also conduct this experiment with flipping $y$ in supplementary 7.11.

[3]On NICO, we implement ConvNet with Batch Balancing as a specifically benchmark in He et al. (2019). The results are $60 \pm 1$ on $m = 8$ and $62.33 \pm 3.06$ on $m = 14$.

the SDA implemented data augmentation with random colors, which decorrelate the color-label. When $S$ cannot be explicitly extracted in general case, the SDA is not tractable.

**Discussions.** The advantage over invariant learning method (IRM) and CE $(X, d_s) \rightarrow Y$ which also takes $d_s$ into prediction can be contributed to the identification of true causal mechanisms. Further, the improvement over sVAE is benefited from our separation of $y$-causative factors (*a.k.a*, $S$) from others to avoid spurious correlation. Besides, as shown from results on

Table 2: Accuracy (%) of OOD prediction. Average over three runs.

| Dataset / Method | NICO | | CMNIST | ADNI ($m = 2$) | |
|---|---|---|---|---|---|
| | $m = 8$ | $m = 14$ | $m = 2$ | $C$ : Age | $C$ : TAU |
| CE $X \rightarrow Y$ | $60.67 \pm 2.52$ | $59.00 \pm 1.73$ | $97.87 \pm 0.19$ | $63.06 \pm 2.26$ | $64.58 \pm 0.90$ |
| DANN | $59.33 \pm 4.93$ | $60.00 \pm 2.65$ | $97.42 \pm 0.13$ | $60.84 \pm 1.83$ | $64.58 \pm 0.90$ |
| MMD-AAE | $61.33 \pm 2.89$ | $66.33 \pm 3.21$ | $81.23 \pm 7.80$ | $62.43 \pm 2.42$ | $65.62 \pm 0.00$ |
| DIVA | $60.67 \pm 2.08$ | $58.67 \pm 1.53$ | $97.97 \pm 0.19$ | $61.37 \pm 3.30$ | $65.10 \pm 0.90$ |
| SDA | - | - | $99.37 \pm 0.03$ | - | - |
| IRM | $61.67 \pm 4.16$ | $65.00 \pm 3.00$ | $98.18 \pm 0.22$ | $63.49 \pm 1.59$ | $65.10 \pm 0.90$ |
| CE $X, d_s \rightarrow Y$ | $57.33 \pm 6.03$ | $64.00 \pm 1.00$ | $98.03 \pm 0.27$ | $62.43 \pm 0.92$ | $65.62 \pm 0.00$ |
| sVAE | $59.67 \pm 3.79$ | $64.33 \pm 0.58$ | $97.89 \pm 0.61$ | $63.67 \pm 1.87$ | $66.67 \pm 0.91$ |
| LaCIM-$d_s$ (**Ours**) | $62.00 \pm 1.73$ | $68.00 \pm 2.64$ | $98.81 \pm 0.14$ | $\mathbf{65.08 \pm 1.59}$ | $66.14 \pm 0.91$ |
| LaCIM-$d$ (**Ours**) | $\mathbf{62.67 \pm 0.58}$ | $\mathbf{68.67 \pm 2.64}$ | $98.78 \pm 0.20$ | $64.44 \pm 0.96$ | $\mathbf{68.23 \pm 0.90}$ |

NICO, a larger $m$ (with the total number of samples $n$ fixed) can bring further benefit, which may due to the easier satisfaction of the diversity condition in theorem 4.3. One thing worth particular mention is that on NICO and ADNI (when $d_s$ denotes TAU), our LaCIM-$d$ performs comparable and even better than LaCIM-$d_s$, due to the existence of unobserved partial variables. For example, each $d_s$ only contains one attribute each time in ADNI. For completeness, we conduct experiments with fully observed confounders in supplement 7.13. Besides, we apply our method on intervened data, the result of which can validate more robustness of LaCIM, as shown in supplement 7.12.

**Interpretability.** We visualize learned $S$ as side proof of interpretability. Specifically, we select the $s^*$ that has the highest correlation with $y$ among all dimension of $S$, and visualize the derivatives of $s^*$ with respect to the image. For CE $x \rightarrow y$ and CE $(x, d_s) \rightarrow y$, we visualize the derivatives of predicted class scores with respect to the image. As shown in Fig. 5.2, LaCIM (the 4th column) can identify more explainable semantic features, which verifies the identifiability and effectiveness of the learning method. Supplement 7.12 provides more results.

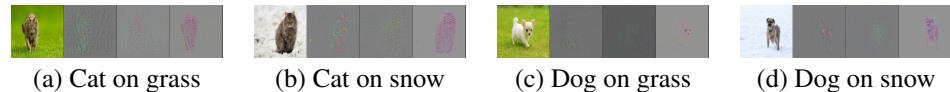

(a) Cat on grass        (b) Cat on snow        (c) Dog on grass        (d) Dog on snow

Figure 3: Visualization via gradient Simonyan et al. (2013). From the left to right: original image, CE $X \rightarrow Y$, CE $(X, d_s) \rightarrow Y$ and LaCIM-$d_s$.

## 5.3 ROBUSTNESS ON SECURITY

We consider the DeepFake-related security problem, which targets on detecting small perturbed fake images that can spread fake news. The Rossler et al. (2019) provides FaceForensics++ dataset from 1000 Youtube videos for training and 1,000 benchmark images from other sources (OOD) for testing. We split the train data into $m = 2$ environments according to video ID. The considerable result in Tab. 5.3 verifies potential value on security.

Table 3: Accuracy (%) of robustness on Face-Forensics++. Average over three runs.

| CE $X \rightarrow Y$ | IRM | LaCIM-$d$ (**Ours**) |
|---|---|---|
| $82.8 \pm 0.99$ | $83.4 \pm 0.59$ | $\mathbf{84.47 \pm 0.90}$ |

## 6 CONCLUSIONS & DISCUSSIONS

We incorporate the causal structure as prior knowledge in proposed LaCIM, by introducing: (i) latent variables and explicitly separate them into $y$-causative factors (*a.k.a*, $S$) and others (*a.k.a*, $Z$) which are spuriously correlated with the output; (ii) the source variable $d_s$ that explains the distributional inconsistency among domains. When the environments are diverse and much enough, we can successfully identify the causal invariant mechanisms, and also $y$-causative factors for prediction without a mix of others. Our LaCIM shows potential value regarding robustness to OOD tasks with confounding bias, selection bias and others such as healthcare and security. A possible drawback of our model lies in our requirement of the number of environments (which may be not satisfied in some scenarios) for identifiability, and the relaxation of which is left in the future work.

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

## 7 SUPPLEMENTARY MATERIALS

### 7.1 O.O.D GENERALIZATION ERROR BOUND

Denote $\mathbb{E}_p[y|x] := \int_{\mathcal{Y}} y p(y|x) dy$ for any $x, y \in \mathcal{X} \times \mathcal{Y}$. We have $\mathbb{E}_{p^e}[y|s] = \int_{\mathcal{Y}} y p(y|s) dy$ according to that $p(y|s)$ is invariant across $\mathcal{E}$, we can omit $p^e$ in $\mathbb{E}_{p^e}[y|s]$ and denote $g(S) := \mathbb{E}[Y|S]$. Then, the OOD bound $\left|\mathbb{E}_{p^{e_1}}(y|x) - \mathbb{E}_{p^{e_2}}(y|x)\right|$, $\forall(x, y)$ is bounded as follows:

**Theorem 7.1** (OOD genearlization error). *Consider two LaCIM $P^{e_1}$ and $P^{e_2}$, suppose that their densities , i.e., $p^{e_1}(s|x)$ and $p^{e_2}(s|x)$ are absolutely continuous having support $(-\infty, \infty)$. For any $(x, y) \in \mathcal{X} \times \mathcal{Y}$, assume that*

- *$g(S)$ is a Lipschitz-continuous function;*
- *$\pi_x(s) := \frac{p^{e_2}(s|x)}{p^{e_1}(s|x)}$ is differentiable and $\mathbb{E}_{p^{e_1}}\left[\pi_x(S)\big|g(S) - \mu_1\big|\right] < \infty$ with $\mu_1 := \mathbb{E}_{p^{e_1}}[g(S)|X = x] = \int_{\mathcal{S}} g(s) p^{e_1}(s|x) ds$;*

*then we have $\left|\mathbb{E}_{p^{e_1}}(y|x) - \mathbb{E}_{p^{e_2}}(y|x)\right| \le \|g'\|_\infty \|\pi'_x\|_\infty \mathrm{Var}_{p^{e_1}}(S|X = x)$.*

When $e_1 \in \mathcal{E}_{\text{train}}$ and $e_2 \in \mathcal{E}_{\text{test}}$, the theorem 7.1 describes the error during generalization on $e_2$ for the strategy that trained on $e_1$. The bound is mainly affected by: (i) the Lipschitz constant of $g$, i.e., $\|g\|_\infty$; (ii) $\|\pi'_x\|_\infty$ which measures the difference between $p^{e_1}(s, z)$ and $p^{e_2}(s, z)$; and (iii) the $\mathrm{Var}_{p^{e_1}}(S|x)$ that measures the intensity of $x \to (s, z)$. These terms can be roughly categorized into two classes: (i),(iii) which are related to the property of CIMe and gave few space for improvement; and the (ii) that describes the distributional change between two environments. Specifically for the first class, the (i) measures the smoothness of $\mathbb{E}(y|s)$ with respect to $s$. The smaller value of $\|g'\|_\infty$ implies that the flatter regions give rise to the same prediction result, hence easier transfer from $e_1$ to $e_2$ and vice versa. For the term (iii), consider the deterministic setting that $\varepsilon_x = 0$ (leads to $\mathrm{Var}_{p^{e_1}}(S|x) = 0$), then $s$ can be determined from $x$ for generalization if the $f$ is bijective function.

The term (ii) measures the distributional change between posterior distributions $p^{e_1}(s|x)$ and $p^{e_2}(s|x)$, which contributes to the difference during prediction: $\left|\mathbb{E}_{p^{e_1}}(y|x) - \mathbb{E}_{p^{e_2}}(y|x)\right| = \int_{\mathcal{S}}(p^{e_1}(s|x) - p^{e_1}(s|x)) p_{f_y}(y|s) ds$. Such a change is due to the inconsistency between priors $p^{e_1}(s, z)$ and $p^{e_2}(s, z)$, which is caused by different value of the confounder $d_s$.

*Proof.* In the following, we will derive the upper bound

$$\left|\mathbb{E}_{p^{e_1}}[Y|X = x] - \mathbb{E}_{p^{e_2}}[Y|X = x]\right| \le \|g'\|_\infty \|\pi'_x\|_\infty \mathrm{Var}_{p^{e_1}}(S|X = x),$$

where $\pi_x(s) =: \frac{p^{e_2}(s|x)}{p^{e_1}(s|x)}$ and $g(s)$ is assumed to be Lipschitz-continuous.

To begin with, note that

$$\mathbb{E}[Y|X] = \mathbb{E}[\mathbb{E}(Y|X, S)|X] = \mathbb{E}[g(S)|X] = \int g(s) p(s|x) ds.$$

Let $p_1(s|x) = p^{e_1}(s|x)$, $p_2(s|x) = p^{e_2}(s|x)$. For ease of notations, we use $P_1$ and $P_2$ denote the distributions with densities $p_1(s|x)$ and $p_2(s|x)$ and suppose $S_1 \sim P_1$ and $S_2 \sim P_2$, where $x$ is omitted as the following analysis is conditional on a fixed $X = x$.

Then we may rewrite the difference of conditional expectations as

$$\mathbb{E}_{p^{e_2}}[Y|X = x] - \mathbb{E}_{p^{e_1}}[Y|X = x] = \mathbb{E}(g(S_2)) - \mathbb{E}(g(S_1)),$$

where $\mathbb{E}[g(S_j))] = \int g(s) p_j(s|x) ds$ denotes the expectation over $P_j$.

Let $\mu_1 := \mathbb{E}_{p^{e_1}}[g(S)|X = x] = \mathbb{E}[g(S_1)] = \int g(s) p_1(s|x) ds$. Then

$$\mathbb{E}_{p^{e_2}}[Y|X = x] - \mathbb{E}_{p^{e_1}}[Y|X = x] = \mathbb{E}(g(S_2)) - \mathbb{E}(g(S_1)) = \mathbb{E}[g(S_2) - \mu_1].$$

Further, we have the following transformation

$$\mathbb{E}[g(S_2) - \mu_1] = \int (g(s) - \mu_1) \pi_x(s) p_1(s|x) ds = \mathbb{E}[(g(S_1) - \mu_1) \pi_x(S_1)]. \tag{3}$$

In the following, we will use the results of the Stein kernel function. Please refer to Definition 7.2 for a general definition. Particularly, for the distribution $P_1 \sim p_1(s|x)$, the Stein kernel $\tau_1(s)$ is

$$\tau_1(s) = \frac{1}{p_1(s|x)} \int_{-\infty}^{s} (\mathbb{E}(S_1) - t)p_1(t|x)dt, \tag{4}$$

where $\mathbb{E}(S_1) = \int s \cdot p_1(s|x)ds$. Further, we define $(\tau_1 \circ g)(s)$ as

$$(\tau_1 \circ g)(s) = \frac{1}{p_1(s|x)} \int_{-\infty}^{s} (\mathbb{E}(g(S_1)) - g(t))p_1(t|x)dt = \frac{1}{p_1(s|x)} \int_{-\infty}^{s} (\mu_1 - g(t))p_1(t|x)dt. \tag{5}$$

Under the second condition listed in Theorem 7.1, we may apply the result of Lemma 7.3. Specifically, by the equation (8), we have

$$\mathbb{E}\left[(g(S_1) - \mu_1)\pi_x(S_1)\right] = \mathbb{E}\left[(\tau_1 \circ g)(S_1)\pi_x'(S_1)\right].$$

Then under the first condition in Theorem 7.1, we can obtain the following inequality by Lemma 7.4,

$$\mathbb{E}\left[(\tau_1 \circ g)(S_1)\pi_x'(S_1)\right] = \mathbb{E}\left[\left(\frac{(\tau_1 \circ g)}{\tau_1}\pi_x'\tau_1\right)(S_1)\right] \leq \mathbb{E}\left[\left|\frac{(\tau_1 \circ g)}{\tau_1}(S_1)\right| \cdot \left|\pi_x'\tau_1(S_1)\right|\right]$$

$$\leq \|g'\|_\infty \mathbb{E}\left[|(\pi_x'\tau_1)(S_1)|\right] \leq \|g'\|_\infty \|\pi_x'\|_\infty \mathbb{E}\left[|\tau_1(S_1)|\right]. \tag{6}$$

In the following, we show that the Stein kernel is non-negative, which enables $\mathbb{E}\left[|\tau_1(S_1)|\right] = \mathbb{E}\left[\tau_1(S_1)\right]$. According to the definition, $\tau_1(s) = \frac{1}{p_1(s|x)} \int_{-\infty}^{s} (\mathbb{E}(S_1) - t)p_1(t|x)dt$, where $\mathbb{E}(S_1) = \int_{-\infty}^{\infty} t \cdot p_1(t|x)dt$. Let $F_1(s) = \int_{-\infty}^{s} p_1(t|x)dt$ be the distribution function for $P_1$. Note that

$$\int_{-\infty}^{s} \mathbb{E}(S_1)p_1(t|x)dt = F_1(s)\mathbb{E}(S_1) = F_1(s)\mathbb{E}(S_1),$$

$$\int_{-\infty}^{s} tp_1(t|x)dt = F_1(s)\int_{-\infty}^{s} t\frac{p_1(t|x)}{F_1(s)}dt = F_1(s)\mathbb{E}(S_1|S_1 \leq s) \leq F_1(s)\mathbb{E}(S_1),$$

The last inequality is based on $\mathbb{E}(S_1|S_1 \leq s) - \mathbb{E}(S_1) \leq 0$ that can be proved as the following

$$\int_{-\infty}^{s} t\frac{p_1(t|x)}{F_1(s)}dt - \int_{-\infty}^{\infty} tp_1(t|x)dt = \int_{-\infty}^{s} t\left(\frac{1}{F_1(s)} - 1\right)p_1(t|x)dt - \int_{s}^{\infty} tp_1(t|x)dt$$

$$\leq s\int_{-\infty}^{s} \left(\frac{1}{F_1(s)} - 1\right)p_1(t|x)dt - s\int_{s}^{\infty} p_1(t|x) = 0.$$

Therefore, $\tau_1(s) \geq 0$ and hence $\mathbb{E}\left[|\tau_1(S_1)|\right] = \mathbb{E}\left[\tau_1(S_1)\right]$ in (6).

Besides, by equation (9), the special case of Lemma 7.3, we have

$$\mathbb{E}\left[\tau_1(S_1)\right] = \mathrm{Var}(S_1) = \mathrm{Var}_{p^{e_1}}(S|X = x).$$

To sum up,

$$\mathbb{E}\left[(\tau_1 \circ g)(S_1)\pi_x'(S_1)\right] \leq \|g'\|_\infty \|\pi_x\|_\infty \mathbb{E}\left[\tau_1(S_1)\right] = \|g'\|_\infty \|\pi_x'\|_\infty \mathrm{Var}_{p^{e_1}}(S|X = x).$$

$\square$

**Definition 7.2** (**the Stein Kernel $\tau_P$ of distribution $P$**). Suppose $X \sim P$ with density $p$. The Stein kernel of $P$ is the function $x \mapsto \tau_P(x)$ defined by

$$\tau_P(x) = \frac{1}{p(x)} \int_{-\infty}^{x} (\mathbb{E}(X) - y)p(y)dy, \tag{7}$$

where Id is the identity function for $\mathrm{Id}(x) = x$. More generally, for a function $h$ satisfying $\mathbb{E}[|h(X)|] < \infty$, define $(\tau_P \circ h)(x)$ as

$$(\tau_P \circ h)(x) = \frac{1}{p(x)} \int_{-\infty}^{x} (\mathbb{E}(h(X)) - h(y))p(y)dy.$$

**Lemma 7.3.** *For a differentiable function $\varphi$ such that $\mathbb{E}[|(\tau_P \circ h)(x)\varphi'(X)|] < \infty$, we have*

$$\mathbb{E}\left[(\tau_P \circ h)(x)\varphi'(X)\right] = \mathbb{E}[(h(X) - \mathbb{E}(h(X))\varphi(X)]. \tag{8}$$

*Proof.* Let $\mu_h =: \mathbb{E}(h(X))$. As $\mathbb{E}(h(X) - \mu_h) = 0$,

$$(\tau_P \circ h)(x) = \frac{1}{p(x)} \int_{-\infty}^{x} (\mu_h - h(y))p(y)dy = \frac{-1}{p(x)} \int_{x}^{\infty} (\mu_h - h(y))p(y)dy.$$

Then

$$\mathbb{E}\left[(\tau_P \circ h)(x)\varphi'(X)\right] = \int_{-\infty}^{0} (\tau_P \circ h)(x)\varphi'(x)p(x)dx + \int_{0}^{\infty} (\tau_P \circ h)(x)\varphi'(x)p(x)dx$$

$$= \int_{-\infty}^{0} \int_{-\infty}^{x} (\mu_h - h(y))p(y)\varphi'(x)dydx - \int_{0}^{\infty} \int_{x}^{\infty} (\mu_h - h(y))p(y)\varphi'(x)dydx$$

$$= \int_{-\infty}^{0} \int_{y}^{0} (\mu_h - h(y))p(y)\varphi'(x)dxdy - \int_{0}^{\infty} \int_{0}^{y} (\mu_h - h(y))p(y)\varphi'(x)dxdy$$

$$= \int_{-\infty}^{0} \int_{0}^{y} (h(y) - \mu_h)p(y)\varphi'(x)dxdy + \int_{0}^{\infty} \int_{0}^{y} (h(y) - \mu_h)p(y)\varphi'(x)dxdy$$

$$= \int_{-\infty}^{\infty} (h(y) - \mu_h)p(y) \left( \int_{0}^{y} \varphi'(x)dx \right) dy = \int_{-\infty}^{\infty} (h(y) - \mu_h)p(y)(\varphi(y) - \varphi(0))dy$$

$$= \int_{-\infty}^{\infty} (h(y) - \mu_h)p(y)(\varphi(y))dy = \mathbb{E}[(h(X) - \mathbb{E}(h(X))\varphi(X)]$$

Particularly, taking $h(X) = X$ and $\varphi(X) = X - \mathbb{E}(X)$, we immediately have

$$\mathbb{E}(\tau_P(X)) = \text{Var}(X) \tag{9}$$

$\square$

**Lemma 7.4.** *Assume that $\mathbb{E}(|X|) < \infty$ and the density $p$ is locally absolutely continuous on $(-\infty, \infty)$ and $h$ is a Lipschitz continuous function. Then we have $|f_h| \le \|h'\|_\infty$ for*

$$f_h(x) = \frac{(\tau_P \circ h)(x)}{\tau_P(x)} = \frac{\int_{-\infty}^{x} (\mathbb{E}(h(X)) - h(y))p(y)dy}{\int_{-\infty}^{x} (\mathbb{E}(X) - y)p(y)dy}.$$

*Proof.* This is a special case of Corollary 3.15 in Döbler et al. (2015), taking the constant $c = 1$. $\square$

### 7.2 PROOF OF THE EQUIVALENCE OF DEFINITION 4.2

**Proposition 7.5.** *The binary relation $\sim_p$ defined in Def. 4.2 is an equivalence relation.*

*Proof.* The equivalence relation should satisfy three properties as follows:

- *Reflexive* property: The $\theta \sim_p \theta$ with $M_z$, $M_s$ being identity matrix and $a_s$, $a_z$ being 0.

- *Symmtric* property: If $\theta \sim_p \tilde{\theta}$, then there exists block permutation matrices $M_z$ and $M_s$ such that

$$\mathbf{T}^s([f_x]_{\mathcal{S}}^{-1}(x)) = M_s \tilde{\mathbf{T}}^s([\tilde{f}_x]_{\mathcal{S}}^{-1}(x)) + a_s, \quad \mathbf{T}^z([f_x]_{\mathcal{Z}}^{-1}(x)) = M_z \tilde{\mathbf{T}}^z([\tilde{f}_x]_{\mathcal{Z}}^{-1}(x)) + a_z,$$

$$p_{f_y}(y|[f_x]_{\mathcal{S}}^{-1}(x)) = p_{\tilde{f}_y}(y|[\tilde{f}_x]_{\mathcal{S}}^{-1}(x)).$$

The we have $M_s^{-1}$ and $M_z^{-1}$ are also block permutation matrices and such that:

$$\tilde{\mathbf{T}}^s([\tilde{f}_x]_{\mathcal{S}}^{-1}(x)) = M_s^{-1}\mathbf{T}^s([f_x]_{\mathcal{S}}^{-1}(x)) + (-a_s), \quad \tilde{\mathbf{T}}^s([\tilde{f}_x]_{\mathcal{Z}}^{-1}(x)) = M_z^{-1}\mathbf{T}^s([f_x]_{\mathcal{Z}}^{-1}(x)) + (-a_z),$$

$$p_{\tilde{f}_y}(y|[\tilde{f}_x]_{\mathcal{S}}^{-1}(x)) = p_{f_y}(y|[f_x]_{\mathcal{S}}^{-1}(x)).$$

Therefore, we have $\tilde{\theta} \sim_p \theta$.

- *Transitive* property: if $\theta_1 \sim_p \theta_2$ and $\theta_2 \sim_p \theta_3$ with $\theta_i := \{f_x^i, f_y^i, \mathbf{T}^{s,1}, \mathbf{T}^{z,1}, \mathbf{\Gamma}^{s,i}, \mathbf{\Gamma}^{z,i}\}$, then we have

$$\mathbf{T}^{s,1}((f_{x,s}^1)^{-1}(x)) = M_s^1 \mathbf{T}^{s,2}((f_{x,s}^2)^{-1}(x)) + a_s^1,$$

$$\mathbf{T}^{z,1}((f_{x,z}^1)^{-1}(x)) = M_z^1 \mathbf{T}^{z,2}((f_{x,z}^2)^{-1}(x)) + a_z^2,$$

$$\mathbf{T}^{s,2}((f_{x,s}^2)^{-1}(x)) = M_s^2 \mathbf{T}^{s,3}((f_{x,s}^3)^{-1}(x)) + a_s^2,$$

$$\mathbf{T}^{z,2}((f_{x,z}^2)^{-1}(x)) = M_z^2 \mathbf{T}^{z,3}((f_z^3)^{-1}(x)) + a_{x,z}^3$$

for block permutation matrices $M_s^1, M_z^1, M_s^2, M_z^2$ and vectors $a_s^1, a_s^2, a_z^1, a_z^2$. Then we have

$$\mathbf{T}^{s,1}((f_{x,s}^1)^{-1}(x)) = M_s^2 M_s^1 \mathbf{T}^{s,3}((f_{x,s}^3)^{-1}(x)) + (M_s^2 a_s^1) + a_s^2,$$

$$\mathbf{T}^{z,1}((f_{x,z}^1)^{-1}(x)) = M_z^2 M_z^1 \mathbf{T}^{z,3}((f_{x,z}^3)^{-1}(x)) + (M_z^2 a_z^1) + a_z^2.$$

Besides, it is apparent that

$$p_{f_y^1}(y|(f_x^1)_s^{-1}(x)) = p_{f_y^2}(y|(f_x^2)_s^{-1}(x)) = p_{f_y^3}(y|(f_x^3)_s^{-1}(x)). \tag{10}$$

Therefore, we have $\theta_1 \sim_p \theta_3$ since $M_s^2 M_s^1$ and $M_z^2 M_z^1$ are also permutation matrices.

With above three properties satisfied, we have that $\sim_p$ is a equivalence relation. $\qquad\square$

### 7.3 PROOF OF THEOREM 4.3

In the following, we write $p^e(x, y)$ as $p(x, y|d^e)$ and also $\Gamma_c^{t=s,z} := \Gamma_c^{t=s,z}(d^e), S_{c,i} = S_i(d^e), Z_{c,i} = Z_i(d^e)$. To prove the theorem 4.3, we first prove the theorem 7.6 for the simplest case when $c|d^e = d^e$, then we generalize to the case when $\mathcal{C} := \cup_r \mathcal{C}_r$. The overall roadmap is as follows: we first prove the $\sim_A$-identifiability in theorem 7.9, and the combination of which with lemma 7.12, 7.11 give theorem 7.6 in the simplest case when $c|d^e = d^e$. Then we generalize the case considered in theorem 7.6 to the more general case when $\mathcal{C} := \cup_r \mathcal{C}_r$.

**Theorem 7.6** ($\sim_p$-identifiability). *For $\theta$ in the LaCIM $p_\theta^e(x, y) \in \mathcal{P}_{\exp}$ for any $e \in \mathcal{E}_{\text{train}}$, we assume that **(1)** the CIMe satisfies that $f_x, f_x'$ and $f_x''$ are continuous and that $f_x, f_y$ are bijective; **(2)** that the $T_{i,j}^t$ are twice differentiable for any $t = s, z, i \in [q_t], j \in [k_t]$; **(3)** the exogenous variables satisfy that the characteristic functions of $\varepsilon_x, \varepsilon_y$ are almost everywhere nonzero; **(4)** the number of environments, i.e., $m \geq \max(q_s * k_s, q_z * k_z) + 1$ and $\left[\mathbf{\Gamma}_{d^{e_2}}^{t=s,z} - \mathbf{\Gamma}_{d^{e_1}}^{t=s,z}, ..., \mathbf{\Gamma}_{d^{e_m}}^{t=s,z} - \mathbf{\Gamma}_{d^{e_1}}^{t=s,z}\right]$ have full column rank for both $t = s$ and $t = z$, we have that the parameters $\theta := \{f_x, f_y, \mathbf{T}^s, \mathbf{T}^z\}$ are $\sim_p$ identifiable.*

To prove theorem 7.6, We first prove the $\sim_A$-identifiability that is defined as follows:

**Definition 7.7** ($\sim_A$-identifiability). *The definition is the same with the one defined in 4.2, with $M_s, M_z$ being invertible matrices which are not necessarily to be the permutation matrices in Def. 4.2.*

**Proposition 7.8.** *The binary relation $\sim_A$ defined in Def. 7.7 is an equivalence relation.*

*Proof.* The proof is similar to that of proposition 7.5. $\qquad\square$

The following theorem states that any LaCIM that belongs to $\mathcal{P}_{\exp}$ is $\sim_A$-identifiable.

**Theorem 7.9** ($\sim_A$-identifiability). *For $\theta$ in the LaCIM $p_\theta^e(x, y) \in \mathcal{P}_{\exp}$ for any $e \in \mathcal{E}_{\text{train}}$, we assume **(1)** the CIMe satisfies that $f_x, f_y$ are bijective; **(2)** the $T_{i,j}^t$ are twice differentiable for any $t = s, z, i \in [q_t], j \in [k_t]$; **(3)** the exogenous variables satisfy that the characteristic functions of $\varepsilon_x, \varepsilon_y$ are almost everywhere nonzero; **(4)** the number of environments, i.e., $m \geq \max(q_s * k_s, q_z * k_z) + 1$ and $\left[[\mathbf{\Gamma}_{d^{e_2}}^t - \mathbf{\Gamma}_{d^{e_1}}^t]^\mathsf{T}, ..., [\mathbf{\Gamma}_{d^{e_m}}^t - \mathbf{\Gamma}_{d^{e_1}}^t]^\mathsf{T}\right]^\mathsf{T}$ have full column rank for $t = s, z$, we have that the parameters $\{f_x, f_y, \mathbf{T}^s, \mathbf{T}^z\}$ are $\sim_p$ identifiable.*

*Proof.* Suppose that $\theta = \{f_x, f_y, \mathbf{T}^s, \mathbf{T}^z\}$ and $\tilde{\theta} = \{\tilde{f}_x, \tilde{g}_y, \tilde{\mathbf{T}}^s, \tilde{\mathbf{T}}^z\}$ share the same observational distribution for each environment $e \in \mathcal{E}_{\text{train}}$, i.e.,

$$p_{f_x, f_y, \mathbf{T}^s, \mathbf{\Gamma}^s, \mathbf{T}^z, \mathbf{\Gamma}^z}(x, y|d^e) = p_{\tilde{f}_x, \tilde{f}_y, \tilde{\mathbf{T}}^s, \tilde{\mathbf{\Gamma}}^s, \tilde{\mathbf{T}}^z, \tilde{\mathbf{\Gamma}}^z}(x, y|d^e). \tag{11}$$

Then we have

$$p_{f_x, f_y, \mathbf{T}^s, \mathbf{\Gamma}^s, \mathbf{T}^z, \mathbf{\Gamma}^z}(x|d^e) = p_{\tilde{f}_x, \tilde{f}_y, \tilde{\mathbf{T}}^s, \tilde{\mathbf{\Gamma}}^s, \tilde{\mathbf{T}}^z, \tilde{\mathbf{\Gamma}}^z}(x|d^e) \tag{12}$$

$$\implies \int_{\mathcal{S}\times\mathcal{Z}} p_{f_x}(x|s,z) p_{\mathbf{T}^s, \mathbf{\Gamma}^s, \mathbf{T}^z, \mathbf{\Gamma}^z}(s,z|d^e) ds dz = \int_{\mathcal{S}\times\mathcal{Z}} p_{\tilde{f}_x}(x|s,z) p_{\tilde{\mathbf{T}}^s, \tilde{\mathbf{\Gamma}}^s, \tilde{\mathbf{T}}^z, \tilde{\mathbf{\Gamma}}^z}(s,z|d^e) ds dz \tag{13}$$

$$\implies \int_{\mathcal{X}} p_{\varepsilon_x}(x-\bar{x}) p_{\mathbf{T}^s, \mathbf{\Gamma}^s, \mathbf{T}^z, \mathbf{\Gamma}^z}(f_x^{-1}(\bar{x})|d^e) \text{vol} J_{f_x^{-1}}(\bar{x}) d\bar{x} \tag{14}$$

$$= \int_{\mathcal{X}} p_{\varepsilon_x}(x-\bar{x}) p_{\tilde{\mathbf{T}}^s, \tilde{\mathbf{\Gamma}}^s, \tilde{\mathbf{T}}^z, \tilde{\mathbf{\Gamma}}^z}(\tilde{f}_x^{-1}(\bar{x})|d^e) \text{vol} J_{\tilde{f}_x^{-1}}(\bar{x}) d\bar{x} \tag{15}$$

$$\implies \int_{\mathcal{X}} \tilde{p}_{\mathbf{T}^s, \mathbf{\Gamma}^s, \mathbf{T}^z, \mathbf{\Gamma}^z, f_x}(\bar{x}|d^e) p_{\varepsilon_x}(x-\bar{x}) d\bar{x} = \int_{\mathcal{X}} \tilde{p}_{\tilde{\mathbf{T}}^s, \tilde{\mathbf{\Gamma}}^s, \tilde{\mathbf{T}}^z, \tilde{\mathbf{\Gamma}}^z, \tilde{f}_x}(\bar{x}|d^e) p_{\varepsilon_x}(x-\bar{x}) d\bar{x} \tag{16}$$

$$\implies (\tilde{p}_{\mathbf{T}^s, \mathbf{\Gamma}^s, \mathbf{T}^z, \mathbf{\Gamma}^z, f_x} * p_{\varepsilon_x})(x|d^e) = (\tilde{p}_{\tilde{\mathbf{T}}^s, \tilde{\mathbf{\Gamma}}^s, \tilde{\mathbf{T}}^z, \tilde{\mathbf{\Gamma}}^z, \tilde{f}_x}) * p_{\varepsilon_x}(x|d^e) \tag{17}$$

$$\implies F[\tilde{p}_{\mathbf{T}^s, \mathbf{\Gamma}^s, \mathbf{T}^z, \mathbf{\Gamma}^z, f_x}](\omega) \varphi_{\varepsilon_x}(\omega) = F[\tilde{p}_{\tilde{\mathbf{T}}^s, \tilde{\mathbf{\Gamma}}^s, \tilde{\mathbf{T}}^z, \tilde{\mathbf{\Gamma}}^z, \tilde{f}_x}](\omega) \varphi_{\varepsilon_x}(\omega) \tag{18}$$

$$\implies F[\tilde{p}_{\mathbf{T}^s, \mathbf{\Gamma}^s, \mathbf{T}^z, \mathbf{\Gamma}^z, f_x}](\omega) = F[\tilde{p}_{\tilde{\mathbf{T}}^s, \tilde{\mathbf{\Gamma}}^s, \tilde{\mathbf{T}}^z, \tilde{\mathbf{\Gamma}}^z, \tilde{f}_x}](\omega) \tag{19}$$

$$\implies \tilde{p}_{\mathbf{T}^s, \mathbf{\Gamma}^s, \mathbf{T}^z, \mathbf{\Gamma}^z, f_x}(x|d^e) = \tilde{p}_{\tilde{\mathbf{T}}^s, \tilde{\mathbf{\Gamma}}^s, \tilde{\mathbf{T}}^z, \tilde{\mathbf{\Gamma}}^z, \tilde{f}_x}(x|d^e) \tag{20}$$

where $\text{vol} J_f(X) := \det(J_f(X))$ for any square matrix $X$ and function $f$ with "$J$" standing for the Jacobian. The $\tilde{p}_{\mathbf{T}^s, \mathbf{\Gamma}^s, \mathbf{T}^z, \mathbf{\Gamma}^z, f_x}(x)$ in Eq. (16) is denoted as $p_{\mathbf{T}^s, \mathbf{\Gamma}^s, \mathbf{T}^z, \mathbf{\Gamma}^z}(f_x^{-1}(x|d^e) \text{vol} J_{f^{-1}}(x)$. The '*' in Eq. (17) denotes the convolution operator. The $F[\cdot]$ in Eq. (18) denotes the Fourier transform, where $\phi_{\varepsilon_x}(\omega) = F[p_{\varepsilon_x}](\omega)$. Since we assume that the $\varphi_{\varepsilon_x}(\omega)$ is non-zero almost everywhere, we can drop it to get Eq. (20). Similarly, we have that:

$$p_{f_y, \mathbf{T}^s, \mathbf{\Gamma}^s}(y|d^e) = p_{\tilde{f}_y, \tilde{\mathbf{T}}^s, \tilde{\mathbf{\Gamma}}^s}(y|d^e) \tag{21}$$

$$\implies \int_{\mathcal{S}} p_{f_y}(y|s) p_{\mathbf{T}^s, \mathbf{\Gamma}^s}(s|d^e) ds = \int_{\mathcal{S}} p_{\tilde{f}_y}(y|s) p_{\tilde{\mathbf{T}}^s, \tilde{\mathbf{\Gamma}}^s}(s|d^e) ds \tag{22}$$

$$\implies \int_{\mathcal{Y}} p_{\varepsilon_y}(y-\bar{y}) p_{\mathbf{T}^s, \mathbf{\Gamma}^s}(f_y^{-1}(\bar{y})|d^e) \text{vol} J_{f_y^{-1}}(\bar{y}) d\bar{y} \tag{23}$$

$$= \int_{\mathcal{Y}} p_{\varepsilon_y}(y-\bar{y}) p_{\tilde{\mathbf{T}}^s, \tilde{\mathbf{\Gamma}}^s}(\tilde{f}_y^{-1}(\bar{y})|d^e) \text{vol} J_{\tilde{g}^{-1}}(\bar{y}) d\bar{y} \tag{24}$$

$$\implies \int_{\mathcal{S}} \tilde{p}_{\mathbf{T}^s, \mathbf{\Gamma}^s, f_y}(\bar{y}|d^e) p_{\varepsilon_y}(y-\bar{y}) d\bar{y} = \int_{\mathcal{S}} \tilde{p}_{\tilde{\mathbf{T}}^s, \tilde{\mathbf{\Gamma}}^s, \tilde{f}_y}(\bar{y}|d^e) p_{\varepsilon_y}(y-\bar{y}) d\bar{y} \tag{25}$$

$$\implies (\tilde{p}_{\mathbf{T}^s, \mathbf{\Gamma}^s, f_y} * p_{\varepsilon_y})(y|d^e) = (\tilde{p}_{\tilde{\mathbf{T}}^s, \tilde{\mathbf{\Gamma}}^s, \tilde{f}_y} * p_{\varepsilon_y})(y|d^e) \tag{26}$$

$$\implies F[\tilde{p}_{\mathbf{T}^s, \mathbf{\Gamma}^s, f_y}](\omega) \varphi_{\varepsilon_y}(\omega) = F[\tilde{p}_{\tilde{\mathbf{T}}^s, \tilde{\mathbf{\Gamma}}^s, \tilde{f}_y}](\omega) \varphi_{\varepsilon_y}(\omega) \tag{27}$$

$$\implies F[\tilde{p}_{\mathbf{T}^s, \mathbf{\Gamma}^s, f_y}](\omega) = F[\tilde{p}_{\tilde{\mathbf{T}}^s, \tilde{\mathbf{\Gamma}}^s, \tilde{f}_y}](\omega) \tag{28}$$

$$\implies \tilde{p}_{\mathbf{T}^s, \mathbf{\Gamma}^s, f_y}(y) = \tilde{p}_{\tilde{\mathbf{T}}^s, \tilde{\mathbf{\Gamma}}^s, \tilde{f}_y}(y), \tag{29}$$

and that

$$p_{f_x, f_y \mathbf{T}^s, \mathbf{\Gamma}^s, \mathbf{T}^z, \mathbf{\Gamma}^z}(x,y|d^e) = p_{\tilde{f}_x, \tilde{f}_y, \tilde{\mathbf{T}}^s, \tilde{\mathbf{\Gamma}}^s, \tilde{\mathbf{T}}^z, \tilde{\mathbf{\Gamma}}^z}(x,y|d^e) \tag{30}$$

$$\implies \int_{\mathcal{S}\times\mathcal{Z}} p_{f_x}(x|s,z) p_{f_y}(y|s) p_{\mathbf{T}^s, \mathbf{\Gamma}^s, \mathbf{T}^z, \mathbf{\Gamma}^z}(s,z|d^e) ds dz$$
$$= \int_{\mathcal{S}\times\mathcal{Z}} p_{\tilde{f}}(x|s,z) p_{\tilde{f}_y}(y|s) p_{\tilde{\mathbf{T}}^s, \tilde{\mathbf{\Gamma}}^s, \tilde{\mathbf{T}}^z, \tilde{\mathbf{\Gamma}}^z}(s,z|d^e) ds dz \tag{31}$$

$$\implies \int_{\mathcal{V}} p_{\varepsilon}(v-\bar{v}) p_{\mathbf{T}^s, \mathbf{\Gamma}^s, \mathbf{T}^z, \mathbf{\Gamma}^z}(h^{-1}(\bar{v})|d^e) \text{vol} J_{h^{-1}}(\bar{v}) d\bar{v} \tag{32}$$

$$= \int_{\mathcal{V}} p_{\varepsilon}(v-\bar{v}) p_{\tilde{\mathbf{T}}^s, \tilde{\mathbf{\Gamma}}^s, \tilde{\mathbf{T}}^z, \tilde{\mathbf{\Gamma}}^z}(\tilde{h}^{-1}(\bar{v})|d^e) \text{vol} J_{\tilde{h}^{-1}}(\bar{v}) d\bar{v} \tag{33}$$

$$\implies \int_{\mathcal{S}\times\mathcal{Z}} \tilde{p}_{\mathbf{T}^s, \mathbf{\Gamma}^s, \mathbf{T}^z, \mathbf{\Gamma}^z, h, c}(\bar{v}|d) p_{\varepsilon}(v-\bar{v}) d\bar{v} = \int_{\mathcal{S}\times\mathcal{Z}} \tilde{p}_{\tilde{\mathbf{T}}^s, \tilde{\mathbf{\Gamma}}^s, \tilde{\mathbf{T}}^z, \tilde{\mathbf{\Gamma}}^z, \tilde{h}, d^e}(\bar{v}|d^e) p_{\varepsilon}(v-\bar{v}) d\bar{v} \tag{34}$$

$$\implies (\tilde{p}_{\mathbf{T}^s, \mathbf{\Gamma}^s, \mathbf{T}^z, \mathbf{\Gamma}^z, h} * p_{\varepsilon})(v) = (\tilde{p}_{\tilde{\mathbf{T}}^s, \tilde{\mathbf{\Gamma}}^s, \tilde{\mathbf{T}}^z, \tilde{\mathbf{\Gamma}}^z, \tilde{h}} * p_{\varepsilon})(v) \tag{35}$$

$$\implies F[\tilde{p}_{\mathbf{T}^s,\mathbf{\Gamma}^s,\mathbf{T}^z,\mathbf{\Gamma}^z,h}](\omega)\varphi_\varepsilon(\omega) = F[\tilde{p}_{\tilde{\mathbf{T}}^s,\tilde{\mathbf{\Gamma}}^s,\tilde{\mathbf{T}}^z,\tilde{\mathbf{\Gamma}}^z,\tilde{h}}](\omega)\varphi_\varepsilon(\omega) \tag{36}$$

$$\implies F[\tilde{p}_{\mathbf{T}^s,\mathbf{\Gamma}^s,\mathbf{T}^z,\mathbf{\Gamma}^z,h}](\omega) = F[\tilde{p}_{\tilde{\mathbf{T}}^s,\tilde{\mathbf{\Gamma}}^s,\tilde{\mathbf{T}}^z,\tilde{\mathbf{\Gamma}}^z,\tilde{h}}](\omega) \tag{37}$$

$$\implies \tilde{p}_{\mathbf{T}^s,\mathbf{\Gamma}^s,\mathbf{T}^z,\mathbf{\Gamma}^z,h}(v) = \tilde{p}_{\mathbf{T}^s,\mathbf{\Gamma}^s,\mathbf{T}^z,\mathbf{\Gamma}^z,h}(v), \tag{38}$$

where $v := [x^\top, y^\top]^\top$, $\varepsilon := [\varepsilon_x^\top, \varepsilon_y^\top]^\top$, $h(v) = [[f_x]_{\mathcal{Z}}^{-1}(x)^\top, f_y^{-1}(y)^\top]^\top$. According to Eq. (29), we have

$$\log \mathrm{vol} J_{f_y}(y) + \sum_{i=1}^{q_s}\left(\log B_i(f_{y,i}^{-1}(y)) - \log A_i(d^e) + \sum_{j=1}^{k_s} T_{i,j}^s(f_{y,i}^{-1}(y))\Gamma_{i,j}^s(d^e)\right)$$
$$= \log \mathrm{vol} J_{\tilde{f}_y}(y) + \sum_{i=1}^{q_s}\left(\log \tilde{B}_i(\tilde{f}_{y,i}^{-1}(y)) - \log \tilde{A}_i(d^e) + \sum_{j=1}^{k_s} \tilde{T}_{i,j}^s(\tilde{f}_{y,i}^{-1}(y))\tilde{\Gamma}_{i,j}^s(d^e)\right) \tag{39}$$

Suppose that the assumption (4) holds, then we have

$$\langle\mathbf{T}^s(f_y^{-1}(y)),\overline{\mathbf{\Gamma}}^s(d^{e_k})\rangle + \sum_i \log \frac{A_i(d^{e_1})}{A_i(d^{e_k})} = \langle\tilde{\mathbf{T}}^s(\tilde{f}_y^{-1}(y)),\overline{\tilde{\mathbf{\Gamma}}}^s(d^{e_k})\rangle + \sum_i \log \frac{\tilde{A}_i(d^{e_1})}{\tilde{A}_i(d^{e_k})} \tag{40}$$

for all $k \in [m]$, where $\overline{\mathbf{\Gamma}}(d) = \mathbf{\Gamma}(d) - \mathbf{\Gamma}(d^{e_1})$. Denote $\tilde{b}_s(k) = \sum_i \frac{\tilde{A}_i(d^{e_1})A_i(d^{e_k})}{\tilde{A}_i(d^{e_k})A_i(d^{e_1})}$ for $k \in [m]$, then we have

$$\overline{\mathbf{\Gamma}}^{s,\top}\mathbf{T}^s(f_y^{-1}(y)) = \overline{\tilde{\mathbf{\Gamma}}}^{s,\top}\tilde{\mathbf{T}}^s(\tilde{f}_y^{-1}(y)) + \tilde{b}_s, \tag{41}$$

Similarly, from Eq. (20) and Eq. (38), there exists $\tilde{b}_z, \tilde{b}_s$ such that

$$\overline{\mathbf{\Gamma}}^{s,\top}\mathbf{T}^s([f_x]_{\mathcal{S}}^{-1}(x)) + \overline{\mathbf{\Gamma}}^{z,\top}\mathbf{T}^z([f_x]_{\mathcal{Z}}^{-1}(x)) = \overline{\tilde{\mathbf{\Gamma}}}^{s,\top}\tilde{\mathbf{T}}^s([\tilde{f}_x]_{\mathcal{S}}^{-1}(x)) + \overline{\tilde{\mathbf{\Gamma}}}^{z,\top}\tilde{\mathbf{T}}^z([\tilde{f}_x]_{\mathcal{Z}}^{-1}(x)) + \tilde{b}_z + \tilde{b}_s, \tag{42}$$

where $\tilde{b}_z(k) = \sum_i \frac{\tilde{Z}_i(d^{e_1})Z_i(d^{e_k})}{\tilde{Z}_i(d^{e_k})Z_i(d^{e_1})}$ for $k \in [m]$; and that,

$$\overline{\mathbf{\Gamma}}^{s,\top}\mathbf{T}^s(f_y^{-1}(y)) + \overline{\mathbf{\Gamma}}^{z,\top}\mathbf{T}^z([f_x^{-1}]_{\mathcal{Z}}(x)) = \overline{\tilde{\mathbf{\Gamma}}}^{s,\top}\tilde{\mathbf{T}}^s(\tilde{f}_y^{-1}(y)) + \overline{\tilde{\mathbf{\Gamma}}}^{z,\top}\tilde{\mathbf{T}}^z([\tilde{f}_x^{-1}]_{\mathcal{Z}}(x)) + \tilde{b}_z + \tilde{b}_s. \tag{43}$$

Substituting Eq. (41) to Eq. (42) and Eq. (43), we have that

$$\overline{\mathbf{\Gamma}}^{z,\top}\mathbf{T}^z([f_x^{-1}]_{\mathcal{Z}}(y)) = \overline{\tilde{\mathbf{\Gamma}}}^{z,\top}\tilde{\mathbf{T}}^z([\tilde{f}_x^{-1}]_{\mathcal{Z}}(y)) + \tilde{b}_z, \quad \overline{\mathbf{\Gamma}}^{s,\top}\mathbf{T}^s([f_x^{-1}]_{\mathcal{S}}(y)) = \overline{\tilde{\mathbf{\Gamma}}}^{s,\top}\tilde{\mathbf{T}}^s([\tilde{f}_x^{-1}]_{\mathcal{S}}(y)) + \tilde{b}_s. \tag{44}$$

According to assumption (4), the $\overline{\mathbf{\Gamma}}^{s,\top}$ and $\overline{\mathbf{\Gamma}}^{z,\top}$ have full column rank. Therefore, we have that

$$\mathbf{T}^z([f_x^{-1}]_{\mathcal{Z}}(x)) = \left(\overline{\mathbf{\Gamma}}^z\overline{\mathbf{\Gamma}}^{z,\top}\right)^{-1}\overline{\tilde{\mathbf{\Gamma}}}^{z,\top}\tilde{\mathbf{T}}^z([\tilde{f}_x^{-1}]_{\mathcal{Z}}(x)) + \left(\overline{\mathbf{\Gamma}}^z\overline{\mathbf{\Gamma}}^{z,\top}\right)^{-1}\tilde{b}_z \tag{45}$$

$$\mathbf{T}^s([f_x^{-1}]_{\mathcal{S}}(x)) = \left(\overline{\mathbf{\Gamma}}^s\overline{\mathbf{\Gamma}}^{s,\top}\right)^{-1}\overline{\tilde{\mathbf{\Gamma}}}^{s,\top}\tilde{\mathbf{T}}^s([\tilde{f}_x^{-1}]_{\mathcal{S}}(x)) + \left(\overline{\mathbf{\Gamma}}^s\overline{\mathbf{\Gamma}}^{s,\top}\right)^{-1}\tilde{b}_s. \tag{46}$$

$$\mathbf{T}^s(f_y^{-1}(y)) = \left(\overline{\mathbf{\Gamma}}^s\overline{\mathbf{\Gamma}}^{s,\top}\right)^{-1}\overline{\tilde{\mathbf{\Gamma}}}^{s,\top}\tilde{\mathbf{T}}^s(\tilde{f}_y^{-1}(y)) + \left(\overline{\mathbf{\Gamma}}^s\overline{\mathbf{\Gamma}}^{s,\top}\right)^{-1}\tilde{b}_s. \tag{47}$$

Denote $M_z := \left(\overline{\mathbf{\Gamma}}^z\overline{\mathbf{\Gamma}}^{z,\top}\right)^{-1}\overline{\tilde{\mathbf{\Gamma}}}^{z,\top}$, $M_s := \left(\overline{\mathbf{\Gamma}}^s\overline{\mathbf{\Gamma}}^{s,\top}\right)^{-1}\overline{\tilde{\mathbf{\Gamma}}}^{s,\top}$ and $a_s = \left(\overline{\mathbf{\Gamma}}^s\overline{\mathbf{\Gamma}}^{s,\top}\right)^{-1}\tilde{b}_s$, $a_z = \left(\overline{\mathbf{\Gamma}}^z\overline{\mathbf{\Gamma}}^{z,\top}\right)^{-1}\tilde{b}_z$. The left is to prove that $M_z$ and $M_s$ are invertible matrices. Denote $\bar{x} = f^{-1}(x)$. Applying the (Khemakhem, Kingma and Hyvärinen, 2020, Lemma 3) we have that there exists $k_s$ points $\bar{x}^1, ..., \bar{x}^{k_s}, \tilde{\bar{x}}^1, ..., \tilde{\bar{x}}^{k_z}$ such that $\left((\mathbf{T}^s)_i'([f_x^{-1}]_{\mathcal{S}_i}(x_i^1)), ..., (\mathbf{T}^s)_i'([f_x^{-1}]_{\mathcal{S}_i}(x_i^{k_s}))\right)$ for each $i \in [q_s]$ and $\left((\mathbf{T}^z)_i'([f_x^{-1}]_{\mathcal{Z}_i}(\tilde{x}_i^1)), ..., (\mathbf{T}^z)_i'([f_x^{-1}]_{\mathcal{S}_i}(\tilde{x}_i^{k_z}))\right)$ for each $i \in [q_t]$ are linearly independent.

By differentiating Eq. (45) and Eq. (46) for each $\bar{x}^i$ with $i \in [q_s]$ and $\tilde{\bar{x}}^i$ with $i \in [q_z]$ respectively, we have that

$$\left(J_{\mathbf{T}^s}(\bar{x}^1), ..., J_{\mathbf{T}^s}(\bar{x}^{k_s})\right) = M_s \left(J_{\mathbf{T}^s \circ \tilde{f}_x^{-1} \circ f_x}(\bar{x}^1), ..., J_{\mathbf{T}^s \circ \tilde{f}_x^{-1} \circ f}(\bar{x}^{k_s})\right) \tag{48}$$

$$\left(J_{\mathbf{T}^z}(\tilde{\bar{x}}^1), ..., J_{\mathbf{T}^z}(\tilde{\bar{x}}^{k_z})\right) = M_z \left(J_{\mathbf{T}^z \circ \tilde{f}_x^{-1} \circ f_x}(\tilde{\bar{x}}^1), ..., J_{\mathbf{T}^z \circ \tilde{f}_x^{-1} \circ f_x}(\tilde{\bar{x}}^{k_z})\right). \tag{49}$$

The linearly independence of $\left((\mathbf{T}^s)'_i([f_x^{-1}]_{\mathcal{S}_i}(x_i^1)), ..., (\mathbf{T}^s)'_i([f_x^{-1}]_{\mathcal{S}_i}(x_i^{k_s}))\right)$ and $\left((\mathbf{T}^z)'_i([f_x^{-1}]_{\mathcal{Z}_i}(\tilde{x}_i^1)), ..., (\mathbf{T}^z)'_i([f_x^{-1}]_{\mathcal{S}_i}(\tilde{x}_i^{k_z}))\right)$ imply that the $\left(J_{\mathbf{T}^s}(\bar{x}^1), ..., J_{\mathbf{T}^s}(\bar{x}^{k_s})\right)$ and $\left(J_{\mathbf{T}^z}(\tilde{\bar{x}}^1), ..., J_{\mathbf{T}^z}(\tilde{\bar{x}}^{k_z})\right)$ are invertible, which implies the invertibility of matrix $M_s$ and $M_z$. The rest is to prove $p_{f_y}(y|[f_x]_{\mathcal{S}}^{-1}(x)) = p_{\tilde{f}_y}(y|[\tilde{f}_x]_{\mathcal{S}}^{-1}(x))$. This can be shown by applying Eq. (31) again. Specifically, according to Eq. (31), we have that

$$\int_{\mathcal{X}} p_{\varepsilon_x}(x - \bar{x}) p(y|[f_x]_{\mathcal{S}}^{-1}(\bar{x})) p_{\mathbf{T}^s, \mathbf{\Gamma}^s, \mathbf{T}^z, \mathbf{\Gamma}^z}(f^{-1}(\bar{x})|d^e) \text{vol} J_{f^{-1}}(\bar{x}) d\bar{x}$$
$$= \int_{\mathcal{X}} p_{\varepsilon_x}(x - \bar{x}) p(y|[\tilde{f}_x]_{\mathcal{S}}^{-1}(\bar{x})) p_{\mathbf{T}^s, \mathbf{\Gamma}^s, \mathbf{T}^z, \mathbf{\Gamma}^z}(\tilde{f}^{-1}(\bar{x})|d^e) \text{vol} J_{\tilde{f}^{-1}}(\bar{x}) d\bar{x}. \tag{50}$$

Denote $l_{\mathbf{T}^s, \mathbf{\Gamma}^s, \mathbf{T}^z, \mathbf{\Gamma}^z, f_y, f_x, y}(x) := p_{f_y}(y|[f_x]_{\mathcal{S}}^{-1}(\bar{x})) p_{\mathbf{T}^s, \mathbf{\Gamma}^s, \mathbf{T}^z, \mathbf{\Gamma}^z}(f^{-1}(\bar{x})|d^e) \text{vol} J_{f_x^{-1}}(\bar{x})$, we have

$$\int_{\mathcal{X}} p_{\varepsilon_x}(x - \bar{x}) l_{\mathbf{T}^s, \mathbf{\Gamma}^s, \mathbf{T}^z, \mathbf{\Gamma}^z, f_y, f_x, y}(\bar{x}) d\bar{x} = \int_{\mathcal{X}} p_{\varepsilon_x}(x - \bar{x}) l_{\tilde{\mathbf{T}}^s, \tilde{\mathbf{\Gamma}}^s, \tilde{\mathbf{T}}^z, \tilde{\mathbf{\Gamma}}^z, \tilde{f}_y, \tilde{f}_x, y}(\bar{x}) d\bar{x} \tag{51}$$

$$\Longrightarrow (l_{\mathbf{T}^s, \mathbf{\Gamma}^s, \mathbf{T}^z, \mathbf{\Gamma}^z, f_y, f_x, y} * p_{\varepsilon_x})(x|d^e) = (l_{\tilde{\mathbf{T}}^s, \tilde{\mathbf{\Gamma}}^s, \tilde{\mathbf{T}}^z, \tilde{\mathbf{\Gamma}}^z, \tilde{f}_y, \tilde{f}_x, y} * p_{\varepsilon_x})(x|d^e) \tag{52}$$

$$\Longrightarrow F[l_{\tilde{\mathbf{T}}^s, \tilde{\mathbf{\Gamma}}^s, \tilde{\mathbf{T}}^z, \tilde{\mathbf{\Gamma}}^z, \tilde{f}_y, \tilde{f}_x, y}](\omega) \varphi_{\varepsilon_x}(\omega) = F[l_{\mathbf{T}^s, \mathbf{\Gamma}^s, \mathbf{T}^z, \mathbf{\Gamma}^z, f_y, f_x, y}](\omega) \varphi_{\varepsilon_x}(\omega) \tag{53}$$

$$\Longrightarrow F[l_{\mathbf{T}^s, \mathbf{\Gamma}^s, \mathbf{T}^z, \mathbf{\Gamma}^z, f_y, f_x, y}](\omega) = F[l_{\tilde{\mathbf{T}}^s, \tilde{\mathbf{\Gamma}}^s, \tilde{\mathbf{T}}^z, \tilde{\mathbf{\Gamma}}^z, \tilde{f}_y, \tilde{f}_x, y}](\omega) \tag{54}$$

$$\Longrightarrow l_{\mathbf{T}^s, \mathbf{\Gamma}^s, \mathbf{T}^z, \mathbf{\Gamma}^z, f_y, f_x, y}(x) = l_{\tilde{\mathbf{T}}^s, \tilde{\mathbf{\Gamma}}^s, \tilde{\mathbf{T}}^z, \tilde{\mathbf{\Gamma}}^z, \tilde{f}_y, \tilde{f}_x, y}(x) \tag{55}$$

$$\Longrightarrow p_{f_y}(y|[f_x]_{\mathcal{S}}^{-1}(x)) p_{\mathbf{T}^s, \mathbf{\Gamma}^s, \mathbf{T}^z, \mathbf{\Gamma}^z}(f^{-1}(x)|d^e) \text{vol} J_{f_x^{-1}}(x)$$
$$= p_{\tilde{f}_y}(y|[\tilde{f}_x]_{\mathcal{S}}^{-1}(x)) p_{\tilde{\mathbf{T}}^s, \tilde{\mathbf{\Gamma}}^s, \tilde{\mathbf{T}}^z, \tilde{\mathbf{\Gamma}}^z}(\tilde{f}^{-1}(x)|d^e) \text{vol} J_{\tilde{f}_x^{-1}}(x). \tag{56}$$

Taking the $\log$ transformation on both sides of Eq. (56), we have that

$$\log p_{f_y}(y|[f_x]_{\mathcal{S}}^{-1}(x)) + \log p_{\mathbf{T}^s, \mathbf{\Gamma}^s, \mathbf{T}^z, \mathbf{\Gamma}^z}(f^{-1}(x)|d^e) + \log \text{vol} J_{f_x^{-1}}(x)$$
$$= \log p_{\tilde{f}_y}(y|[\tilde{f}_x]_{\mathcal{S}}^{-1}(x)) + \log p_{\tilde{\mathbf{T}}^s, \tilde{\mathbf{\Gamma}}^s, \tilde{\mathbf{T}}^z, \tilde{\mathbf{\Gamma}}^z}(\tilde{f}^{-1}(x)|d^e) + \log \text{vol} J_{\tilde{f}_x^{-1}}(x). \tag{57}$$

Subtracting Eq. (57) with $y_2$ from Eq. (57) with $y_1$, we have

$$\frac{p_{f_y}(y_2|[f_x]_{\mathcal{S}}^{-1}(x))}{p_{f_y}(y_1|[f_x]_{\mathcal{S}}^{-1}(x))} = \frac{p_{\tilde{f}_y}(y_2|[\tilde{f}_x]_{\mathcal{S}}^{-1}(x))}{p_{\tilde{f}_y}(y_1|[\tilde{f}_x]_{\mathcal{S}}^{-1}(x))} \tag{58}$$

$$\Longrightarrow \int_{\mathcal{Y}} \frac{p_{f_y}(y_2|[f_x]_{\mathcal{S}}^{-1}(x))}{p_{f_y}(y_1|[f_x]_{\mathcal{S}}^{-1}(x))} dy_2 = \int_{\mathcal{Y}} \frac{p_{\tilde{f}_y}(y_2|[\tilde{f}_x]_{\mathcal{S}}^{-1}(x))}{p_{\tilde{f}_y}(y_1|[\tilde{f}_x]_{\mathcal{S}}^{-1}(x))} dy_2 \tag{59}$$

$$\Longrightarrow p_{f_y}(y_1|[f_x]_{\mathcal{S}}^{-1}(x)) = p_{\tilde{f}_y}(y_1|[\tilde{f}_x]_{\mathcal{S}}^{-1}(x)), \tag{60}$$

for any $y_1 \in \mathcal{Y}$. This completes the proof. $\square$

**Understanding the assumption (4) in Theorem 7.9 and 7.6.** Recall that we assume the confounder $d_s$ in LaCIM is the source variable for generating data in corresponding domain. Here we also use the $\mathcal{C}$ to denote the space of $d_s$ (since $d_s := c$), then we have the following theoretical conclusion that the as long as the image set of $\mathcal{C}$ is not included in any sets with Lebesgue measure 0, the assumption (4) holds. This conclusion means that the assumption **(4)** holds generically.

**Theorem 7.10.** *Denote* $h^{t=s,z}(d) := \left(\Gamma_{1,1}^t(d) - \Gamma_{1,1}^t(d^{e_1}), ..., \Gamma_{q_t, k_t}^t(d) - \Gamma_{1,1}^t(d^{e_1})\right)^\top$, $h(\mathcal{C}) := h^s(\mathcal{S}) \oplus h^z(\mathcal{Z}) \subset \mathbb{R}^{q_z * k_z} \oplus \mathbb{R}^{q_s * k_s}$, *then assumption (4) holds if* $h(\mathcal{C})$ *is not included in any zero-measure set of* $\mathbb{R}^{q_z * k_z} \oplus \mathbb{R}^{q_s * k_s}$. *Denote* $r_s := q_s * k_s$ *and* $r_z := q_z * k_z$.

*Proof.* With loss of generality, we assume that $r_s \leq r_z$. Denote $Q$ as the set of integers $q$ such that there exists $d^{e_2}, ..., d^{q+1}$ that the $\text{rank}([h^z(d^{e_2}), ..., h^z(d^{e_{q+1}})]) = \min(q, r_z)$ and $\text{rank}([h^s(d^{e_2}), ..., h^s(d^{e_{q+1}})]) = \min(q, r_s)$. Denote $u := \max(Q)$. We discuss two possible cases for $u$, respectively:

- Case 1. $u < r_s \leq r_z$. Then there exists $d^{e_2}, ..., d^{e_{u+1}}$ s.t. $h^z(d^{e_2}), ..., h^z(d^{e_{u+1}})$ and $h^s(d^{e_2}), ..., h^s(d^{e_{u+1}})$ are linearly independent. Then $\forall c$, we have $h^z(d) \in L(h^z(d^{e_2}), ..., h^z(d^{e_{u+1}}))$ or $h^s(d) \in L(h^s(d^{e_2}), ..., h^s(d^{e_{u+1}}))$. Therefore, so we have $h^z(d) \oplus h^s(d) \in [L(h^z(d^{e_2}), ..., h^z(d^{e_{u+1}})) \oplus \mathbb{R}^{r_s}] \cup [\mathbb{R}^{r_z} \oplus L(h^s(d^{e_2}), ..., h^s(d^{e_{u+1}}))]$, which has measure 0 in $\mathbb{R}^{r_z} \oplus \mathbb{R}^{r_s}$.

- Case 2. $r_s \leq u < r_z$. Then there exists $d^{e_2}, ..., d^{e_{u+1}}$ s.t. $h^z(d^{e_2}), ..., h^z(d^{e_{u+1}})$ are linearly independent and $rank([h^s(d^{e_1}), ..., h^s(d^{e_u})]) = r_s$. Then $\forall c$, we have $h^z(d) \in L(h^z(d^{e_1}), ..., h^z(d^{e_{u+1}}))$, which means that $h^z(d) \oplus h^s(d) \in L(h^z(d^{e_1}), ..., h^z(d^{e_{u+1}})) \oplus \mathbb{R}^{r_s}$, which has measure 0 in $\mathbb{R}^{r_z} \oplus \mathbb{R}^{r_s}$.

The above two cases are contradict to the assumption that $h(\mathcal{C})$ is not included in any zero-measure set of $\mathbb{R}^{r_z} \oplus \mathbb{R}^{r_s}$. $\square$

**Lemma 7.11.** *Consider the cases when $k_s \geq 2$. Then suppose the assumptions in theorem 7.9 are satisfied. Further assumed that*

- *The sufficient statistics $\mathbf{T}^s_{i,j}$ are twice differentiable for each $i \in [q_s]$ and $j \in [k_s]$.*

- *$f_y$ is twice differentiable.*

*Then we have $M_s$ in theorem 7.9 is block permutation matrix.*

*Proof.* Directly applying (Khemakhem, Kingma and Hyvärinen, 2020, Theorem 2) with $f_x, A, b, \mathbf{T}, x$ replaced by $f_y, M_s, a_s, \mathbf{T}^s, y$. $\square$

**Lemma 7.12.** *Consider the cases when $k_s = 1$. Then suppose the assumptions in theorem 7.9 are satisfied. Further assumed that*

- *The sufficient statistics $\mathbf{T}^s_i$ are not monotonic for $i \in [q_s]$.*

- *$g$ is smooth.*

*Then we have $M_s$ in theorem 7.9 is block permutation matrix.*

*Proof.* Directly applying (Khemakhem, Kingma and Hyvärinen, 2020, Theorem 3) with $f_x, A, b, \mathbf{T}, x$ replaced by $f_y, M_s, a_s, \mathbf{T}^s, y$. $\square$

*Proof of Theorem 7.6.* According to theorem 7.9, there exist invertible matrices $M_s$ and $M_z$ such that

$$\mathbf{T}(f_x^{-1}(x)) = A\tilde{\mathbf{T}}(\tilde{f}_x^{-1}(x)) + b$$
$$\mathbf{T}^s([f_x^{-1}]_{\mathcal{S}}(x)) = M_s\tilde{\mathbf{T}}^s([\tilde{f}_x^{-1}]_{\mathcal{S}}(x)) + a_s.$$
$$\mathbf{T}^s(f_y^{-1}(y)) = M_s\tilde{\mathbf{T}}^s(\tilde{f}_y^{-1}(y)) + a_s,$$

where $\mathbf{T} = [\mathbf{T}^{s,\top}, \mathbf{T}^{z,\top}]^\top$, and

$$A = \begin{pmatrix} M_s & 0 \\ 0 & M_z \end{pmatrix}. \tag{61}$$

By further assuming that the sufficient statistics $\mathbf{T}^s_{i,j}$ are twice differentiable for each $i \in [q_s]$ and $j \in [k_s]$ for $k_s \geq 2$ and not monotonic for $k_s = 1$. Then we have that $M_s$ is block permutation matrix. By further assuming that $\mathbf{T}^z_{i,j}$ are twice differentiable for each $i \in [n_z]$ and $j \in [k_z]$ for $k_z \geq 2$ and not monotonic for $k_z = 1$ and applying the lemma 7.11 and 7.12 respectively, we have that $A$ is block permutation matrix. Therefore, $M_z$ is also a block permutation matrix. $\square$

*Proof of Theorem 4.3.* We consider the general case when $\mathcal{C} := \cup_{r=1}^R \mathcal{C}_r$, in which each $\mathcal{C}_r$ can be simplified as a representative point $c_r$. For environment $d^e$, let $\mathrm{P}_{d^e} = [\mathrm{P}(C=c_1|d^e), \cdots, \mathrm{P}(C= c_R|d^e)]$ be the vector of probability mass of $C$ in the environment $d^e$. And $\mathcal{E}_{train}$ has $m$ environments with indexes $d^{e_1}, \cdots, d^{e_m}$. The latent factors $(S, Z)$ belongs to the exponential family distribution $p(s,z|c) = p_{\mathbf{T}^z, \mathbf{\Gamma}^z(d)}(z) p_{\mathbf{T}^s, \mathbf{\Gamma}^s(d)}(s)$. Suppose that $\theta = \{f_x, f_y, \mathbf{T}^s, \mathbf{T}^z\}$ and $\tilde{\theta} = \{\tilde{f}_x, \tilde{g}_y, \tilde{\mathbf{T}}^s, \tilde{\mathbf{T}}^z\}$ share the same observational distribution for each environment, *i.e.*, $p_\theta(x, y|d^e) = p_{\tilde{\theta}}(x, y|d^e)$, then we have that

$$\sum_{r=1}^R p_\theta(x,y|c_r)\,\mathrm{P}(C=c_r|d^e) = \sum_{r=1}^R p_{\tilde{\theta}}(x,y|c_r)\,\mathrm{P}(C=c_R|d^e). \tag{62}$$

Let $\Delta_{x,y} = [p_\theta(x,y|c_1)-p_{\tilde{\theta}}(x,y|c_1), \cdots, p_\theta(x,y|c_m)-p_{\tilde{\theta}}(x,y|c_m)]^\top$, then Eq. (62) can be written as $A\Delta_{x,y} = 0$. Denote $A := \mathrm{P}_{d^{e_1}}^\top \in \mathbb{R}^{m \times R}$. According the *diversity condition*, we have that $A$ and the $[[\mathbf{\Gamma}^t(c_2)-\mathbf{\Gamma}^t(c_1)]^\top, ..., [\mathbf{\Gamma}^t(c_m)-\mathbf{\Gamma}^t(c_1)]^\top]^\top$ have full column rank, therefore we have that $\Delta_{x,y} = 0$, *i.e.* $p_\theta(x,y|c_r) = p_{\tilde{\theta}}(x,y|c_r)$ for each $r \in [R]$. The left proof is the same with the one in theorem 7.6. $\square$

## 7.4 PROOF OF THEOREM 4.4

*Proof of Theorem 4.4.* Due to Eq. (62), it is suffices to prove the conclusion for every $c_r \in \{c_r\}_{r\in[R]}$. Motivated by Barron and Sheu (1991, Theorem 2) that the distribution $p^e(s,z)$ defined on bounded set can be approximated by a sequence of exponential family with sufficient statistics denoted as polynomial terms, therefore the $\mathbf{T}^{t=s,z}$ are twice differentiable hence satisfies the assumption (2) in theorem 4.3 and assumption (1) in lemma 7.11. Besides, the lemma 4 in Barron and Sheu (1991) informs us that the KL divergence between $p_{\theta_0}(s,z|c_r)$ ($\theta_0 := (f_x, f_y, \mathbf{T}^z, \mathbf{T}^s, \mathbf{\Gamma}_0^z, \mathbf{\Gamma}_0^s)$) and $p_{\theta_1}(s,z|c_r)$ ($\theta_1 := (f_x, f_y, \mathbf{T}^z, \mathbf{T}^s, \mathbf{\Gamma}_1^z, \mathbf{\Gamma}_1^s)$ (the $p_{\theta_0}(s,z|c_r), p_{\theta_1}(s,z|c_r)$ belong to exponential family with polynomial sufficient statistics terms) can be bounded by the $\ell_2$ norm of $[(\mathbf{\Gamma}^s(c_r) - \mathbf{\Gamma}_1^s(c_r))^\top, (\mathbf{\Gamma}_0^z(c_r) - \mathbf{\Gamma}_1^z(c_r))^\top]^\top$. Therefore, $\forall \epsilon > 0$, there exists a open set of $\Gamma(c_r)$ such that the $D_{\mathrm{KL}}(p(s,z|c_r), p_\theta(s,z|c_r)) < \epsilon$. Such an open set is with non-zero Lebesgue measurement therefore can satisfy the assumption (4) in theorem 4.3, according to result in theorem 7.10. The left is to prove that for any $p$ defined by a LaCIM following Def. 4.1, there is a sequence of $\{p_m\}_n \in \mathcal{P}_{\exp}$ such that the $d_{\mathrm{Pok}}(p, p_n) \to 0$ that is equivalent to $p_n \xrightarrow{d} p$. For any $A, B$, we consider to prove that

$$I_n \triangleq \left| p(x \in A, y \in B|c_r) - p_n(x \in A, y_n \in B|c_r) \right| \to 0, \tag{63}$$

where $p_n(x \in A, y_n \in B|c_r) = \int_{\mathcal{S}} \int_{\mathcal{Z}} p(x \in A|s,z)p(y_n \in B|s)p_n(s,z|c_r)dsdz$ with

$$y_n(i) = \frac{\exp((f_{y,i}(\boldsymbol{s}) + \varepsilon_{y,i})/T_n)}{\sum_i \exp((f_{y,i}(\boldsymbol{s}) + \varepsilon_{y,i})/T_n)}, \ i = 1, ..., k, \tag{64}$$

for $y \in \mathbb{R}^k$ denoting the $k$-dimensional one-hot vector for categorical variable and $\varepsilon_{y,1,...,k}$ are Gumbel i.i.d. According to (Maddison et al., 2016, Proposition 1) that the $y_n(i) \xrightarrow{d} y(i)$ with

$$p(y(i) = 1) = \frac{\exp(f_{y,i}(\boldsymbol{s}))}{\sum_i \exp((f_{y,i}(\boldsymbol{s}))}, \ as \ T_n \to 0. \tag{65}$$

As long as $f_y$ is smooth, we have that the $p(y_n|s)$ is continuous. We have that

$$
\begin{aligned}
I_n &= \left| p(x \in A, y \in B|c_r) - \int_{\mathcal{S}\times\mathcal{Z}} p(x \in A|s,z)p(y_n \in B|s)p_n(s,z|c_r)dsdz \right| \\
&\leq \left| p(x \in A, y \in B|c_r) - p(x \in A, y_n \in B|c_r) \right| \\
&\quad + \left| p(x \in A, y_n \in B|c_r) - \int_{\mathcal{S}\times\mathcal{Z}} p(x \in A|s,z)p(y_n \in B|s)p_n(s,z|c_r)dsdz \right| \\
&= \left| \int_{\mathcal{S}\times\mathcal{Z}} p(x \in A|s,z)\left(p(y \in B|s) - p(y_n \in B|s)\right)p(s,z|c_r)dsdz \right| \\
&\quad + \left| \int_{\mathcal{S}\times\mathcal{Z}} p(x \in A|s,z)p(y_n \in B|s)\left(p(s,z|c_r) - p_n(s,z|c_r)\right) \right|
\end{aligned}
$$

$$\leq \underbrace{\left| \int_{M_s \times M_z} p(x \in A|s, z) \left( p(y \in B|s) - p(y_n \in B|s) \right) p(s, z|c_r) ds dz \right|}_{I_{n,1}}$$

$$+ \underbrace{\left| \int_{(M_s \times M_z)^{c_r}} p(x \in A|s, z) \left( p(y \in B|s) - p(y_n \in B|s) \right) p(s, z|c_r) ds dz \right|}_{I_{n,2}}$$

$$+ \underbrace{\left| \int_{M_s \times M_z} p(x \in A|s, z) p(y_n \in B|s) \left( p(s, z|c_r) - p_n(s, z|c_r) \right) \right|}_{I_{n,3}}$$

$$+ \underbrace{\left| \int_{(M_s \times M_z)^{c_r}} p(x \in A|s, z) p(y_n \in B|s) \left( p(s, z|c_r) - p_n(s, z|c_r) \right) \right|}_{I_{n,4}}. \tag{66}$$

For $I_{n,1}$, if $y$ is itself additive model with $y = f_y(s) + \varepsilon_y$, then we just set $y_n \overset{d}{=} y$, then we have that $I_{n,1} = 0$. Therefore, we only consider the case when $y$ denotes the categorical variable with softmax distribution, *i.e.*, Eq. (65). $\forall c_r \in \mathcal{C} := \{c_1, ..., c_R\}$ and $\forall \epsilon > 0$, there exists $M_s^{c_r}$ and $M_z^{c_r}$ such that $p(s, z \in M_s^{c_r} \times M_z^{c_r}|c_r) \leq \epsilon$; Denote $M_s \overset{\Delta}{=} \cup_{k=1}^m M_s^{c_r}$ and $M_z \overset{\Delta}{=} \cup_{k=1}^m M_z^{c_r}$, we have that $p(s, z \in M_s \times M_z|c) \leq 2\epsilon$ for all $c_r \in \mathcal{C}$. Since $\forall s_1 \in M_s, \exists N_{s_1}$ such that $\forall n \geq N_{s_1}$, we have that $\left| p(y \in B|s_1) - p(y \in B|s_1) \right| \leq \epsilon$ from that $y_n \overset{d}{\to} y$. Besides, there exists open set $\mathcal{O}_{s_1}$ such that $\forall s \in \mathcal{O}_{s_1}$ and

$$\left| p(y \in B|s_1) - p(y \in B|s_1) \right| \leq \epsilon, \; \left| p(y_n \in B|s_1) - p(y_n \in B|s_1) \right| \leq \epsilon.$$

Again, according to Heine–Borel theorem, there exists finite $s$, namely $s_1, ..., s_l$ such that $M_s \subset \cup_{i=1}^l \mathcal{O}(s_i)$. Then there exists $N \overset{\Delta}{=} \max\{N_{s_1}, ..., N_{s_l}\}$ such that $\forall n \geq N$, we have that

$$\left| p(y \in B|s) - p(y_n \in B|s) \right| \leq 3\epsilon, \; \forall s \in M_s. \tag{67}$$

Therefore, $I_{n,1} \leq \int_{M_s \times M_z} 3\epsilon p(x \in A|s, z) p(s, z|c) ds dz \leq 3\epsilon$. Hence, $I_{n,1} \to 0$ as $n \to \infty$. Besides, we have that $I_{n,2} \leq \int_{M_s \times M_z} 2\epsilon p(s, z|c_r) ds dz \leq 2\epsilon$. Therefore, we have that $\left| \int_{\mathcal{S} \times \mathcal{Z}} p(x \in A|s, z) \left( p(y \in B|s) - p(y_n \in B|s) \right) p(s, z|c_r) ds dz \right| \to 0$ as $n \to \infty$. For $I_{n,3}$, we have that

$$I_{n,3} = \left| \int_{M_s \times M_z} p(x \in A|s, z) p(y_n \in B|s) \mathbb{1}(s, z \in M_s \times M_z) \left( p(s, z|c_r) - p_n(s, z|c_r) \right) ds dz \right|$$

$$\leq \underbrace{\left| \int_{M_s \times M_z} p(x \in A|s, z) p(y_n \in B|s) p(s, z|c_r) \left( \frac{1}{p(s, z \in M_s \times M_z|c_r)} - 1 \right) ds dz \right|}_{I_{n,3,1}}$$

$$+ \underbrace{\left| \int_{M_s \times M_z} p(x \in A|s, z) p(y_n \in B|s) p(s, z|c_r) \left( \frac{1}{p(s, z \in M_s \times M_z|c_r)} - 1 \right) ds dz \right|}_{I_{n,3,2}}. \tag{68}$$

The $I_{n,3,1} \leq \frac{\epsilon}{1-\epsilon}$. Denote $\tilde{p}(s, z|c_r) := \frac{p(s, z|c_r) \mathbb{1}(s, z \in M_s \times M_z)}{p(s, z \in M_s \times M_z|c_r)}$, according to (Barron and Sheu, 1991, Theorem 2), there exists a sequence of $p_n(s, z|c)$ defined on a compact support $M_s \times M_z$ such that $\forall c_r \in \mathcal{C}$, we have that

$$p_n(s, z|c_r) \overset{d}{\to} p(s, z|c_r).$$

Applying again the Heine–Borel theorem, we have that $\forall \epsilon, \exists N$ such that $\forall n \geq N$, we have

$$\left| \tilde{p}(s, z|c_r) - p_n(s, z|c_r) \right| \leq \epsilon, \tag{69}$$

which implies that $I_{n,3,2} \to 0$ as $n \to \infty$ combining with the fact that $p(x, y|s, z)$ is continuous with respect to $s, z$. For $I_{n,4}$, we have that

$$I_{n,4} = \left| \int_{M_s \times M_z} p(x \in A|s, z)p(y_n \in B|s)p(s, z|c_r) \right| \leq \left| \int_{M_s \times M_z} p(s, z|c_r) \right| \leq \epsilon, \quad (70)$$

where the first equality is from that the $p_n(s, z|c_r)$ is defined on $M_s \times M_z$. Then we have that

$$\left| \int_{\mathcal{S} \times \mathcal{Z}} p(x \in A|s, z)p(y_n \in B|s) \left( p(s, z|c_r) - p_n(s, z|c) \right) \right| \to 0, \ as \ n \to \infty. \quad (71)$$

The proof is completed. $\qquad\square$

## 7.5 REPARAMETERIZATION FOR LaCIM-$d$

We provide an alternative training method to avoid parameterization of prior $p(s, z|d^e)$ to increase the diversity of generative models in different environments. Specifically, motivated by Hyvärinen and Pajunen (1999) that any distribution can be transformed to isotropic Gaussian with the density denoted by $p_{\text{Gau}}$, we have that for any $e \in \mathcal{E}_{\text{train}}$, we have

$$p^e(x, y) = \int_{\mathcal{S} \times \mathcal{Z}} p_{f_x}(x|s, z)p_{f_y}(y|s)p(s, z|d^e)dsdz$$

$$= \int_{\mathcal{S} \times \mathcal{Z}} p(x|(\rho_s^e)^{-1}(s'), (\rho_z^e)^{-1}(z'))p(y|\rho_s(s'))p_{\text{Gau}}(s', z')ds'dz',$$

with $s', z' := \rho_s^e(s), \rho_z^e(z) \sim \mathcal{N}(0, I)$. We can then rewrite ELBO for LaCIM-$d$ for environment $e$ as:

$$\mathcal{L}_{\phi,\psi,\rho^e}^e = \mathbb{E}_{p^e(x,y)} \left[ -\log q_\psi^e(y|x) \right]$$

$$+ \mathbb{E}_{p^e(x,y)} \left[ -\mathbb{E}_{q_\psi^e(s,z|x)} \frac{q_\psi(y|(\rho_s^e)^{-1}(s))}{q_\psi^e(y|x)} \log \frac{p_\phi((\rho_s^e)^{-1}(s), (\rho_z^e)^{-1}(z))p_{\text{Gau}}(s, z)}{q_\psi^e(s, z|x)} \right]. \quad (72)$$

## 7.6 IDENTIFIABILITY

Earlier works that identify the latent confounders rely on strong assumptions regarding the causal structure, such as the linear model from latent to observed variable or ICA in which the latent component are independent Silva et al. (2006), or noise-free model Shimizu et al. (2009); Davies (2004). The Hoyer et al. (2008); Janzing, Peters, Mooij and Schölkopf (2012) extend to the additive noise model (ANM) and other causal discovery assumptions. Although the Lee et al. (2019) relaxed the constraints put on the causal structure, it required the latent noise is with small strength, which does not match with many realistic scenarios, such as the structural MRI of Alzheimer's Disease considered in our experiment. The works which also based on the independent component analysis (ICA), *i.e.*, the latent variables are (conditionally) independent, include Davies (2004); Eriksson and Koivunen (2003); recently, a series of works extend the above results to deep nonlinear ICA (Hyvarinen and Morioka, 2016; Hyvärinen et al., 2019; Khemakhem, Kingma and Hyvärinen, 2020; Khemakhem, Monti, Kingma and Hyvärinen, 2020; Teshima et al., 2020). However, these works require that the value of confounder of these latent variables is fixed, which cannot explain the spurious correlation in a single dataset. In contrast, our result can incorporate these scenarios by assuming that each sample has a specific value of the confounder. Other works assume discrete distribution for latent variables, such as Janzing, Sgouritsa, Stegle, Peters and Schölkopf (2012); Kocaoglu et al. (2018); Sgouritsa et al. (2013). However, in the literature, no existing works can disentangle the prediction-causative features from others, in the scenario of avoiding spurious correlation in order for OOD generalization.

## 7.7 COMPARISON WITH EXISTING WORKS

### 7.7.1 $Y \to S$ OR $S \to Y$?

Many existing works Rojas-Carulla et al. (2018); Khemakhem, Monti, Kingma and Hyvärinen (2020); Ilse et al. (2020; 2019) assumed $Y \to S(X)$ as the causal direction. Such an difference from ours

can mainly be contributed to the generating process of $Y$. Different understanding leads to different causal graph. The example of digital hand-writing in Peters et al. (2017) provides a good explanation. Consider the case that the writer is provided with a label first (such as "2") before writing the digit (denoted as $X$), then it should be $Y \rightarrow X$. Consider another case, when the writing is based on the incentive (denoted as $S$) of which digit to write, then the writer record the label $Y$ and the digit $X$ concurrently, in which case it should be $X \leftarrow S \rightarrow Y$. For $Y \rightarrow S$, the $Y$ is thought to be the source variable that generates the latent components and is observed before $X$. In contrast, we define $Y$ as ground-truth labels given by humans. Taking image classification as an example, it is the human that give the classification of all things such as animals. In this case, it can be assumed that the label given by humans are ground-truth labels. This assumption can be based by the work Biederman (1987) in the field of psychology that humans can factorize the image $X$ by many components due to the powerful perception learning ability of human beings. These components which denoted as $S$, can be accurately detected by humans, therefore we can approximately assume that it is the $S$ generating the label $Y$. Consider the task of early prediction in Alzheimer's Disease, the disease label is given based on the pathological analysis and observed after the MRI $X$. Such a labelling outcome can be regarded as the ground-truth which itself is defined by medical science. The corresponding pathology features, as the evidences for labelling, can also thought as the generators of $X$. In these cases, it is more appropriate to assume the $Y$ as the outcome than the cause. For example, the Peters et al. (2016); Kuang et al. (2018) assumed $X_{\mathcal{S}} \rightarrow Y$. As an adaptation to sensory-level data such as image, we assume $S \rightarrow Y$ with $S$ are latent variables to model high-level explanatory factors, which coincides with existing literature Teshima et al. (2020). Another difference lies in the definition of $Y$. The Invariant Risk Minimization (we will give a detailed comparison later) Arjovsky et al. (2019) assumes that $X \rightarrow \tilde{S} \rightarrow Y$ by defining the $Y$ as the label with noise. The $\tilde{S}$ denoted as the extracted hidden components by observer.

### 7.7.2 COMPARISONS WITH DATA AUGMENTATION & ARCHITECTURE DESIGN

The goal of data augmentation Shorten and Khoshgoftaar (2019) is increase the variety of the data distribution, such as geometrical transformation Kang et al. (2017); Taylor and Nitschke (2017), flipping, style transfer Gatys et al. (2015), adversarial robustness Madry et al. (2017). On the other way round, an alternative kind of approaches is to integrate into the model corresponding modules that improve the robustness to some types of variations, such as Worrall et al. (2017); Marcos et al. (2016).

However, these techniques can only make effect because they are included in the training data for neural network to memorize Zhang et al. (2016); besides, the improvement is only limited to some specific types of variation considered. As analyzed in Xie et al. (2020); Krueger et al. (2020), the data augmentation trained with empirical risk minimization or robust optimization Ben-Tal et al. (2009) such as adversarial training Madry et al. (2017); Sagawa et al. (2019) can only achieve robustness on interpolation (convex hull) rather than extrapolation of training environments.

### 7.7.3 COMPARISONS WITH EXISTING WORKS IN DOMAIN ADAPTATION

Apparently, the main difference lies in the problem setting that (i) the domain adaptation (DA) can access the input data of the target domain while ours cannot; and (ii) our methods need multiple training data while the DA only needs one source domain. For methodology, our LaCIM shares insights but different with DA. Specifically, both methods assume some types of invariance that relates the training domains to the target domain. For DA, one stream is to assume the same conditional distribution shared between the source and the target domain, such as covariate shift Huang et al. (2007); Ben-David et al. (2007); Johansson et al. (2019); Sugiyama et al. (2008) in which $P(Y|X)$ are assumed to be the same across domains, concept shift Zhang et al. (2013) in which the $P(X|Y)$ is assumed to be invariant. Such an invariance is related to representation, such as $\Phi(X)$ in Zhao et al. (2019) and $P(Y|\Phi(X))$ in Pan et al. (2010); Ganin et al. (2016); Magliacane et al. (2018).

However, these assumptions are only distribution-level rather than the underlying causation which takes the data-generating process into account. Taking the image classification again as an example, our method first propose a causal graph in which the latent factors are introduced as the explanatory/causal factors of the observed variables. These are supported by the framework of generative model Khemakhem, Kingma and Hyvärinen (2020); Khemakhem, Monti, Kingma and Hyvärinen (2020); Kingma and Welling (2014); Suter et al. (2019) which has natural connection with the causal

graph Schölkopf (2019) that the edge in the causal graph reflects both the causal effect and also the generating process. Until now, perhaps the most similar work to us are Romeijn and Williamson (2018) and Teshima et al. (2020) which also need multiple training domains and get access to a few samples in the target domain. Both work assumes the similar causal graph with us but unlike our LaCIM, they do not separate the latent factors which can not explain the spurious correlation learned by supervised learning Ilse et al. (2020). Besides, the multiple training datasets in Romeijn and Williamson (2018) refer to intervened data which may hard to obtain in some applications. We have verified in our experiments that explicitly disentangle the latent variables into two parts can result in better OOD prediction power than mixing them together.

### 7.7.4 COMPARISONS WITH DOMAIN GENERALIZATION

For domain generalization (DG), similar to the invariance assumption in DA, a series of work proposed to align the representation $\Phi(X)$ that assumed to be invariant across domains Li et al. (2017; 2018); Muandet et al. (2013). As discussed above, these methods lack the deep delving of the underlying causal structure and precludes the variations of unseen domains.

Recently, a series of works leverage causal invariance to enable OOD generalization on unseen domains, such as Ilse et al. (2019) which learns the representation that is domain-invariant. Notably, the Invariant Causal Prediction Peters et al. (2016) formulates the assumption in the definition of Structural Causal Model and assumes that $Y = X_{\mathcal{S}}\beta_{\mathcal{S}}^{\star} + \varepsilon_Y$ where $\varepsilon_Y$ satisfies Gaussian distribution and $\mathcal{S}$ denotes the subset of covariates of $X$. The Rojas-Carulla et al. (2018); Bühlmann (2018) relaxes such an assumption by assuming the invariance of $f_y$ and noise distribution $\varepsilon_y$ in $Y \leftarrow f_y(X_{\mathcal{S}}, \varepsilon_y)$ which induces $P(Y|X_{\mathcal{S}})$. The similar assumption is also adopted in Kuang et al. (2018). However, these works causally related the output to the observed input, which may not hold in many real applications in which the observed data is sensory-level, such as audio waves and pixels. It has been discussed in Bengio et al. (2013); Bengio (2017) that the causal factors should be high-level abstractions/concepts. The Heinze-Deml and Meinshausen (2017) considers the style transfer setting in which each image is linear combination of shape-related variable and contextual-related variable, which respectively correspond to $S$ and $Z$ in our LaCIM in which the nonlinear mechanism (rather than linear combination in Heinze-Deml and Meinshausen (2017)) is allowed. Besides, during testing, our method can generalize to the OOD sample with intervention such as adversarial noise and contextual intervention.

Recently, the most notable work is Invariant Risk Minimization Arjovsky et al. (2019), which will be discussed in detail in the subsequent section.

### 7.7.5 COMPARISONS WITH INVARIANT RISK MINIMIZATION ARJOVSKY ET AL. (2019) AND REFERENCES THERE IN

The Invariant Risk Minimization (IRM) Arjovsky et al. (2019) assumes the existence of invariant representation $\Phi(X)$ that induces the optimal classifier for all domains, *i.e.*, the $\mathbb{E}[Y|Pa(Y)]$ is domain-independent in the formulation of SCM. Similar to our LaCIM, the $Pa(Y)$ can refer to latent variables. Besides, to identify the invariance and the optimal classifier, the training environments also need to be diverse enough. As aforementioned, this assumption is almost necessary to differentiate the invariance mechanism from the variant ones. To learn such an invariance, a regularization function is proposed.

The difference of our LaCIM with IRM lies in two aspects: the direction of causal relation and the methodology. For the direction, as aforementioned in section 7.7.1, the IRM assumes $X \rightarrow S$ rather than the $S, Z \rightarrow X$ in our LaCIM. This is because the IRM defines $Y$ as label with noise while ours definie the $Y$ as the ground-truth label hence should be generated by the ground-truth hidden components that generating $S$. Such an inconsistency can be reflected by experiment regarding to the CMNIST in which the number is the causal factors of the label $Y$, rather than only invariant correlation. Besides, in terms of methodology, the theoretical claim of IRM only holds in linear case; in contrast, the CIMe $f_x, f_y$ are allowed to be nonlinear.

Some other works share the similar spirit with or based on IRM. The Risk-Extrapolation (REx) Krueger et al. (2020) proposed to enforce the similar behavior of $m$ classifiers with variance of which proposed as the regularization function. The work in Xie et al. (2020) proposed a Quasi-distribution framework that can incorporate empirical risk minimization, robust optimization and

REx. It can be concluded that the robust optimization only generalizes the convex hull of training environments (defined as interpolation) and the REx can generalize extrapolated combinations of training environments. This work lacks model of underlying causal structure, although it performs similarly to IRM experimentally. Besides, the Teney et al. (2020) proposed to unpool the training data into several domains with different environment and leverages Arjovsky et al. (2019) to learn invariant information for classifier. Recently, the Bellot and van der Schaar (2020) also assumes the invariance to be generating mechanisms and can generalize the capability of IRM when unobserved confounder exist. However, this work also lacks the analysis of identifiability result.

We finish this section with the following summary of methods in section 7.7.4 and the IRM, in terms of causal factor, invariance type, direction of causal relation, theoretical judgement and the ability to generalize to intervened data.

Table 4: Our LaCIM with related works.

| | Causal Factor | Direction | Invariance Type | Theoretical Judgement | Intervention |
|---|---|---|---|---|---|
| Peters et al. (2016) | Subset of covariates $X$ | $X_{\mathcal{S}} \to Y$ | Linear model with Gaussian noise | Identifiability | Yes |
| Rojas-Carulla et al. (2018) | Subset of covariates $X$ | $X_{\mathcal{S}} \to Y$ | Linear Model | - | - |
| Kuang et al. (2018) | Subset of covariates $X$ | $X_{\mathcal{S}} \to Y$ | Nonlinear | Confounder Balancing | - |
| Bühlmann (2018) | Subset of covariates $X$ | $X_{\mathcal{S}} \to Y$ | Nonlinear | Identifiability | - |
| Arjovsky et al. (2019) | Latent variables $S$ | $X \to S \to Y$ | Linear | Identifiability | - |
| LaCIM (**Ours**) | Latent variables $S, Z$ | $S, Z \to X, S \to Y$ | Nonlinear | Identifiability | Yes |

## 7.8 IMPLEMENTATION DETAILS AND MORE RESULTS FOR SIMULATION

**Data Generation** We set $m = 5, n_e = 1000$ for each $e$. The generating process of $d_s \in \mathbb{R}^{q_{d_s}}$, $Z \in \mathbb{R}^{q_z}$, $S \in \mathbb{R}^{q_s}$, $X \in \mathbb{R}^{q_x}$ and $Y \in \mathbb{R}^{q_y}$ is introduced in the supplement 7.8. We set $q_{d_s} = q_s = q_z = q_y = 2$ and $q_x = 4$. For each environment $e \in [m]$ with $m = 5$, we generate 1000 samples $\mathcal{D}^e = \{x_i, y_i\} \overset{i.i.d}{\sim} \int p_{f_x}(x|s, z) p_{f_y}(y|s) p^e(s, z|d_s^e) ds dz$. The $d_s^e = \left(\mathcal{N}(0, I_{q_{d_s} \times q_{d_s}}) + 5 * e\right) * 2$; the $s, z|d_s^e \sim \mathcal{N}\left(\mu_{\phi_{s,z}^{\star}}(s, z|d_s^e), \sigma^2_{\phi_{s,z}^{\star}}(s, z|d_s^e)\right)$ with $\mu_{\phi_{s,z}^{\star}} = A_{s,z}^{\mu} * d_s^e$ and $\log \sigma_{\phi_{s,z}^{\star}} = A_{s,z}^{\sigma} * d_s^e$ ($A_{s,z}^{\mu}, A_{s,z}^{\sigma}$ are random matrices); the $x|s, z \sim \mathcal{N}\left(\mu_{\phi_x^{\star}}(x|s, z), \sigma^2_{\phi_x^{\star}}(x|s, z)\right)$ with $\mu_{\phi_{s,z}^{\star}} = h(A_x^{\mu,3} * h(A_x^{\mu,2} * h(A_x^{\mu,2} * [s^{\top}, z^{\top}]^{\top}])))$ and $\log \sigma_{\phi_{s,z}^{\star}} = h(A_x^{\sigma,3} * h(A_x^{\sigma,2} * h(A_x^{\sigma,2} * [s^{\top}, z^{\top}]^{\top}])))$ ($h$ is LeakyReLU activation function with slope $= 0.5$ and $A_x^{\mu,i=1,2,3}, A_x^{\sigma,i=1,2,3}$ are random matrices); the $y|s$ is similarly to $x|s, z$ with $A_x^{\mu,i=1,2,3}, A_x^{\sigma,i=1,2,3}$ respectively replaced by $A_y^{\mu,i=1,2,3}, A_y^{\sigma,i=1,2,3}$.

**Implementation Details** We parameterize $p_{\theta}(s, z|d)$, $q_{\phi}(s, z|x, y, d)$, $p_{\theta}(x|s, z)$ and $p_{\theta}(y|s)$ as 3-layer MLP with the LeakyReLU activation function. The Adam with learning rate $5 \times 10^{-4}$ is implemented for optimization. We set the batch size as 512 and run for 2,000 iterations in each trial.

**Visualization.** As shown from the visualization of $S$ is shown in Fig. 7.8, our LaCIM can identify the causal factor $S$.

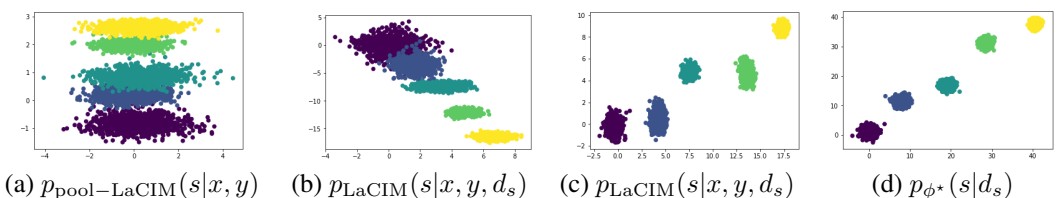

(a) $p_{\text{pool-LaCIM}}(s|x, y)$    (b) $p_{\text{LaCIM}}(s|x, y, d_s)$    (c) $p_{\text{LaCIM}}(s|x, y, d_s)$    (d) $p_{\phi^{\star}}(s|d_s)$

Figure 4: Visualization of $S$. From left to right are: estimated posterior by pool-LaCIM: $p_{\text{pool-LaCIM}}(s|x, y)$, by LaCIM with $c$ as input: $p_{\text{LaCIM}}(s|x, y, d_s)$, by LaCIM with $D$ as input: $p_{\text{LaCIM}}(s|x, y, d)$; the ground-truth $p_{\phi^{\star}}(s|d_s)$.

**The setting when $C$ can take a value in a sample-level.** We consider the generation process of $\mathcal{D}^e$ as $\mathcal{D}^e = \{x_i, y_i\} \overset{i.i.d}{\sim} \int p_{f_x}(x|s, z) p_{f_y}(y|s) p(s, z|c) p(c|d^e) ds dz dc$, with $q_c := 2$. The generation is the same except that the after obtaining $d_s$, we additionally generate $c$ with $c := \mathcal{N}(d_s, I)$. The results are summarized in Tab. 7.8.

Table 5: MCC of identified latent variables for $p^e(x, y) = \int p(x|s, z)p(y|s)p(s, z|c)p(c|d^e)dcdsdz$. Average over 20 times for each data.

| | Data #1 | | Data #2 | | Data #3 | | Data #4 | | Data #5 | | Average | |
|---|---|---|---|---|---|---|---|---|---|---|---|---|
| | $Z$ | $S$ | $Z$ | $S$ | $Z$ | $S$ | $Z$ | $S$ | $Z$ | $S$ | $Z$ | $S$ |
| pool-LaCIM | 0.26 | 0.61 | 0.26 | 0.67 | 0.44 | 0.70 | 0.51 | 0.78 | 0.58 | 0.77 | 0.41 | 0.71 |
| LaCIM-$d_s$ (**Ours**, $m = 3$) | 0.70 | 0.79 | 0.72 | 0.79 | 0.69 | 0.74 | 0.74 | 0.85 | 0.64 | 0.88 | 0.70 | 0.81 |
| LaCIM-$d_s$ (**Ours**, $m = 5$) | 0.73 | 0.85 | 0.70 | 0.89 | **0.85** | 0.91 | 0.81 | 0.84 | **0.83** | **0.93** | 0.78 ↑ | 0.89 ↑ |
| LaCIM-$d_s$ (**Ours**, $m = 7$) | **0.92** | **0.90** | **0.83** | **0.90** | 0.84 | **0.93** | **0.85** | **0.94** | **0.83** | 0.90 | **0.86** ↑ | **0.91** ↑ |

## 7.9 IMPLEMENTATION DETAILS FOR OPTIMIZATION OVER $S, Z$

Recall that we first optimize $s^*, z^*$ according to

$$s^*, z^* = \arg \max_{s,z} \log p_\phi(x|s, z).$$

We first sample some initial points from each posterior distribution $q_\psi^e(s|x)$ and then optimize for 50 iterations. We using Adam as optimizer, with learning rate as 0.002 and weight decay 0.0002. The Fig. 7.9 shows the optimization effect of one run in CMNIST. As shown, the test accuracy keeps growing as iterates. For time saving, we chose to optimize for 50 iterations.

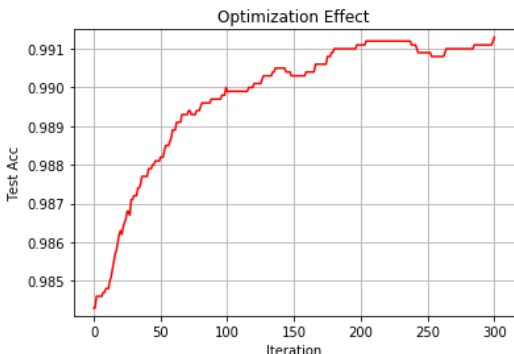

Figure 5: The optimization effect in CMNIST, starting from the point with initial sampling from inference model $q$ of each branch. As shown, the test accuracy increases as iterates.

## 7.10 IMPLEMENTATIONS FOR BASELINE

For the CE $X \to Y$ and the CE $X, d_s \to Y$, they both composed of two parts: (i) feature extractor, followed by (ii) classifier. The network structure of the feature extractor for CE $X \to Y$ is the same with that of our encoder; while the extracted features for CE $X, d \to Y$ is the concatenation of the features encoded from $X \to S, Z$ via the network with the same network structure of our encoder; and the network with the same structure of our prior network for LaCIM-$d$. The network structures of the classifier for both methods are the same to that of our $p_\phi(y|s)$. The IRM and SDA adopt the same structure as CE $X \to Y$. DANN adopt the same structure of CE $X \to Y$ and a additional domain classifier which is the same as that of $p_\phi(y|s)$. sVAE adopt the same structure as LaCIM-$d_s$ with the exception that the $p_\phi(y|s)$ is replaced by $p_\phi(y|z, s)$. MMD-AAE adopt the same structure of encoder, decoder and classifier as LaCIM-$d$ and a additional 2-layer MLP with channel 256-256-$dim_z$ is used to extract latent $z$. The detailed number of parameters and channel size on each dataset for each method are summarized in Tab. 13, 14.

## 7.11 SUPPLEMENTARY FOR COLORED MNIST

**Implementation details** The network structure for inference model is composed of two parts, with the first part shared among all environments and multiple branches corresponding to each environment

for the second part. The network structure of the first-part encoder is composed of four blocks, each block is the sequential of Convolutional Layer (Conv), Batch Normalization (BN), ReLU and max-pooling with stride 2. The output number of feature map is accordingly 32, 64, 128, 256. The second part network structure that output the mean and log-variance of $S, Z$ is Conv-bn-ReLU(256) $\rightarrow$ Adaptive (1) $\rightarrow$ FC(256, 256) $\rightarrow$ ReLU $\rightarrow$ FC(256, $q_{t=s,z}$) with FC stands for fully-connected layer. The structure of $\rho_{t=s,z}$ in Eq. (72) is FC($q_t$, 256) $\rightarrow$ ReLU $\rightarrow$ FC(256, $q_t$). The network structure for generative model $p_\phi(x|s, z)$ is the sequential of three modules: (i) Upsampling with stride 2; (ii) four blocks of Transpose-Convolution (TConv), BN and ReLU with respective output dimension being 128, 64, 32, 16; (iii) Conv-BN-ReLU-Sigmoid with number of channels in the output as 3, followed by cropping step in order to make the image with the same size as input dimension, *i.e.*, $3 \times 28 \times 28$. The network structure for generative model $p_\phi(y|s)$ is commposed of FC (512) $\rightarrow$ BN $\rightarrow$ ReLU $\rightarrow$ FC (256) $\rightarrow$ BN $\rightarrow$ ReLU $\rightarrow$ FC ($|\mathcal{Y}|$). The $q_{t=s,z}$ is set to 32. We implement SGD as optimizer with learning rate 0.5, weight decay $1e-5$ and we set batch size as 256. The total training epoch is 80.

We first explain why we do not flip $y$ with 25% in the manuscript, and then provide further exploration of our method for the setting with flipping $y$.

**Invariant *Causation* v.s. Invariant *Correlation* by Flipping $y$ in Arjovsky et al. (2019)** The $y$ is further flipped with 25% to obtain the final label in IRM setting and this step is omitted in ours. The difference lies in the definition of invariance. Our LaCIM defines invariance as the causal relation between $S$ and the label $Y$, while the one in IRM can be correlation. As illustrated in Handwriting Sample Form in Fig. 7.11 in Grother (1995), the generting direction should be $Y \rightarrow X$. If we denote the variable by flipping $Y$ as $\tilde{Y}$ (*a.k.a*, the final label in IRM), then the causal graph should be $X \leftarrow Y \rightarrow \tilde{Y}$. In this case, the $\tilde{Y}$ is correlated rather than causally related to the digit $X$. For our LaCIM, we define the label as interpretable human label (which can approximate to $y$ for any image $x$) and represented by $Y$ in our experiments. The reason why we do not define the $Y$ as ground-truth label is that (i) the prediction is only based on the extracted components of image which may be determined not only by the ground-truth label; (ii) the learning of ground-truth is interpretable that relevant to human. For example, if a writer is provided with digit "2" but he wrote it mistakenly as "4", then it is more interpretable that we can predict the digit as "4" rather than "2". For the digit with ambiguous label from the perspective of image, even if we predict it mistakenly, it is also interpretable in terms of prediction given the information of only digit. Returning back to the IRM setting, the label is flipping without reference to the semantic shape of digit. Therefore, the flipping may happen to noiseless digits rather than noisy and unsure ones, making the shape of number less semantically related to the label.

**Experiment with IRM setting** We further conduct the experiment on IRM setting, with the final label $y$ defined by flipping original label with 25%, and further color $p^e$ proportions of digits with corresponding color-label mapping. If we assume the original ground-truth label to be the effect of the digit number of $S$, then the anti-causal relation with $Z$ and $Y$ can make the identifiability of $S$ difficult in this flipping scenario. Note that the causal effect between $S$ and $Y$ is invariant across domains, therefore we adopt to regularize the branch of inferring $S$ to be shared among inference models for multiple environments. Besides, we regularize the causal effect between $S$ and $Z$ to be shared among different environments via pairwise regularization. The combined loss is formulated as:

$$\tilde{\mathcal{L}}_{\psi,\phi} = \mathcal{L}_{\psi,\phi} + \frac{\Gamma}{2m^2} \sum_{i=1}^{m} \sum_{j=1}^{m} \|\mathbb{E}_{(x,y)\sim p^{e_i}(x,y)}[y|x] - \mathbb{E}_{(x,y)\sim p^{e_j}(x,y)}[y|x]\|_2^2,$$

with $q_\psi^e(s, z|x)$ in Eq. (72) factorized as $q_{\psi_z^e}(z)q_{\psi_s}(s)$ and $\rho_s$ shared among $m$ environments. The appended loss is coincide with recent study Risk-Extropolation (REx) in Krueger et al. (2020), with the difference of separating $y$-causative factors $S$ from others. We name such a training method as LaCIM-REx. For implementation details, in addition to shared encoder regarding $S$, we set learning rate as 0.1, weight decay as 0.0002, batch size as 256. we have that $p(y|x) = \int_{\mathcal{S}} q_{\psi_s}(s|x)p_\phi(y|\rho_s(s))$ for any $x$. We consider two settings: setting#1 with $m2$ and $p^{e_1} = 0.9, p^{e_2} = 0.8$; and setting#2 with $m = 4$ with $p^{e_1} = 0.9, p^{e_2} = 0.8, p^{e_3} = 0.7, p^{e_4} = 0.6$. We only report the number of IRM since the cross entropy performs poorly in both settings. As shown, our model performs comparably with LaCIM-$d_s$ and better than IRM Arjovsky et al. (2019) due to separation of $S$ znd $Z$.

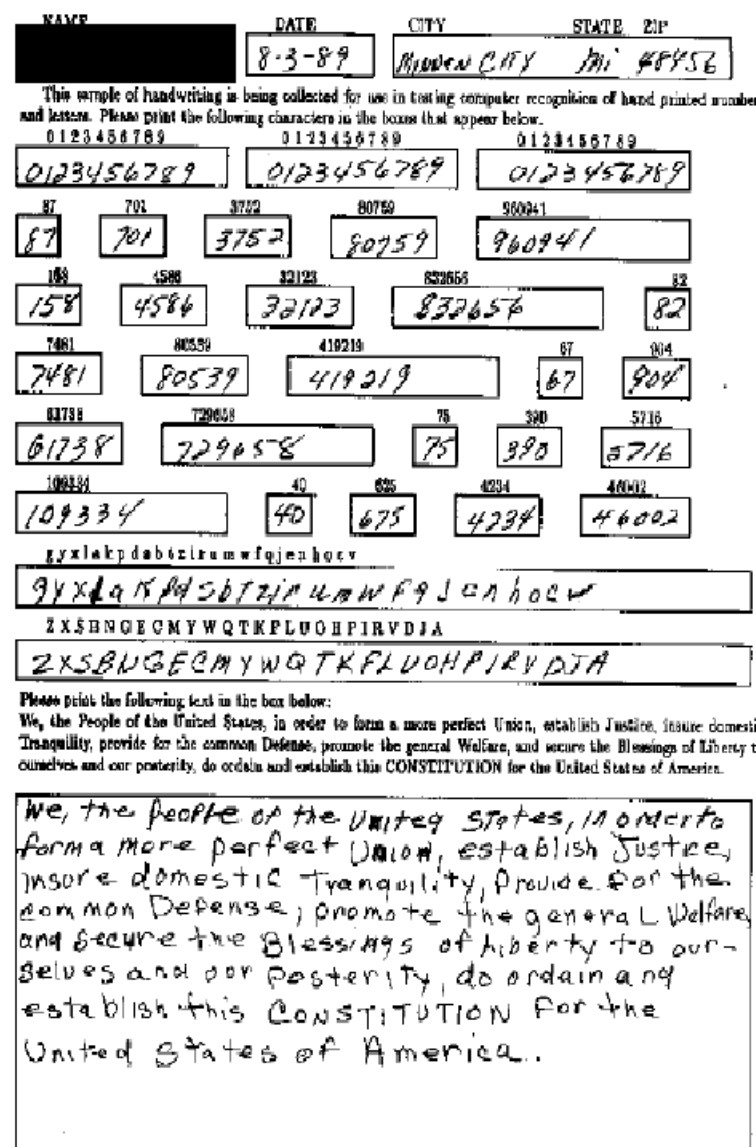

Figure 6: Hand-writing Sample Form. The writer print the digit/character (*i.e.*, $X$) with the label (*i.e.*, $Y$) provided first.

Table 6: Accuracy (%) of Colored MNIST on IRM setting in Arjovsky et al. (2019). Average over three runs.

|         | IRM            | LaCIM-$d_s$ (**Ours**) | LaCIM-REx (**Ours**) |
|---------|----------------|------------------------|----------------------|
| $m = 2$ | $67.15 \pm 3.79$ | $\mathbf{68.16 \pm 2.13}$ | $67.57 \pm 1.37$ |
| $m = 4$ | $69.37 \pm 1.14$ | $\mathbf{69.55 \pm 1.60}$ | $69.50 \pm 0.57$ |

### 7.12 SUPPLEMENTARY FOR NICO

**Implementation Details** Due to size difference among images, we resize each image into 256×256. The network structure of $p_\theta(z, s|d_s), q_\phi(z, s|x, d_s), p_\theta(x|z, s), p_\theta(y|s)$ for cat/dog classification is the same with the one implemented in early prediction of Alzheimer's Disease with exception of 3D convolution/Deconvolution replaced by 2D ones. For each model, we train for 200 epochs using sgd, with learning rate (lr) set to 0.01, and after every 60 epochs the learning rate is multiplied by lr decay parameter that is set to 0.2. The weight decay coefficients parameter is set to $5 \times 10^{-4}$. The

Table 7: Training and test environments (characterized by $d_s$)

| | cat% on grass | dog% on grass | cat% on snow | cat% on snow |
|---|---|---|---|---|
| | Training Environment | | | |
| Env#1 ($d_s^{e_1}$) | 0.6 | 0.4 | 0.1 | 0.9 |
| Env#2 ($d_s^{e_2}$) | 0.8 | 0.2 | 0.1 | 0.9 |
| Env#3 ($d_s^{e_3}$) | 0.5 | 0.5 | 0.2 | 0.8 |
| Env#4 ($d_s^{e_4}$) | 0.8 | 0.2 | 0.2 | 0.8 |
| Env#5 ($d_s^{e_5}$) | 0.7 | 0.3 | 0.2 | 0.8 |
| Env#6 ($d_s^{e_6}$) | 0.8 | 0.2 | 0.3 | 0.7 |
| Env#7 ($d_s^{e_7}$) | 0.7 | 0.3 | 0.3 | 0.7 |
| Env#8 ($d_s^{e_8}$) | 0.9 | 0.1 | 0.3 | 0.7 |
| Env#9 ($d_s^{e_9}$) | 0.4 | 0.6 | 0.3 | 0.7 |
| Env#10 ($d_s^{e_{10}}$) | 0.6 | 0.4 | 0.3 | 0.7 |
| Env#11 ($d_s^{e_{11}}$) | 0.5 | 0.5 | 0.4 | 0.6 |
| Env#12 ($d_s^{e_{12}}$) | 0.4 | 0.6 | 0.4 | 0.6 |
| Env#13 ($d_s^{e_{13}}$) | 0.7 | 0.3 | 0.4 | 0.6 |
| Env#14 ($d_s^{e_{14}}$) | 0.8 | 0.2 | 0.4 | 0.6 |
| | Testing Environment | | | |
| Env Test $d_s^{\text{test}}$ | 0.2 | 0.8 | 0.8 | 0.2 |

Table 8: Comparison on constructed interventional dataset in terms of ACC.

| Method | CE $X \rightarrow Y$ | IRM | DANN | NCBB | MMD-AAE | DIVA | LaCIM-$d$ |
|---|---|---|---|---|---|---|---|
| ACC | 52.50 | 50.00 | 49.17 | 49.17 | 49.17 | 50.00 | 55.00 |

batch size is set to 30. The training environments which is characterized by $c$ can be referenced in Table 7.12. For visualization, we implemented the gradient-based method Simonyan et al. (2013) to visualize the neuron (in fully connected layer for both CE $x \rightarrow y$ and CE $(x, d_s) \rightarrow y$; in $s$ layer for LaCIM-$d_s$) that is most correlated to label $y$.

**The $d_s$ for $m$ environments** We summarize the $d_s$ of $m = 8$ and $m = 14$ environments in Table 7.12. As shown, the value of $d_s$ in the test domain is the extrapolation of the training environments, *i.e.*, the $d_s^{\text{test}}$ is not included in the convex hull of $\{d^{e_i}\}_{i=1}^{14}$.

**More Visualization Results** Fig. 7 shows more visualization results.

**Results on Intervened Data.** We test our model and the baseline on intervened data, in which each image is generated by intervention on $Z$, i.e., taking a specific value of $Z$. This intervention breaks the correlation between $S$ and $Z$, thus the distribution of which can be regarded as a specific type of OOD. Specifically, we replace the scene of an image with the scene from the another image, as shown in Fig. 8. We generate 120 images, including 30 images of types: cat on grass, dog on grass, cat on snow, and dog on grass. We evaluate LaCIM-$d$, CE $X \rightarrow Y$, IRM, DANN, NCBB, MMD-AAE, and DIVA methods on this intervened dataset. As shown in Tab 9, our LaCIM-$d$ can performs the best among all methods, which validate the robustness of our LaCIM.

## 7.13 DISEASE PREDICTION OF ALZHEIMER'S DISEASE

**Dataset Description.** The dataset contains in total 317 samples with 48 AD, 75 NC, and 194 MCI.

**Denotation of Attributes $d_s$.** The $C \in \mathbb{R}^9$ includes personal attributes (*e.g.*, age Guerreiro and Bras (2015), gender Vina and Lloret (2010) and education years Mortimer (1997) that play as potential

Table 9: Comparison on constructed interventional dataset in terms of ACC.

| Method | CE $X \rightarrow Y$ | IRM | DANN | NCBB | MMD-AAE | DIVA | LaCIM-$d$ |
|---|---|---|---|---|---|---|---|
| ACC | 52.50 | 50.00 | 49.17 | 49.17 | 49.17 | 50.00 | 55.00 |

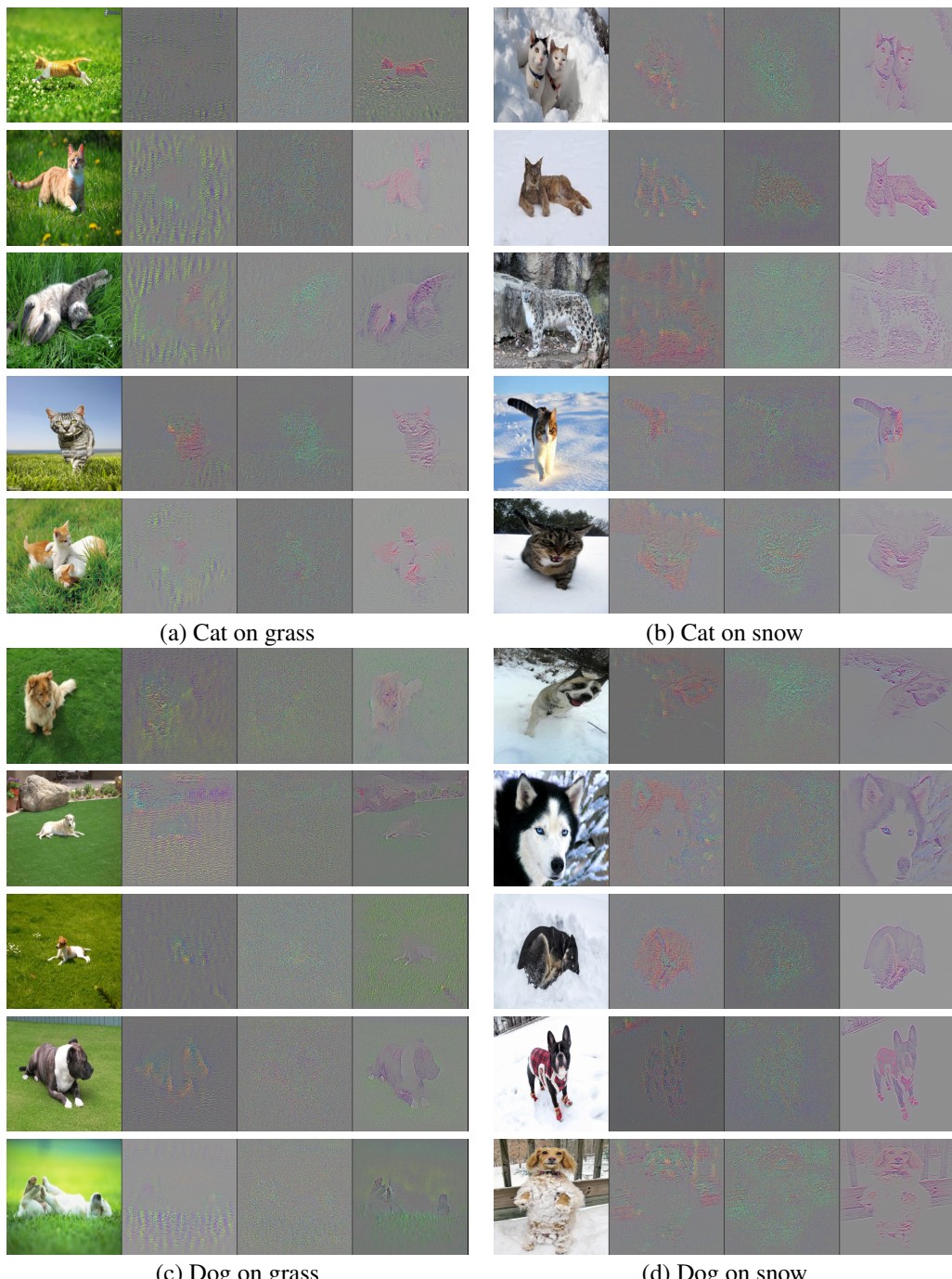

Figure 7: Visualization on the NICO via gradient-based method Simonyan et al. (2013) for CE $X \to Y$, CE $(X, d_s) \to Y$ and LaCIM. The selected images are (a) cat on grass, (b) cat on snow, (c) dog on grass and (d) dog on snow.

risks of AD), gene ($\varepsilon_4$ allele), and biomarkers (*e.g.*, changes of CSF, TAU, PTAU, amyloid$_\beta$, cortical amyloid deposition (AV45) Humpel and Hochstrasser (2011)).

**Implementation Details** For LaCIM-$d_s$, we parameterize inference model $q_\psi(s, z|x, d_s)$, $p_\phi(s, z|d_s)$, $p_\phi(x|z, s)$ and $p_\phi(y|s)$ and $S, Z \in \mathbb{R}^{64}$. For $q_\psi(s, z|x, d_s)$, we concatenate outputs of feature extractors of $X$ and $d_s$: the feature extractor for $x$ is composed of four Convolution-Batch

Dog on snow

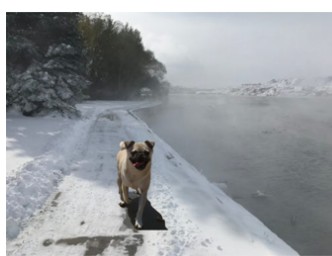

Cat on snow

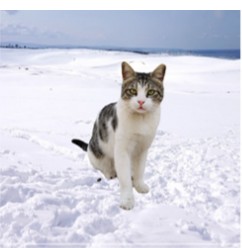

Dog on grass

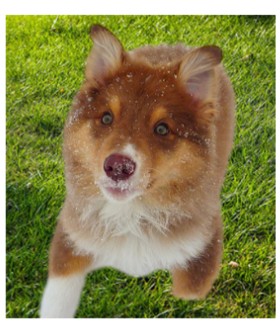

Cat on grass

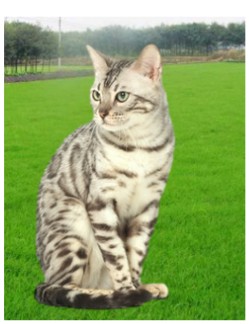

Figure 8: The constructed interventional dataset which includes of dog on snow, dog on grass, cat on snow, and dog on grass.

Normalization-ReLU (CBNR) blocks and four Convolution-Batch Normalization-ReLU-MaxPooling (CBNR-MP) blocks with structure 64 BNR → 128 CBNR-MP → 128 CBNR → 256 CBNR-MP → 256 CBNR → 512 CBNR-MP → 512 CBNR → 1024 CBNR-MP; the feature extractor of $c$ is composed of three Fully Connection-Batch Normalization-ReLU (FC-BNR) blocks with structure $128 → 256 → 512$. The concatenated features are further transformed by four 64 FC-BNR to generate $\mu_{s,z}(x, d_s)$ and $\log \sigma_{s,z}(x, d_s)$. For the prior model $p_\theta(s, z|d_s)$, it shares the same structure without feature extractor of $x$. For $p_\phi(x|s, z)$, the network is composed of three DeConvolution-Batch Normalization-ReLU (DCBNR) blocks and three Convolution-Batch Normalization-ReLU (CBNR) blocks, followed by a convolutional layer, with structure 256 DCBNR → 256 CBNR → 128 DCBNR → 128 CBNR → 64 DCBNR → 64 CBNR → 48 Conv. For $p_\phi(y|s)$, the network is composed of 256 FC-BNR → 512 FC-BNR → 3 FC-BNR. For prior model $p_\phi(s, z|d_s)\mathcal{N}(\mu_{s,z}(d_s), \mathrm{diag}(\sigma^2_{s,z}(d_s)))$ the $\mu_{s,z}(x, d_s)$ and $\log \sigma_{s,z}(x, d_s)$ are parameterized by Multi Perceptron Neural Network (MLP). The decoders $p_\phi(x|s, z)$ are $p_\phi(y|s)$ parameterized by Deconvolutional neural network. For all methods, we train for 200 epochs using SGD with weight decay $2 \times 10^{-4}$ and learning rate 0.01 and is multiplied by 0.2 after every 60 epochs. The batch size is set to 4. For each variable in biomarker vector $C \in \mathbb{R}^9$, each person may have multiple records, and we take its median as representative to avoid extreme values due to device abnormality.

As for LaCIM-d, we adopt the same decoder $p_\phi(x|z, s)$ and classifier $p_\phi(y|s)$. For $q_\psi(s, z|x, d)$, we adopt the same network for the shared part; for the part specific to each domain, $\mu_{s,z}(x, d)$ and $\log \sigma_{s,z}(x, d)$ are generated by the sub-network which is composed of 1024 FC-BNR → 1024 FC-BNR → $q_{z,s}$ FC-BNR. The $z, s$ can be reparameterized by $\mu_{s,z}(x, d)$ and $\log \sigma_{s,z}(x, d)$ are fed into a sub-network which is composed of $q_{z,s}$ FC-BNR → 1024 FC-BNR → $q_{z,s}$ FC-BNR to get rid of the constraint of Gaussian distribution. Then the reconstructed images and predicted label are computed by $p_\phi(x|z, s)$ and $p_\phi(y|s)$ which have the same network structure of LaCIM-C with the $z, s$.

Table 10: Training and test environments (characterized by $c$) in early prediction of AD

|  | Training Env#1 | Training Env#1 | Test |
|---|---|---|---|
|  | Age | | |
| Number of AD | 17 | 17 | 14 |
| Number of MCI | 76 | 83 | 35 |
| Number of NC | 34 | 27 | 14 |
| Average value of $d_s$ (years): | 68.75 | 72.78 | 81.74 |
|  | TAU | | |
| Number of AD | 11 | 22 | 15 |
| Number of MCI | 75 | 78 | 41 |
| Number of NC | 40 | 27 | 18 |
| Average value of $d_s$: | 215.34 | 286.69 | 471.72 |

**The $d_s$ variable in training and test.** The selected attributes include Education Years, Age, Gender (0 denotes male and 1 denotes female), AV45, amyloid$_\beta$ and TAU. We split the data into $m = 2$ training environments and test according to different value of $d_s$. The Tab. 7.13 describes the data distribution in terms of number of samples, the value of $d_s$ (Age and TAU).

### 7.13.1 EXPERIMENTS WITH COMPLETE OBSERVABLE SOURCE VARIABLE

In image-based diagnosis, the personal attributes, genes and biomarkers are often available. Therefore, we consider the setting when $d_s$ can be fully observed. In this case, the value of $d_s$ is person-by-person. Therefore, the number of environments $m$ is equal to the number of samples. In this case, the dataset turns to $\{x_i, y_i, d_s^i\}_{i=1}^n$. The expected risk turns to:

$$\mathcal{L}_{\psi,\phi} = \mathbb{E}_{p(x,y|d_s)} \left[ -\log q_\psi(y|x,d_s) - \mathbb{E}_{q_\psi(s,z|x,d_s)} \left[ \frac{p_\phi(y|s)}{q_\psi(y|x,d_s)} \log \frac{p_\phi(x|s,z)p_\phi(s,z|d_s)}{q_\psi(s,z|x,d_s)} \right] \right]. \tag{73}$$

And the corresponding empirical risk is:

$$\tilde{\mathcal{L}}_{\psi,\phi} = \frac{1}{n} \left[ -\log q_\psi(y_i|x_i,d_{s,i}) - \mathbb{E}_{q_\psi(s,z|x_i,d_{s,i})} \left[ \frac{q_\psi(y_i|s)}{q_\psi(y_i|x_i,d_{s,i})} \log \frac{p_\phi(x_i|s,z)p_\phi(s,z|d_{s,i})}{q_\psi(s,z|x_i,d_{s,i})} \right] \right]. \tag{74}$$

The $d_s$ here is re-defined as the 9-dimensional vector that includes all attributes, genes and biomarkers mentioned above. We re-split the data into 80% train and 20% test, according to different average value of specific variable in the whole vector $d_s$.

**The $d_s$ variable in training and test** We implemented OOD tasks in which the value of $d_s$ is different between training and test. Specifically, we repeatedly split the dataset into the training and the test according to a selected attribute in $d_s$ for three times. The average value of these attributes in train and test are recorded in Table 7.13.1.

**Experimental Results** We conduct OOD experiments with source variables Age, Gender, amyloid$_\beta$ and TAU different between training data and the test. The results are shown in Table 12.

### 7.14 SUPPLEMENTARY FOR DEEPFAKE

**Implementation Details.** We implement data augmentations, specifically images with 30 angle rotation, with flipping horizontally with 50% probability. We additionally apply random compressing techniques, such as JpegCompression. For inference model, we adopt Efficient-B5 Tan and Le (2019), with the detailed network structure as: FC(2048, 2048) $\rightarrow$ BN $\rightarrow$ ReLU $\rightarrow$ FC(2048, 2048) $\rightarrow$ BN $\rightarrow$ ReLU $\rightarrow$ FC(2048, $q_{t=s,z}$). The structure of reparameterization, *i.e.*, $\rho_{t=s,z}$ is FC($q_{t=s,z}$, 2048) $\rightarrow$ BN $\rightarrow$ ReLU $\rightarrow$ FC(2048, 2048) $\rightarrow$ BN $\rightarrow$ ReLU $\rightarrow$ FC(2048, $q_{t=s,z}$). The network structure for generative model, *i.e.*, $p_\psi(x|s,z)$ is TConv-BN-ReLU($q_{t=s,z}$, 256) $\rightarrow$ TConv-BN-ReLU(256, 128) $\rightarrow$ TConv-BN-ReLU(128, 64)$\rightarrow$ TConv-BN-ReLU(64, 32) $\rightarrow$ TConv-BN-ReLU(32, 32) $\rightarrow$ TConv-BN-ReLU(32, 16) $\rightarrow$ TConv-BN-ReLU(16, 16) $\rightarrow$ Conv-BN-ReLU(16, 3) $\rightarrow$ Sigmoid,

Table 11: Training and test environments (characterized by $c$) in early prediction of AD

| | Education Years | Age | Gender (0/1) | AV45 | amyloid$_\beta$ | TAU |
|---|---|---|---|---|---|---|
| | | | Setting #1 | | | |
| Training | 15.34 | 70.56 | 1.29 | 1.21 | 745.61 | 249.38 |
| Test | 19.44 | 81.74 | 1.83 | 1.56 | 1322.47 | 471.72 |
| | | | Setting #2 | | | |
| Training | 15.34 | 70.62 | 1.29 | 1.21 | 743.01 | 250.21 |
| Test | 19.43 | 81.19 | 1.83 | 1.57 | 1332.94 | 446.67 |
| | | | Setting #3 | | | |
| Training | 15.34 | 70.62 | 1.29 | 1.21 | 743.39 | 254.7 |
| Test | 19.44 | 81.19 | 1.83 | 1.56 | 1331.4 | 446.9 |

Table 12: Accuracy (%) of OOD prediction on ADNI. Average over three runs.

| ACC (%) / Method | Setting#1 | Setting#2 | Setting#3 | Setting#1 | Setting#2 | Setting#3 |
|---|---|---|---|---|---|---|
| OOD source | | Education Years | | | AV45 | |
| CE $X \to Y$ | $61.9 \pm 0.0$ | $66.7 \pm 1.6$ | $63.0 \pm 0.9$ | $67.7 \pm 0.9$ | $66.1 \pm 3.3$ | $66.1 \pm 1.8$ |
| DANN | $62.4 \pm 0.9$ | $62.4 \pm 0.9$ | $63.0 \pm 1.8$ | $64.6 \pm 0.9$ | $67.2 \pm 0.9$ | $66.1 \pm 0.9$ |
| CE $(X, d_s) \to Y$ | $67.2 \pm 1.8$ | $66.7 \pm 3.2$ | $63.0 \pm 1.8$ | $66.1 \pm 3.3$ | $66.1 \pm 1.8$ | $64.0 \pm 0.9$ |
| sVAE | $67.2 \pm 0.9$ | $67.2 \pm 0.9$ | $67.2 \pm 0.9$ | $65.6 \pm 1.8$ | $66.7 \pm 2.7$ | $65.1 \pm 1.6$ |
| LaCIM-$d_s$ (**Ours**) | $\mathbf{69.8 \pm 1.6}$ | $\mathbf{68.8 \pm 0.9}$ | $\mathbf{69.8 \pm 1.6}$ | $\mathbf{69.3 \pm 1.8}$ | $\mathbf{67.7 \pm 0.9}$ | $\mathbf{67.7 \pm 0.0}$ |
| OOD source | | Age | | | Gender | |
| CE $X \to Y$ | $63.6 \pm 2.6$ | $65.6 \pm 6.0$ | $64.8 \pm 4.7$ | $60.5 \pm 0.9$ | $60.5 \pm 1.8$ | $60.5 \pm 0.9$ |
| DANN | $60.8 \pm 1.8$ | $58.7 \pm 0.0$ | $58.7 \pm 0.0$ | $58.5 \pm 1.5$ | $61.5 \pm 0.0$ | $60 \pm 1.5$ |
| CE $(X, d_s) \to Y$ | $60.4 \pm 2.9$ | $64.5 \pm 2.4$ | $64.4 \pm 3.8$ | $63.2 \pm 0.9$ | $65.6 \pm 1.8$ | $64.1 \pm 0.9$ |
| sVAE | $58.2 \pm 0.9$ | $60.0 \pm 1.8$ | $58.7 \pm 1.6$ | $64.1 \pm 0.9$ | $65.6 \pm 1.8$ | $64.1 \pm 0.9$ |
| LaCIM-$d_s$ (**Ours**) | $\mathbf{64.0 \pm 2.4}$ | $\mathbf{70.4 \pm 2.4}$ | $\mathbf{66.1 \pm 3.7}$ | $\mathbf{65.6 \pm 0.9}$ | $\mathbf{67.2 \pm 1.8}$ | $\mathbf{68.2 \pm 0.9}$ |

| ACC (%) / Method | Setting#1 | Setting#2 | Setting#3 | Setting#1 | Setting#2 | Setting#3 |
|---|---|---|---|---|---|---|
| OOD source | | amyloid$_\beta$ | | | TAU | |
| CE $X \to Y$ | $59.2 \pm 0.9$ | $63.5 \pm 4.2$ | $63.1 \pm 5.1$ | $64.6 \pm 0.9$ | $64.1 \pm 0.0$ | $66.0 \pm 1.1$ |
| DANN | $60.8 \pm 0.9$ | $60.8 \pm 0.9$ | $60.8 \pm 0.9$ | $64.6 \pm 0.9$ | $65.1 \pm 0.9$ | $64.6 \pm 0.9$ |
| CE $(X, d_s) \to Y$ | $64.6 \pm 1.8$ | $64.6 \pm 3.7$ | $64.2 \pm 2.4$ | $64.6 \pm 0.9$ | $66.7 \pm 0.9$ | $67.0 \pm 1.3$ |
| sVAE | $66.1 \pm 0.9$ | $64.6 \pm 0.9$ | $63.5 \pm 3.2$ | $68.2 \pm 0.9$ | $68.8 \pm 2.7$ | $67.2 \pm 1.6$ |
| LaCIM-$d_s$ (**Ours**) | $\mathbf{68.3 \pm 1.6}$ | $\mathbf{66.1 \pm 1.8}$ | $\mathbf{65.6 \pm 2.4}$ | $\mathbf{69.8 \pm 0.9}$ | $\mathbf{71.4 \pm 1.8}$ | $\mathbf{68.8 \pm 0.0}$ |

followed by cropping the image to the same size $3 \times 224 \times 224$. We set $q_{t=s,z}$ as 1024. We implement SGD as optimizer, with learning rate 0.02, weight decay 0.00005, and run for 9 epochs.

Table 13: General framework table for our method and baselines on Data $\in$ {CMNIST, NICO, ADNI, DeepFake} Dataset. We denote the dimension of $z$ or $zs$ as $\dim_{z,zs}$. We list the output dimension (*e.g.* the channel number) of each module, if it is different from the one in Tab. 14.

| Dataset / Method | CE $X \to Y$ | CE $X, d \to Y$ | MMD-AAE | DANN | DIVA | LaCIM-$d_s$ | LaCIM-$d$ |
|---|---|---|---|---|---|---|---|
| **Data:CMNIST** | Enc$_x^{Data}$ FC(256,dim$_z$) Dec-CE$_y^{Data}$ | Enc$_x^{Data}$;Enc$_d^{Data}$ FC(512,dim$_z$) Dec-CE$_y^{Data}$ | Enc$_x^{Data}$ FC-BN-ReLU(256,256) FC(256,256) $\to z$ Dec$_y^{Data}$; Dec$_x^{Data}$ | Enc$_x^{Data}$ DANN-CLS$_y^{Data}$; DANN-CLS$_y^{Data}$ | $p_\theta^{Data}(x\|z_d,z_x,z_y)$ $p_{\theta_d}^{Data}(z_d\|d)$ $p_{\theta_y}^{Data}(z_y\|y)$ $q_{\phi_d}^{Data}(z_d\|x)$ $q_{\phi_x}^{Data}(z_x\|x)$ $q_{\phi_y}^{Data}(z_y\|x)$ | Enc$_x^{Data}$;Enc$_d^{Data}$ FC-BN-ReLU(512,256) FC(256,dim$_{zs}$) Dec$_y^{Data}$;Dec$_x^{Data}$ prior: Enc$_d^{Data}$ | Enc$_x^{Data}$ Enc$_{Data\times m}^{z,s}$ $\Phi_{Data\times m}^{z,s}$ Dec$_y^{Data}$;Dec$_x^{Data}$ |
| # of Params | 1.12M | 1.16M | 1.23M | 1.1M | 1.69M | 1.09M | 0.92M |
| hyper-Params | lr: 0.1 wd:0.00005 | lr: 0.2 wd: 0.0005 | lr: 0.01 wd: 0.0001 | lr: 0.1 wd: 0.0002 | lr: 0.001 wd: 0.00001 | lr: 0.1 wd: 0.0001 | lr: 0.01 wd: 0.0002 |
| **Data:NICO** | Enc$_x^{Data}$ FC(1024,dim$_z$) Dec-CE$_y^{Data}$ | Enc$_x^{Data}$;Enc$_d^{Data}$ FC(1536,dim$_z$) Dec-CE$_y^{Data}$ | Enc$_x^{Data}$ FC-BN-ReLU(1024,1024) FC(1024,1024) $\to z$ Dec$_y^{Data}$; Dec$_x^{Data}$ | Enc$_x^{Data}$ DANN-CLS$_y^{Data}$; DANN-CLS$_y^{Data}$ | $p_\theta^{Data}(x\|z_d,z_x,z_y)$ $p_{\theta_d}^{Data}(z_d\|d)$ $p_{\theta_y}^{Data}(z_y\|y)$ $q_{\phi_d}^{Data}(z_d\|x)$ $q_{\phi_x}^{Data}(z_x\|x)$ $q_{\phi_y}^{Data}(z_y\|x)$ | Enc$_x^{Data}$;Enc$_d^{Data}$ FC(1536,dim$_{zs}$) Dec$_y^{Data}$;Dec$_x^{Data}$ prior: Enc$_d^{Data}$ | Enc$_x^{Data}$ Enc$_{Data\times m}^{z,s}$ $\Phi_{Data\times m}^{z,s}$ Dec$_y^{Data}$;Dec$_x^{Data}$ |
| # of Params ($m=8$) | 18.08M | 19.01M | 19.70M | 19.13M | 14.86M | 16.31M | 18.25M |
| # of Params ($m=14$) | 18.08M | 19.01M | 19.70M | 26.49M | 14.87M | 18.08M | 19.70M |
| hyper-Params | lr: 0.01 wd: 0.0002 | lr: 0.01 wd: 0.0002 | lr: 0.2 wd: 0.0001 | lr: 0.05 wd: 0.0005 | lr: 0.001 wd: 0.0001 | lr: 0.01 wd: 0.0005 | lr: 0.01 wd: 0.0001 |
| **Data:ADNI** | Enc$_x^{Data}$ FC(1024,dim$_z$) Dec-CE$_y^{Data}$ | Enc$_x^{Data}$;Enc$_d^{Data}$ FC(1536,dim$_z$) Dec-CE$_y^{Data}$ | Enc$_x^{Data}$ FC-BN-ReLU(1024,1024) FC(1024,1024) $\to z$ Dec$_y^{Data}$; Dec$_x^{Data}$ | Enc$_x^{Data}$ DANN-CLS$_y^{Data}$; DANN-CLS$_y^{Data}$ | $p_\theta^{Data}(x\|z_d,z_x,z_y)$ $p_{\theta_d}^{Data}(z_d\|d)$ $p_{\theta_y}^{Data}(z_y\|y)$ $q_{\phi_d}^{Data}(z_d\|x)$ $q_{\phi_x}^{Data}(z_x\|x)$ $q_{\phi_y}^{Data}(z_y\|x)$ | Enc$_x^{Data}$;Enc$_d^{Data}$ FC(1536,dim$_{zs}$) Dec$_y^{Data}$;Dec$_x^{Data}$ prior: Enc$_d^{Data}$ | Enc$_x^{Data}$ Enc$_{Data\times m}^{z,s}$ $\Phi_{Data\times m}^{z,s}$ Dec$_y^{Data}$;Dec$_x^{Data}$ |
| # of Params | 28.27M | 28.27M | 36.68M | 30.21M | 33.22M | 33.07M | 37.78M |
| hyper-Params | lr: 0.01 wd: 0.0002 | lr: 0.01 wd: 0.0002 | lr: 0.005 wd: 0.0002 | lr: 0.01 wd: 0.0002 | lr: 0.005 wd: 0.0001 | lr: 0.005 wd: 0.0002 | lr: 0.01 wd: 0.0002 |

Table 14: Network Structure of Modules used in our method and baselines.

| Method | CMNIST | NICO | ADNI |
|---|---|---|---|
| $\mathrm{Enc}_x^{\mathrm{Data}}$ | Conv-BN-ReLU(dim$_{\mathrm{input}}$,64,3,1,1)
MaxPool(2)
Conv-BN-ReLU(64,128,3,1,1)
MaxPool(2)
Conv-BN-ReLU(128,256,3,1,1)
MaxPool(2)
Conv-BN-ReLU(256,256,3,1,1)
AdaptivePool(1)
Flatten() | Conv-BN-ReLU(dim$_{\mathrm{input}}$,128,3,1,1)
Conv-BN-ReLU(128,256,3,2,0)
MaxPool(2)
Conv-BN-ReLU(256,256,3,1,1)
Conv-BN-ReLU(256,512,3,1,1)
MaxPool(2)
Conv-BN-ReLU(512,512,3,1,1)
Conv-BN-ReLU(512,512,3,1,1)
MaxPool(2)
Conv-BN-ReLU(512,512,3,1,1)
Conv-BN-ReLU(512,1024,3,1,1)
AdaptivePool(1)
Flatten() | Conv3d-BN-ReLU(dim$_{\mathrm{input}}$,128,3,1,1)
Conv3d-BN-ReLU(128,256,3,2,0)
MaxPool(2)
Conv3d-BN-ReLU(256,256,3,1,1)
Conv3d-BN-ReLU(256,512,3,1,1)
MaxPool(2)
Conv3d-BN-ReLU(512,512,3,1,1)
Conv3d-BN-ReLU(512,512,3,1,1)
MaxPool(2)
Conv3d-BN-ReLU(512,512,3,1,1)
Conv3d-BN-ReLU(512,1024,3,1,1)
AdaptivePool(1)
Flatten() |
| $\mathrm{Dec}_x^{\mathrm{Data}}$ | UnFlatten()
Upsample(2)
Tconv-BN-ReLU(dim$_{\mathrm{input}}$,128,2,2,0)
Tconv-BN-ReLU(128,64,2,2,0)
Tconv-BN-ReLU(64,32,2,2,0)
Tconv-BN-ReLU(32,16,2,2,0)
Conv(16,3,3,1,1)
Sigmoid()
Cropping(28) | UnFlatten()
Upsample(16)
Tconv-BN-ReLU(dim$_{\mathrm{input}}$,256,2,2,0)
Conv-BN-ReLU(256,256,3,1,1)
Tconv-BN-ReLU(256,128,2,2,0)
Conv-BN-ReLU(128,128,3,1,1)
Tconv-BN-ReLU(128,64,2,2,0)
Conv-BN-ReLU(64,64,3,1,1)
Tconv-BN-ReLU(64,32,2,2,0)
Conv-BN-ReLU(32,32,3,1,1)
Conv(32,3,3,1,1)
Sigmoid() | UnFlatten()
Upsample(6)
Tconv3d-BN-ReLU(dim$_{\mathrm{input}}$,256,2,2,0)
Conv3d-BN-ReLU(256,256,3,1,1)
Tconv3d-BN-ReLU(256,128,2,2,0)
Conv3d-BN-ReLU(128,128,3,1,1)
Tconv3d-BN-ReLU(128,64,2,2,0)
Conv3d-BN-ReLU(64,64,3,1,1)
Tconv3d-BN-ReLU(64,64,2,2,0)
Conv3d-BN-ReLU(64,64,3,1,1)
Conv3d(64,1,3,1,1)
Sigmoid() |
| $\mathrm{Enc}_d^{\mathrm{Data}}$ | FC-BN-ReLU($d$, 128)
FC-BN-ReLU(128, 256) | FC-BN-ReLU($d$, 256)
FC-BN-ReLU(256, 512)
FC-BN-ReLU(512, 512) | FC-BN-ReLU($d$, 256)
FC-BN-ReLU(256, 512)
FC-BN-ReLU(512, 512) |
| $\mathrm{Dec}_y^{\mathrm{Data}}$ | FC-BN-ReLU(dim$_{z,s}$, 512)
FC-BN-ReLU(512, 256)
FC(256,2) | FC-BN-ReLU(dim$_{z,s}$, 512)
FC-BN-ReLU(512, 256)
FC(256,2) | FC-BN-ReLU(dim$_{z,s}$, 512)
FC-BN-ReLU(512, 256)
FC(256,2) |
| $\mathrm{Dec\text{-}CE}_y^{\mathrm{Data}}$ | FC-BN-ReLU(dim$_{z,s}$, 512)
FC-BN-ReLU(512, 256)
FC(256,2) | FC-BN-ReLU(dim$_{z,s}$, 1024)
FC-BN-ReLU(1024, 2048)
FC(2048,2) | FC-BN-ReLU(dim$_{z,s}$, 512)
FC-BN-ReLU(512, 256)
FC(256,2) |
| $\mathrm{DANN\text{-}CLS}_y^{\mathrm{Data}}$ | FC-BN-ReLU(256, 32)
FC-BN-ReLU(32, 2) | FC-BN-ReLU(1024, 2048)
FC-BN-ReLU(2048, 2) | FC-BN-ReLU(1024, 1024)
FC-BN-ReLU(1024, 2) |
| $\Phi_{z,s}^{\mathrm{Data}}$ | FC-ReLU(dim$_{z,s}$, 256)
FC-ReLU(256, dim$_{z,s}$) | FC-ReLU(dim$_{z,s}$, 1024)
FC-ReLU(1024, dim$_{z,s}$) | FC-ReLU(dim$_{z,s}$, 1024)
FC-ReLU(1024, dim$_{z,s}$) |
| $\mathrm{Enc}_{z,s}^{\mathrm{Data}}$ | FC-ReLU(256, 256)
FC-ReLU(256, dim$_{z,s}$) | FC-ReLU(1024, 1024)
FC-ReLU(1024, dim$_{z,s}$) | FC-ReLU(1024, 1024)
FC-ReLU(1024, dim$_{z,s}$) |
| $p_\theta^{\mathrm{Data}}(x|z_d, z_x, z_y)$ | FC-BN-ReLU(1024)
UnFlatten()
Upsample(8)
TConv-BN-ReLU(64,128,5,1,0)
Upsample(24)
TConv-BN-ReLU(128,256,5,1,0)
Conv(256, 256*3,1,1,0) | FC-BN-ReLU(1024)
UnFlatten()
Upsample(16)
TConv-BN-ReLU(64,128,5,1,0)
Upsample(64)
TConv-BN-ReLU(128,256,5,1,0)
Upsample(256)
Conv(256, 3,1,1,0) | FC-BN-ReLU(1024)
UnFlatten()
Upsample(8)
TConv3d-BN-ReLU(16,64,5,1,0)
Conv3d-BN-ReLU(64,128,3,1,1)
Upsample(24)
TConv3d-BN-ReLU(128,128,5,1,0)
Conv3d-BN-ReLU(128,128,3,1,1)
Upsample(48)
Conv3d-BN-ReLU(128,32,3,1,1)
Conv3d(32, 1,1,1,0) |
| $p_{\theta_d}^{\mathrm{Data}}(z_d|d)$
$p_{\theta_y}^{\mathrm{Data}}(z_y|y)$ | FC-BN-ReLU(dim$_{d,y}$, 64)
FC(64,64); FC(64,64) | FC-BN-ReLU(dim$_{d,y}$, 64)
FC(64,64); FC(64,64) | FC-BN-ReLU(dim$_{d,y}$, 64)
FC(64,64); FC(64,64) |
| $q_{\phi_d}^{\mathrm{Data}}(z_d|x)$
$q_{\phi_x}^{\mathrm{Data}}(z_x|x)$
$q_{\phi_y}^{\mathrm{Data}}(z_y|x)$ | Conv-BN-ReLU(3,32,5,1,0)
MaxPool(2)
Conv-BN-ReLU(32,64,5,1,0)
MaxPool(2)
Flatten()
FC(1024, 64); FC(1024, 64) Data | Conv-BN-ReLU(3,32,3,2,1)
MaxPool(2)
Conv-BN-ReLU(32,64,3,2,1)
MaxPool(2)
Conv-BN-ReLU(64,64,3,2,1)
MaxPool(2)
Flatten()
FC(1024, 64); FC(1024, 64) Data | Conv3d-BN-ReLU(1,64,3,2,1)
Conv3d-BN-ReLU(64,128,3,1,1)
MaxPool(3)
Conv3d-BN-ReLU(128,256,3,1,1)
Conv3d-BN-ReLU(256,256,3,1,1)
MaxPool(2)
Conv3d-BN-ReLU(256,256,3,1,1)
Conv3d-BN-ReLU(256,128,3,1,1)
MaxPool(2)
Flatten()
FC(1024, 64); FC(1024, 64) Data |

