# OpenReview forum: "Latent Causal Invariant Model"
_ICLR.cc/2021/Conference — Reject_

### Official Review · AnonReviewer1 · 2020-10-28
**leveraging causality to learn invariant models across domains**

**Rating:** 5
**Confidence:** 4

**Review:**

This paper proposes a VAE based model for learning latent causal factors given data from multiple domains. Similar to [Kingma and Hyv¨arinen, 2020], it utilizes additional labels as supervision signals and learns the model using a Bayesian optimization approach given a fixed hypothetical causal structure. The identifiability is obtained by assuming the casual mechanism to be domain invariant, which is partially supported by some empirical experiments.
I have three concerns for the current version of the paper.
1.	Directly mitigating the identifiability result of [Kingma and Hyv¨arinen, 2020] to this model seems to be inappropriate. The result of [Kingma and Hyv¨arinen, 2020] shows that the sufficient statistics of the latent code are recoverable. However, it does not mean the causal structure is unique: additional transformations may be allowed to be applied to the adjacency matrix. As a result, it is in question whether your causal mode learns the correct factors under given structure.
2.	The symbols are a bit confusing. Some important concepts stay intuitive and lack rigorous mathematical definitions. For example, “output-causative” “cross-domain causal effect” stability measure etc should be defined. A clear table listing the symbols and its meaning would be helpful.
3.	The empirical results of table 1 do not fully support your conclusion “true causal factors are learnable”. Simply computing a MCC to the original factors is not enough to me. Some experiments, like examining the vulnerability and performance of the system under the condition that the latent factors are controlled or intervened, for the claimed “invariant causal model” are better to be included to convince readers.

---

> ### Author Response · Authors · 2020-11-23
> **Response to Reviewer 4**
>
> Thanks for your valuable suggestions. Here is the response to your concerns:
>
> - There may exist some misunderstandings regarding our identifiability results.
>
>   * First, our result generalizes the one in [1] in the following aspects: 1. In contrast to the unsupervised learning considered in [1], our goal is to disentangle the $y$-causative features, i.e., $S$ from the $y$-non-causative features, i.e., $Z$. 2. Besides, we relax the exponential family assumption on latent variables in theorem 4.4. In addition, our analysis can incorporate the classification task in which the $Y$ is categorically distributed. Both **the conclusion and the corresponding technique** are different, as explained in supplement 7.3.  3. Last but not least, in our updated version, we generalize our results to the setting when the confounder C is allowed to take a value in a sample-level, **rather than the domain-level** (each domain has only 1 fixed value of confounder) considered in [1].
>
>   * Second, our goal is **not** to identify the unique causal structure; rather, the goal is to (i) identify/disentangle the causal factors $S$ from others, i.e., $Z$ during inference; and (ii) identify the invariant causal mechanisms for OOD generalization, as explained in the third paragraph in section 4.1, the theorem 4.3 and the last paragraph in the introduction. Correspondingly, the $\sim_p$-identifiability results in theorem 4.3 (and also definition 4.2) claims that we can identify the latent variables up to linear and point-wise transformation (i.e., $\mathbf{T}^{t=s,z}(f_{x,\mathcal{S}}^{-1}(x)) = M_{t=s,z} \tilde{\mathbf{T}}^{t=s,z}(\tilde f_{x,\mathcal{S}}^{-1}(x)) + a_{t=s,z}$), which implies the disentanglement of the $S$ and $Z$; and that we can learn the ground-truth predicting mechanism, i.e., $p_{f_y}(f_{x,S}^{-1}(x))= p_{\tilde f_y}(\tilde f_{x,\mathcal{S}}^{-1}(x))$ (consider $f_x:=f_x^{\star},f_y=f_y^{\star}$ and the $\tilde f_x, \tilde f_y$ are our estimated structural equations in definition 4.2). In fact, the unique identification is impossible. Consider a simple example, let $\tilde{s} = \frac{1}{2}s, \tilde{f}_x(s, z) = f(2s, z), \tilde{f}_y(s) = f_y(2s)$,
> then $\tilde{x} = \tilde{f}_x(\tilde{s}, z) = f_x(s, z) = x, \tilde{y} = \tilde{f}_y(\tilde{s}) = f_y(s) = y$. That is, $\tilde{f}_x, \tilde{f}_y$ generate the same observational data of $(x, y)$. Therefore, without $s$ and $\tilde{s}$ observed, we could not distinguish between the two data generating processes $f_x, f_y$ and $\tilde{f}_x, \tilde{f}_y$. Sorry for the misleading, we have introduced in the second last paragraph in section 4.1 to briefly introduce the goal of our identifiability result.
>
> - We have modified the concepts to make them more readable, e.g., change the "output-causative" to output($y$)-causative" in the second paragraph of the introduction; and change the "$y$-causative" to "$y$ (output)-causative" in the second paragraph in section 4.1. Thanks for your reminder.
>
>
> - For Tab.1, the result validates the identifiability in theorem 4.3 that we can identify the latent variables up to permutation and point-wise transformation, which is indeed our claim (**rather than unique identification**) is sufficient to disentangle the causal factors from others. Besides, we additionally examine the robustness of our model **on intervened data**, the results are as follows (for detailed discussion please refer to supplement 7.10):
>
> |  	|  	|  	|  	|  	|     |    |    |
> |  -	|  -	|  -	|  -	|  -	|  -  |  - |  - |
> | `Method` 	|  CE ($X\to Y$)	|  DANN 	|     DIVA 	|      IRM 	| MMD-AAE |  NCBB      |  LaCIM-$d$ (**Ours**)
>  | `Accuracy` 	|  	52.5                |   49.17	        |  50	        |  50                |  49.17	      |    49.17	   |  **56.67**
> |  	|  	|  	|  	|  	|     |    |    |
>
>
>
>
>
>
> [1] Khemakhem, Ilyes, et al. "Variational autoencoders and nonlinear ica: A unifying framework." International Conference on Artificial Intelligence and Statistics. PMLR, 2020.

---

### Official Review · AnonReviewer4 · 2020-10-29
**A thorough study on causally robust supervised learning**

**Rating:** 6
**Confidence:** 3

**Review:**

# Summary of the review
- The authors make a strong case regarding their approach to causally robust supervised learning. Their contributions are thorough and carefully situated in the relevant subfields. I believe the authors' research will be of interest to the relevant research communities.

# Summary of the paper
- The authors suggest a causal latent variable model for high dimensional supervised data, such as images (input, $X$) with labels (output, $Y$), which can be used to obtain robust predictions as well as learn disentangled representations for high dimensional input given heterogenuous observational data, collected in different environments $e \in \mathcal{E}$. In their generative model, the authors distinguish two types of latent variables. Continuing the image classification example, a group of latent variables (called $S$ in the paper) correspond to the object level abstractions such as shape and contour, that causally determine the label $Y$ as well as the image $X$. The other group of latent variables (called $Z$ in the paper) correspond to contextual information such as pose or lighting, that causally only effect $X$ but also can be spuriously correlated with $Y$ (see below). Accordingly, the foundation for the authors' model is the robust / invariant / transferable functional relationships between the latent variables and observed variables, $f_x: (S, Z) \to X$ and $f_y: S \to Y$ that are constant across $e$, whereas $P^e(S, Z)$ is subject to change given the environment $e \in \mathcal{E}$. The authors name $f_x$ and $f_y$ Causal Invariant Mechanisms (CIMe). The model is completed by an observed confounder $C$ that causally affects both $S$ and $Z$, and corresponds to information determining the distribution of $P^e(S,Z)$. $C$ enables the backdoor path that spuriously relates $Z$ and $Y$. When $C$ is inaccessible, the authors assume that a binary index $D$ is accessible instead, which serves as a label for the environment which observations were made. This completes the Latent Causal Invariance Model (LaCIM).
- The overall aim of the authors approach is to formulate the learning problem such that when a function mapping input to output $f: X \to Y$ is to be learnt, this learned function is not biased by the spurious correlations between $X$ and $Y$ that likely would not generalize to new environments or tasks.
- Based on their model LaCIM the authors:
    - demonstrate the $\sim_p$ identifiability of $f_x$ and $f_y$
    - propose variational inference methods to approximate these functions given $C$ or $D$
    - conduct experiments demonstrating how their method performs on out-of-distribution (OOD) generalization tasks, interpretability tasks, and adversarial robustness tasks.

# Strenghts of the paper
- The authors model accommodates different types of high-level latent variables that might affect the observations causally or induce spurious relationships between them. The model also makes clear what relationships are expected to be invariant due to being causal mechanisms, an how spurious relationships can be blocked by conditioning on environment properties (if accessible), and. The authors also provide a variant of their model given only the label/index of the environments are accessible.
- Building on the work by Khemakhem et al. (2020), the authors prove the identifiability of the inveriant causal relationships between latent and observed variables, given certain conditions on the number and diversity of environments observed.
- They provide a variational-autoencoder algorithm for training for the case when confounders are observed and unobserved.
- They provide extensive experiments to 1- verify their theoretical claims, 2- show that their model shows the desired robust prediction properties under different settings 3- demonstrate their method estimates meaningful latent variables.
- The authors make extensive comparisons with the literature both methodologically and experimentally. Actually they provide a more thorough review in the supplementary material. They do not mention this additional review in the main text, which I think they should.
- Though being dense, the paper is easy to follow (save for occasional typos, see below).

# Questions and potential improvements
- As the authors note, their identifiability conditions rely on the existence of a virtually unknowable number of different domains. How should we expect the performance of their algorithms to degrade when the number of domains are lower than this unknown number? An initial idea for this could be obtained through a simulation study, where the dimensions of the latent variables could be controlled.
- In the experiments, the LaCIM-D variant of the algorithm seems to perform better in the cases where the back-door paths are less likely to be exhaustively blocked. Given finding such a set of variables should be even more difficult in realistic scenarios, do the authors see a realistic use for LaCIM-C? If so, how would they compare the two algorithms in terms of their applicability in different tasks? Do they see a case when the LaCIM-D would be disadvantageous?
- The authors mention that given insufficient number of environments, the environments/datasets at hand could be e.g. clustered and treated as heterogenuous (as does e.g. Buhlmann 2018). How successful would the authors expect such an approach to be? Though such an approach could be helpful to some extent, it is unlikely that any homogenous data with $\mid \mathcal{E} \mid = 1$ can be successfully treated as heterogenous after some processing.
- The authors mention the diversity condition at Pg. 5 without having introduced it before. The authors might want to clarify this aspect of their exposition.

## Minor comments, typos
- Section 3: The authors' back-door path explanation is hard to understand, they might want to replace it with something to the effect of: "an unblocked, undirected path between $V_a$ and $V_b$ that has edges going into $V_a$ and $V_b$".
- Section 3: $d^e \in \mathbb{R}^m$ could be replaced by $d^e \in {0, 1}^m$.
- Section 3 and later: The abbreviation for structural causal models could be capitalized: SCM
- Section 4: "and priori", "less and equal"
- Section 4.1: "the confounder $C$ blocks the back-door path from $Z$ to $Y$, making $Z$ and $Y$ spuriously correlated": the authors might want to restate this part, when the path is "blocked" ($C$ is observed/conditioned on) $Z$ and $Y$ are no longer spuriously correlated.
- Section 4.1: By "brute force data fitting" the authors must mean an algorithm that tries to directly learn a functional relationship between $X$ and $Y$. They could express this part a little more clearly.
- Section 5: title "Experiment"s
- The word "condition" in Theorem 4.3 is capitalized, I am assuming by mistake.

---

> ### Author Response · Authors · 2020-11-23
> **Response to Reviewer 3.**
>
> Thanks for your supportive and valuable suggestions. We will explain our update and responses to your questions as follows:
>
> - About the identifiability. We additionally conduct our LaCIM-$d$ on a various number of domains, i.e., $m=3,m=5,m=7$, as marked by the appended lines in Tab.1 (LaCIM-$d$ (ours with $m=3$) and LaCIM-$d$ (ours with $m=7$)). Note that in our setting, the threshold for the number of environments to satisfy the diversity condition (in theorem 4.3) is $m = max(k_z \times q_z,k_s \times q_s)+1 = 5$ ($q_z=q_s=2$, $k_z=k_s=2$). As shown in Tab.1, more environments can result in higher MCC (as a measurement of identifiability); besides, although the $m=3$ does not satisfy the diversity condition, it performs largely better than only pool-LaCIM which mixes all data together. This can validate the effectiveness of multiple domains in avoiding spurious correlation, compared to only 1 domain.
>
> - About LaCIM-$C$ v.s. LaCIM-$D$. When the source variable of each domain is fully observed, the distributional change across domains can be completely explained away. In this case, the LaCIM-$C$ is the ceiling of LaCIM-$D$ and can generally perform better than LaCIM-$D$ since it can leverage more information, as shown in CMNIST in which the source variable is denoted as the correlation between the color and the digit label. However, in many realistic scenarios, it generally not tractable to require all the source variables to be observed (i.e., the partially observed source variable). In this case, the LaCIM-$C$ may perform poorer than LaCIM-$D$, due to the existence of unobserved source variables, as illustrated in the result of ADNI. The efficacy of LaCIM-$C$ is task-dependent. When the observed confounders are assumed to explain the distributional change, it is encouraged to apply it since it can leverage more information for easier learning. Otherwise, we suggest implementing LaCIM-$D$, in which the source variable is not required to be observed.
>
> - About the Clustering method when $|\mathcal{E}|=1$. We can implement the unsupervised clustering method to split the dataset with heterogeneous data. For example, in our DeepFake experiment, we split the dataset according to the source ID of the data. When $|\mathcal{E}|=1$ with homogeneous data, these methods (to the best of our knowledge, all existing methods) cannot make any effect anymore since it is impossible to differentiate the causation from the correlation if only the input and the output are provided for learning.
>
> - About the typos. We define the diversity condition in our theorem 4.3 and correct other typos. We appreciate your great efforts and suggestions.

---

### Official Review · AnonReviewer2 · 2020-10-29
**Assumptions Risk Taking Away the Essence of the Problem**

**Rating:** 4
**Confidence:** 4

**Review:**

I would like to thank the authors for the interesting work they proposed. I tried to explain my concerns below and I am open to changing my score with their feedback.

1) My main concern is whether the considered causal graph indeed captures the phenomenon that the authors are attempting to address. What I mean is that, yes, indeed C induces correlation between Z and Y. But only in the combined dataset where we do not fix C. For each realization of C=c, Z and Y become independent. This is in contrast with what we experience in practice, e.g., we see cows on pasture sceneries more often than not in a single dataset. So the correlation should exist even when we only look at a single dataset. Then this correlation may change in a different dataset where we see cows on the beach. In both datasets, there is spurious correlation which needs to be resolved. The only reason there is spurious correlation in data obtained from this graph is because we assume C is not conditioned on, i.e., we are given mixed data from multiple environments. One of the proposals of IRM was to make use of this environment information, rather than work with the combined data and still there are many challenges there. Theirs is one attempt to address this. In short, I am not certain if this causal graph is suitable to model the phenomenon we are facing in ML today, which the authors talk about in the introduction.

2) Another concern is about the invertibility of f_x to obtain s,z. If s can be uniquely obtained from x, then how does C affect p(y|x) across different environments? Writing down Bayes rule, normally posterior is affected by p^e(s,z) but when p(x|s,z) is deterministic, this is not the case: For each x, one can uniquely go to s only to then go to y, without C playing any role in the process. Then I think this invertibility assumption might make the effect of C void. Please comment on this.

3) When the new (test) environment is one of the environments that we trained on (c^e is known) then we can just use the training data p(y|x) in that environment. Why is this analysis needed then for that case?

4) When the new environment is not known (paragraph before Section 5), the authors propose optimizing s and z and to maximize the data likelihood for the trained model and finding the most likely y given this s. This is connected to the second condition of Definition 4.2, correct? In other words, under the assumptions, this inference will give the correct distribution p(y|x) in ANY new environment. Please verify if I am understanding the result correctly here. But then again, based on 2) above, I am having a hard time seeing the value of the identifiability result under these assumptions.

5) Assumptions are revealed sequentially one by one and too late in the paper. It would also be great if the authors could make an itemized list of assumptions (ANM, exponential family, invertibility of functions etc.).

The following are my detailed feedback and further questions:

The authors assume a simple causal graph where features (image) and labels do not cause each other but are caused by latent factors that are affected by the environment (different interventions). To get to p(y|x) in a way that does not depend on the environment to be able to do inference on new data, the authors attempt to learn the mechanism to generate S from X and Y, where S is not observed, for each environment. They show that under a number of assumptions, an identifiability result arises, i.e., there is a set of models which all give the same \sum p(s|x)p(y|s), implying that we can predict Y from X for any new environment and this can be recovered from p^e(x,y) given enough number of diverse environments.

The narrative is a little overwhelming and unnecessarily confusing in that, very simply, authors propose using a specific causal graph, with specific modeling assumptions within the SCM framework. Their main contribution is the identifiability result which shows that under the given assumptions, we can identify an invariant mechanism p(y|x) that does not depend on the environment (if i am understanding the theorem correctly).

Experimental sections are too briefly described and should be expanded on. Also please comment of Figure 2.

Please explain the variables corresponding to those in the considered causal graph in each real experiment. This is very important to assess the validity of the analyzed causal graph and the assumed missing edges between the variables. For example, what are the variables in CMNIST? I believe Color should cause Image but not the Label and Digit should cause both the Image and the Label. But the narrative is different than this.

Additive noise model, which is a huge assumption is only revealed in page 4, whereas claims about identifiability of the SCM are made starting abstract. This should be given much before. ANM assumption is made for the SCM of both X and Y.

"Information intersection property" in first par. of Sec. 4.2
f_y^{-1}(y')=[f^{-1}]_S(x') for all x',y' will not be true unless the exogenous variables are zero since y=f(x)+e. What am I missing here? Maybe the authors mean invertibility from f(x) back to x rather than from y to x.

Please explain the notation in Definition 4.2 in detail. What is [f_x^{-1}]_S(x)? Without these, it is very hard to parse the statement. I was able to decode these eventually but they need to be introduced for ease of reading.

Other notation is also very confusing. Boldface capital T is a matrix but also a function that takes inputs. Again, I can decode what authors meant but a more rigorous and precise notation is needed.

"It is shown in supplement 7.2 that ∼p is an equivalence relation."
Can you please elaborate in the main text?

There is some mismatch with the experiments and the theory. Additive noise models and the exponential families enable the identifiability results whereas experiments use a VAE formulation. Why not assume the additive noise models and the exponential families in the experiments as well? Then I am sure other inference methods could be used to discover the latent space from the exponential family and still leverage the proposed theory for identification. Did you try this and does it not perform as well? Please comment if possible.

Can you give intuition on why the identifiability result requires a condition on the rank of \Gamma matrix but not on the other parameters? For example, what if all T's are zeros? Then no condition on Gamma should be sufficient since it doesn't appear in the equations anymore. How is this and similar corner cases are covered by the theorem?

"For causal prediction, the old-school causal learning frameworks (Peters et al., 2016; Buhlmann, 2018) causally related the output label Y to the observed input X, which however is NOT conceptually reasonable in scenarios with sensory-level observed data (e.g. pixels in image classification)."
I understand what this implies but it is very implicit. Please consider elaborating: Modeling pixels as variables of a causal graph does not make much sense.

"For such applications, we rather adopt the manner in"
This divergence is also confusing. The hinted difference is not a fundamental difference but a difference in how one models the causal structure.

"into a novel causal model"
I would consider changing this statement. The authors assume a specific causal graph, calling it a novel causal model does not seem accurate.

Figure 1 is interesting because it makes the strong assumption that the features X do not cause the label Y. It also makes the strong assumption that S and Z cannot cause each other. Please comment on these non-causality restrictions imposed by the graph.

"Notably, far beyond the scope in existing literature (Khemakhem, Kingma and Hyvarinen, 2020), our results can implicitly, and are the first to disentangle the output-causative factors (a.k.a, S) from others (a.k.a, Z) for prediction, to ensure the isolation of undesired spurious correlation."
I believe the related work for identifiability of causal factors should be much richer than this. Please consider including identifiability of SCMs in linear and related settings from the causal inference literature.

I do not think there is a need for a new name "Causal Invariant Mechanisms (CIMe)" where what the authors mean is to learn the structural equations in the SCM alongside the distributions.

"From the perspective of causality, the confounder C blocks the back-door path from Z to Y , making the Z spuriously correlated with Y ."
Maybe splitting into two sentences will help with clarity here. The path induces the correlation, not blocking the path by C.

"brutal-force"->"brute-force"

In (1), it looks like the joint distribution on S,Z is assumed to factorize as p(z)p(s), i.e., they are independent. But this is per environment, indicated by the index e. I believe it is better if the authors could write this as p(s,z|c) instead of p^{e}(s,z).

The arxiv identifier of Tan et al. is in boldface.

I think the authors should discuss identifiability of SCMs especially in the ANM setting more explicitly. It would also help to discuss that once the causal graph is known, under the ANM assumption one can trivially obtain the SCMs. The challenge arises when there are latent factors, which is the setting considered here.

What is assumption (4) mentioned in the proof? Is it the fullrank condition written in caps? Please cross-reference.

Causal Markov condition is invoked before (3) to factorize interventional distribution, but this connection between different environments and them essentially representing different interventions is not made before. I think this will be very helpful if the authors enable the connection.

%%% After the Author Responses and Paper Updates %%%

I would like to thank the authors for very seriously considering my recommendations and genuinely attempting to implement many of them, even in their proofs. I do think their updates made the paper stronger in general. A minor general note is that many of the newly edited sections have typos and could benefit from proof-reading. Following are my final remarks:

After going through the paper again, I realized a step in the proofs which indicates an extra assumption that is not mentioned in the main paper. The proof in lines (14,15) of Section 7 (supplementary material) seems to assume that the exogenous variable has a fixed distribution, i.e., despite two different models inducing the same observed distribution having two different functions f_x and \tilde{f_x}, their exogenous noise \epsilon_x must have identical distributions for the proof to go through. Similar steps are used for the exogenous noise of Y as well. These steps can only be explained by the very strong assumption that the exogenous noise term of every variable has a fixed distribution across different causal models. Such an assumption is not mentioned anywhere in the paper as far as I can see - this has to be definitely addressed. Moreover, I do not think the identifiability result is very useful when obtained under such a strong assumption. Unfortunately, I cannot recommend acceptance due to this. But I encourage the authors to pursue this direction and seek out ways to relax this condition.

I had brought up the point that C is not used to induce correlation within a dataset, which takes away from the essence of the practical problem they are trying to address. In order to address this, i.e., to be able to handle the confounding within each dataset, the authors added an extra condition on the effect of confounding: They assume that once p_1(x,y)=p_2(x,y), this implies p_1(x,y|c)=p_2(x,y|c) for all c. This extra assumption allows the authors to use the machinery they developed as is with this additional argument.

Unfortunately, this assumption, much like the others, is also presented in between the lines. I think it will be really helpful if the authors could explicitly write down their assumptions in a theorem environment (
\begin{assumption} ... \end{assumption}
) and make them very explicit rather than only within the theorems or in-text: Assumption 1: ANM Assumption 2: Exponential family Assumption 3: ... etc.

This relates to the implicitness of another key assumption made in the paper: \Gamma matrix is assumed to be full rank. This intuitively suggests that the experimental conditions are sufficiently different. But this is an algebraic statement and is hard to interpret. I would recommend the authors to think about how to interpret this condition, i.e., assess how it impacts the conditional distributions - exponential assumption allows them to make an algebraic assumption here, rather than probabilistic; but a probabilistic interpretation would be more intuitive.

Some of the typos that I can see: "and functions the prior of distribution of C"->"and functions as the prior of distribution of C"

typo in (62): "R"->"r"

"We generalize the identifiable result in theorem 4.3 "->"We generalize the identifiability result in theorem 4.3 "

Many of the arXiv citations are actually published in various venues, please go through the bibliography and update.

---

> ### Author Response · Authors · 2020-11-23
> **Response to Reviewer 2 (part1)**
>
> We are very grateful for your great efforts and valuable suggestions! We will explain our modifications and respond to your concerns as follows:
>
> - About the role of the causal graph to explain the spurious correlation. Thanks for your valuable suggestion!  Indeed, the causal graph in our previous version can only cover the case when there exists a spurious correlation between domains and the label. To make our work more general and can explain the correlation between $Z$ and the label $Y$ in a single dataset, **we extend our causal model** to allow that each confounder $C$ can take a value for each sample unit. Besides, we correspondingly **extend our identifiability result** in theorem 4.3 (for $S, Z$ belongs to exponential family) to this scenario (the proof is in section 7.3 on page 21). In our current version, as shown in our definition 4.1 associated with our causal graph in Fig.1, the source variable denoted as $D$ (domain index as a substitute) takes a specific value for each domain, which functions as a priori for the confounder $C$. Further, this domain-dependent $P(C|d^e)$ can explain the distributional change of $P(S,Z|d^e)$ across domains, which refers to the varied correlations of each domain, as mentioned in your comments. Reflected in the generating mechanism, each domain has a fixed value of source variable, i.e., $d^e$, then it generates a value of $c_i$ from $p(c|d^e)$ for each sample $i$, followed by $s_i,z_i$ via $p(s,z|c_i)$ and $x_i,y_i$ via $p(x,y|s_i,z_i)$. We modify Fig.1 and the description, the definition 4.1, and the paragraph before it. As an illustration, we also make the cat/dog classification, in which the associated scene can be correlated with the cat or dog. For example, the dog comes up on the grass more often than on the snow in a single dataset, which is known as sample selection bias. The source variable $D$ can thus denote the sampler, which affects the confounder $C$ that refers to the (time, weather) to collect samples. For the sampler who prefers to go out in the sunny morning or in the quiet and snowy evening, he correspondingly tends to see that the dog is more associated with the grass and the cat is more associated with the snow. Moreover, our identifiability result can generalize to this case, which can refer to the updated theorem 4.3, section 7.3, and 7.4.
>
> - About why not $X \to Y$ and $S \to Z$ (or $Z \to S$). For why we do not assume that $X$ cause $Y$, we have explained in the introduction and in section 4.1 that in many scenarios with sensory level data, the causal factors should be high-level abstractions, such as the shape, light rather than pixels in $X$ in image classification (in applications when the input is attribute-level, there can be $X \to Y$, which is however beyond the scope of this paper). For the association between $S$ and $Z$, if $Z \to S$, then the $Z$ can also be thought of as the causal factors of $S$ and also the (indirect) cause of $Y$. Rather, we model $Z$ to be spuriously correlated to $Y$. This correlation can be learned if directly learn from $X$ to $Y$, which is known as spurious correlation. Therefore, rather than causation, we model this association between $S$ and $Z$ by introducing an additional confounder. This assumption can better explain existing phenomena of suffering from spurious correlation, e.g., the contextual information in object classification or the color in CMNIST, compared to that $Z \to S$. The assumption of $S \to Z$ is interesting since the $Z$ is also correlated but not causally to $Y$. Although this is beyond the scope of tasks considered in this work, we would like to consider it in the future.
>
> - About the ANM assumption. The ANM assumption states that the $f_x(s,z,\varepsilon_x)$ is assumed to be $\hat{f}_x(s,z) + \varepsilon_x$, which can cover a broad family of both continuous and discrete variables and has been widely adopted in the literature of causal inference and Independent Component Analysis (ICA). In our scenario, we only require the ANM for the generation of continuous-variable $X$. For $Y$, we first narrow our interest to ANM for $Y$; **but more importantly**, we extend this analysis to a more general setting which can **relax this ANM assumption for $Y$** and can **incorporate the classification task** when $Y$ is categorical distribution, as explained in **theorem 4.4**. To make it more readable, we introduce the ANM assumption together with our identifiability earlier, i.e., in section 4.1 that introduces our LaCIM model.

---

> ### Author Response · Authors · 2020-11-23
> **Response to Reviewer 2 (part 2)**
>
> - About the distributional change of $p(y|x)$ across domains. We firstly explain the role of $C$ (re-denoted as $d^e$ here) in affecting $p(y|x)$. For each domain $e$, we have that $p^e(x,y) := p(x,y|d^e) = \int p(x,y|s,z)p(s,z|d^e)dsdz$. Since the prior distribution $p(s,z|d^e)$ can change across domains, the joint distribution $p^e(x,y)$ (and $p^e(y|x)$) can also change across domains. For the effect of $d^e$ on causing the change of $p^e(s,z)$, since the $p^e(c):=p(c|d^e)$ can change (across domains), the prior $p^e(s,z):=p(s,z|d^e)=\int p(s,z|c)p(c|d^e)$ can also change. As for the concern on the invertibility of $f_x(s,z)$, we can use $p(y|x)$ to predict only when $\operatorname{Var(s|x)}=0$ (``for each $x$, one can uniquely go to $s$), in which case the effect of $C$ is void, as suggested theorem 7.1. But under more general case with $x = f_x(s, z) + \varepsilon_x$ in which $\operatorname{Var(s|x)} \neq 0$, the invertibility of $f_x$ is not enough to obtain a unique value of $s$ from $x$, with the existence of $\varepsilon_x$. Therefore, the invertibility of $f_x$ will not omit the effect of $d^e$. Specifically, the posterior of $S,Z$, i.e., the $p^e(s,z|x):=p(s,z|x,d^{e})$ can also change across domains. During inference on the test domain $e'$, since $p^{e'}(y|x)=\int p(s,z|x,d^{e'})p(y|s)dsdz \neq p^{e'}(y|x)$ since $p^e(s,z|x)\neq p^{e'}(s,z|x)$. To make it clearer, we expand our explainations in more details in the paragraph before section 4.2 and theorem 7.1. On the other hand, the invariant mechanism we assumes, **rather than** the posterior model $p^e(s,z|x)$, **is the generating process, $p(x|s,z)$ and $p(y|s)$** which follows the physical rules to generate the image and the semantic label. Therefore, during inference, we implement the invariant $p(x|s,z)$, rather than the $p(s,z|x)$ that change between the training and the test domain to inference $s,z$ as $\arg\max_{s,z} \log{p(x|s,z)}$. Then we feed the estimated $s$ into invariant predictor $p(y|s)$.
>
>
>
> - About Identifiability of invariant mechanisms. To make sure that the estimated $S$ for classification will not mix the information of non-causal factor $Z$, and that we can identify the correct predicting mechanism $p(y|s)$ for prediction, we establish the identifiability result which can ensure the disentanglement of $S$ from $Z$, followed by $p(y|s)$. To see why we can identify the ground-truth predicting mechanism, note that the theorem 4.3 (also the definition 4.2) states that our identified $\tilde f_x, \tilde f_y$ satisfy that $p_{\tilde f_y}(y|\tilde f_{x,S}^{-1}(x)) = p_{f_y}(y|f_{x,S}^{-1}(x))$ if the $f_x,f_y$ are ground-truth structural equations. That is, we estimate $s$ from the $p_{f_x}(x|s,z)$ induced by $f_x$, followed by $p_{f_y}(y|s)$ for final prediction. Since this work specifically considers the OOD problem, we call the $f_x,f_y$ as causal invariant mechanisms that can be used for invariant OOD prediction,  in order to distinguish these roles (for prediction) from other structural equations $f_c,f_s,f_z$. More related works about identifiability in the literature of causal inference are introduced, in the first paragraph in section 4.2 on page 5 and in supplement 7.6. Our results can extend beyond the exponential family and ANM assumption for $Y$ (**in theorem 4.4**), to incorporate the general distribution of $S, Z$ and categorical $Y$, which is matched with real applications. Besides, we have conducted a simulation in which the ANM assumption and that $S, Z$ belongs to the exponential family.
>
>
> - About the case when the test environment $e$ is trained. In this paper, we focus more on out-of-distribution prediction, in which the test environment does not belong to the training environments. In this OOD setting, we cannot directly implement $p^e(y|x)$ (for any $e \in \mathcal{E}_{train}$) for prediction since $p^{e'}(y|x)$ may not be equal to $p^e(y|x)$, as explained in the "About the distributional change of $p(y|x)$ across domains" item.
>
>
> - About the case when the new environment is not known. Yes, we first optimize $s,z$ and then use the invariant predictor $p(y|s)$ for prediction. The identifiability is commonly used in the literature of statistics (including causal inference), which means that the quantity of interest (e.g., the parameter of a model) can be determined by the model. This concept is the premise of the learning method. In our scenario, since we use $p(y|s),p(x|s,z)$ for prediction, we first need to make sure they are identifiable, or learnable. Specifically, the identifiability asks the question: if the ($f_x,f_y$) and ($\tilde f_x, \tilde f_y$) give rise to the same observational distribution, i.e., $p^e_{f_x,f_y}(x,y) = p^e_{\tilde f_x, \tilde f_y}(x,y)$, can we derive the same predicting mechanism, i.e., $p_{f_y}(f_{x,S}^{-1}(x))$? Our answer in theorem 4.3 confirms this result. Then under this premise, we design the learning method based on VAE for practical inference.

---

> ### Author Response · Authors · 2020-11-23
> **Response to Reviewer 2 (part 3)**
>
> - About the back-door path.  In section 4.1, we modified the original sentence to "the back-door path $Z \gets C \to S \to Y$ induces the correlation between $Z$ and $Y$ in every single domain". This is because the updated causal graph is allowed to describe the correlation between $Z$ and $S$ in a single domain. Since this correlation is domain-dependent, therefore it is a spurious correlation and cannot hold invariantly across domains.
>
> - About Notations. We would like to appreciate your great efforts in reading our paper.  We have listed the assumptions in a sequential list of items in theorem 4.3, theorem 7.6, 7.9 in supplement 7.3. For experimental sections, we add a paragraph that summarizes the implementation details in section 5.2, with more details about network structure and parameters reported in section 7.8-7.13 and in Tab.13,14, a description of Figure 2, an explanation of all variables in each experiment, and also the implementation details about interpretability. For $[f_x^{-1}]_{\mathcal{S}}(x)$, we have mentioned in the introduction of notations in section 3 that the "$[f]_\mathcal{A}$ denotes the $f$ restricted on dimensions of $\mathcal{A}$". We have elaborated the definition of the equivalent relation property of $\sim_p$ identifiability right after definition 4.2. We correct some mistakes: change from "brutal force" to "brute-force", adjust the font size of references into normal, replaced the $d^{e} \in \mathbb{R}^{m}$ with $d^{e}\in \\{0,1\\}^{m}$, capitalized the abbreviation of the structural causal model as "SCM", corrected the capitalized "CONDITION" in theorem 4.3 as ``diversity condition", from "pixels in image classification" to "modeling pixels as causal factors of $Y$ does not make much sense" in the second paragraph.
>
> - About Information intersection property. This is the high-level illustration of proof of theorem 4.3, which considers the case when $y$ follows the ANM model. In this case, rather than $f_y^{-1}(y')= f^{-1}_{x,S}(x')$ for $(x',y') \in \mathcal{X} \times \mathcal{Y}$, we assume that the $(x',y') \in f_x(S,Z) \times f_y(S)$, i.e., $f_y^{-1}(y')=f^{-1}_{x,S}(x')$ for $(x',y') \in f_x(\mathcal{S},\mathcal{Z})\times f_y(\mathcal{S})$. That is the $x',y'$ belong to the image space of $f_x$ and $f_y$, since in this case the $X$ and $Y$ are ANM model. Sorry for the misleading, we have clarified the ANM setting (in the last line on page 4) to make it clearer.
>
> - About Sufficient Statistics. The sufficient statistics $\boldsymbol{T}^{t}(t)$ ($t = s,z$) is matrix function, in which the $(i,j)$ element of $\boldsymbol{T}^{t}(t)$ is the $i$-th function for the $j$-th dimension of latent variable $t$, i.e., $t_j$. Therefore, they cannot degenerate to 0, since as functions, their values are dependent on latent variables. Rather, we put the assumption on the natural parameters, $\Gamma$ that characterize the distribution. For example in normal distribution when $x \in \mathcal{N}(\mu,\sigma^2)$, the sufficient statistics are $x$ and $x^2$; and the natural parameter $\mathbf{\Gamma} =(\mu,\sigma^2)$. Here the diversity condition on $\mathbf{\Gamma}_{c_1},...,\mathbf{\Gamma}_{c_R}$ implies that the distributions $p(s,z|c)$ with different values of $c$ are different, i.e., the $p(s,z|c)$ varies across $c$, which can provide the clue for the invariant mechanisms $p(x|s,z)$ and $p(y|s)$ to be identified.

---

### Official Review · AnonReviewer3 · 2020-10-31
**Paper 729 Review**

**Rating:** 6
**Confidence:** 3

**Review:**

### Summary

In this manuscript, the authors introduce a method for supervised learning in an out-of-distribution (o.o.d.) setting.

In particular, the authors consider the case where labelled data pairs $(x, y) ~ p^{e}(x, y)$ are available for training sampled from multiple environments / interventions $e \in \epsilon_{\mathrm{train}}$ and the goal is to learn a predictive model $\hat{y} = f(x)$ that will generalize to data sampled from environments $e \in \epsilon_{\mathrm{test}}$ not present in the training set.

In a nutshell, the authors propose to tackle this problem by fitting a deep generative model, in their case, a variational autoencoder, that models both features $x$ and targets $y$ as being jointly generated by a set of latent factors $(z, s)$. Crucially, their model removes the connection between the latent factors $z$ and the target $y$, encouraging disentanglement of causative (ideally, captured by $s$) and non-causative (ideally, captured by $z$) factors at the latent variable level. In their model, variation across environments is limited to the prior over the latent factors $(z, s)$, which is modelled conditional on the confounded $c$ (when known) or environment/domain indicator $d$ (when the true confounder is unknown) as $p(z, s | c) = p(z | c)p(s | c)$ (resp. for $d$).

From a theoretical standpoint, the model builds heavily on recent advances on nonlinear independent component analysis (ICA), such as Hyvärinen et al. 2019 and Khemakhem et al. 2020. To frame the problem in the language of that line of work, the authors introduce additional assumptions, some of which include:
(1) The (causal) mechanisms generating $x$ from $(z, s)$ and $y$ from $s$ must be invariant across training environments.
(2) Noise in those mechanisms must be additive.
(3) Additionally, assumptions analogous to those in e.g. Theorem 1 of Khemakhem et al. 2020.

Provided all those assumptions apply, the authors are able to adapt results in that line of work to their proposed model and prove similar identifiability results for their new setting.

Finally, the authors test their approach on a simulated toy dataset and three real-world datasets (a variation of the colored MNIST task introduced by Arjovsky et al. 2019, the NICO dataset from He et al. 2019 and a medical imaging classification task from the Alzheimer’s Disease Neuroimaging Initiative, ADNI). Results suggest that the proposed approach slightly outperforms basilines such as IRM (Arjovsky et al. 2019) or DANN (Ganin et al. 2016), among others.


### High-level assessment

Out-of-distribution generalization is, in my opinion, a topic of utmost relevance for the machine learning community.

From a methodological perspective, the authors here propose a novel application of existing ideas, models and identifiability results (e.g. Hyvärinen et al. 2019 and Khemakhem et al. 2020, among others) which they adapted to disentangle latent factors that jointly affect $x$ and $y$ from those which only affect $x$ while being able to borrow most of the theoretical machinery from that line of work to provide similar identifiability results. Because of this, I believe the contribution to be a sound, sufficiently innovative step worthy of publication, albeit arguably leaning on the incremental side.

All in all, I lean slightly towards recommending acceptance of the manuscript. Nonetheless, I also believe the paper has substantial room for improvement in key areas such as (i) soundness and depth of the experimental results and (ii) clarity of exposition, description and writing style. Those shortcomings, coupled to the incremental nature of the contribution, prevent me, for the moment, from giving a more enthusiastic endorsement of the manuscript for publication.

### Major points / suggestions for improvement

1. To the best of my knowledge, the manuscript is currently lacking key information to allow a reader to reproduce the results. In particular, I could not find any clear description of the architectures and training procedures used for all baselines and how these differ from (1) their original publications and/or (2) the architecture and training procedures of the proposed approach.

2. Related to the previous point, I believe it would be essential that the authors disentangle the effect of model architecture / model capacity from the actual methodological contribution. Because of the aforementioned issue, it is at this point not possible for me to assess whether the experiments were carried in such a way that all baselines have a similar capacity or whether a part of the performance improvements achieved by the proposed approach could be explained away by the model having more parameters and/or expliciting using for prediction additional variables (e.g. the domain indicators) than some of the baselines.

3. I believe the methods in Isle et al. 2020 and Teshima et al. 2020 are sufficiently relevant and should be included as additional baselines to better assess the practical impact of the differences between the proposed approach and those methods.

4. In my opinion, the manuscript is at the present moment not particularly clear. For example, in sections such as 4.3, I believe that the writing does not make it sufficiently clear which statements are supposed to be modelling choices / approximations / compromises for computational tractability and which statements follow mathematically without loss of generality. All in all, I would also recommend having the manuscript proof-read for English style and grammar issues.

---

> ### Author Response · Authors · 2020-11-23
> **Response to Reviewer 1 (part 1)**
>
> Thanks for your valuable suggestions regarding the experiments and clarity of methodology! We have modified our draft accordingly as follows:
>
> - About Reproducibility of Experiments. We report the implementation details, network structure, and the number of parameters of all methods in supplement 7.9, 7.10, 7.11,7.12,7.13, Tab.13, 14, with a summarizing paragraph in section 5.2.  To compare fairly, we adjust the model capacity (without changing the model structure) to keep the number of parameters of all methods at the same level, as shown in Tab.13. The superiority of our method over others is still kept, as shown in Tab.2.
>
> -  About Comparison to Baselines. We compare with the Domain Invariant variational autoencoders (DIVA) in [1] and Selecting Data Augmentation (SDA) in [2], as shown in Tab.2. The problem setting in [3] is few-shot domain adaptation, in which a few training samples are provided for the target domain, which is different from the OOD setting of ours that no samples in the test domain are provided. As shown in Tab.2, our method can outperform DIVA but not SDA, which is only implemented on CMNIST. Specifically, the SDA adopts to generate intervened data and then implement empirical risk minimization for optimization. However, this data augmentation step theoretically requires that the causal (non-causal) factors are known ahead, which cannot satisfied in ADNI and NICO in which the $S, Z$ are latent variables cannot be explicitly extracted. Therefore, we only implement it on CMNIST, in which the color and the label are spuriously correlated. As shown in Tab.2, the SDA can achieve 99.3\% in CMNIST, which is because it augmented data with a random color, which can alleviate the spurious correlation between the color and the label. Therefore, it can achieve comparable results than the one achieved on the original MNIST dataset. However, in many real scenarios, the $S, Z$ cannot be explicitly extracted. We have summarized the above analysis in the "Compared baselines" on page 8 and "Discussions" on page 9.
>
> - About the clarity. Thank you for your suggestions! We rewrite section 4.3. Specifically, we introduce our learning method which can be understood as follows:
>    * Our goal: "to learn the CIMe and $p_{f_x}(x|s,z),p_{f_y}(y|s)$ for invariant prediction".
>    * The reason why implementing the generative model to achieve this goal: identifiability. "we implement the generative model to learn $p^e(x,y)$ for $e \in \mathcal{E}_{\mathrm{train}}$, which has been guaranteed by identifiability result in theorem 4.3, 4.4 to identify the predicting mechanism.
>    *  Summarization of our method: we reformulate VAE, "as a generative model proposed in [4]"
>    * The detailed background introduction of VAE: "the VAE introduces the variational distribution $q_\psi$ to maximize the ELBO, as a tractable surrogate of MLE." (this is the compromise since ELBO is the surrogate of maximum likelihood)
>    * The optimization property of ELBO, "maximizing the ELBO will drive $q_\psi(z|x)$ to learn $p_\phi(z|x)$ and $p_{\phi}$ to learn the ground-truth model $p$ (including $p_\phi(x|z)$ to learn $p(x|z)$)." (this is the approximation)
>    * Similarly, in our supervised learning scenario, the property turns to "drive $p_\phi(x|s,z), p_\phi(y|s)$ to learn the CIMe (i.e., $p_{f_x}(x|s,z), p_{f_y}(y|s)$, and also "$q^e_\psi(s,z|x,y)$ to learn $p^e_\phi(s,z|x,y)$." (this is the approximation).
>    * In our supervised scenario, we introduce the ELBO with the variational distribution, the inference model of latent variables (*i.e.* $s,z$) given the observational variables (*i.e.*, $x,y$), $q(s,z|x,y)$ and corresponding ELBO as a surrogate (this is also the compromise).
>   * Reparameterization. " 1. the $q_\psi$ can inherit the modeling properties of $p_\phi$" (this is the modeling choice).  2. As $p^e_{\phi}(s,z|x,y) = \frac{p^e_{\phi}(s,z|x)p_{\phi}(y|s)}{p^e_{\phi}(y|x)}$ for our DAG, we can similarly reparameterize $q^e_{\psi}(s,z|x,y)$ as $\frac{q^e_{\psi}(s,z|x)q_{\psi}(y|s)}{q^e_{\psi}(y|x)}$." (this is the modeling choice). 3. "replace $q_{\psi}(y|s)$ with $p_\phi(y|s)$." (this is the modeling choice).
>   * Finally, "Substituting the above reparameterization into the ELBO", we can rewrite our reformulated ELBO.
>   * Optimize ELBO to learn $p(x|s,z), p(y|s)$.
> [1] Ilse, Maximilian, et al. "Diva: Domain invariant variational autoencoders." Medical Imaging with Deep Learning. PMLR, 2020.
>
> [2] Ilse, Maximilian, Jakub M. Tomczak, and Patrick Forré. "Designing Data Augmentation for Simulating Interventions." arXiv preprint arXiv:2005.01856 (2020).
>
> [3] Teshima, Takeshi, Issei Sato, and Masashi Sugiyama. "Few-shot Domain Adaptation by Causal Mechanism Transfer." arXiv preprint arXiv:2002.03497 (2020).
>
> [4] Kingma, D. P., and Welling, M. (2014), Auto-encoding variational Bayes, in "Proceedings of the International Conference on Learning Representations (ICLR 2014)", ICLR Committee, Banff, Canada.

---

> ### Author Response · Authors · 2020-11-23
> **Response to Reviewer 1 (part 2)**
>
> -  Allow us to restate the novelty of our theoretical analysis about interpretability, which is far beyond the one in [5] in terms of
>
>     * We extend the result of [5] in unsupervised learning to the supervised learning scenario, in which the main objective is to disentangle the causal factor $S$ from the non-causal factor $Z$. The novel technique we implement to achieve this goal is to leverage that the $S$ is the confounder of both $X$ and $Y$, as illustrated in the first paragraph in section 4.2 and a more detailed analysis in section 7.3.
>
>     * Moreover, the [5] only considers the case when the latent variable belongs to the exponentially family. In contrast, we have extended this result to any distribution of $S,Z$ as long as the $p(s,z)$ belongs to **Sobolev space**; and also to the case when $Y$ is categorical distribution, which can thus incorporate the classification task. **These results are summarized in theorem 4.4 and the technique is novel**, as explained in section 7.4 with more details.
>
>    *  Last but not least, in this modified draft, we have extended the result in **a more general setting** that the confounder $C$ can take a specific value for each sample unit, which can explain the phenomena of spurious correlation in the single dataset. In contrast, the [5] only consider the case when the $C$ (it is denoted as $u$ in [5]) can only have a fixed value shared by all samples in each domain.
>
> [5] Khemakhem, Ilyes, et al. "Variational autoencoders and nonlinear ica: A unifying framework." International Conference on Artificial Intelligence and Statistics. 2020.

---

### Author Response · Authors · 2020-11-23
**Paper Revision**

We appreciate the great efforts and valuable suggestions. We have revised our draft mainly in the following aspects.

- Regarding the causal graph, we have extended it to the case when the confounder $C$ is allowed to take the value for each sample unit, according to the suggestion from Reviewer 2. This extension can explain the correlation between non-causal factors and causal factors in the single dataset, which is more matched with real scenarios. We have correspondingly generalized the identifiability result in this case.

- We clarify the notations and words that can incur misunderstandings, to make it clearer.

- We conduct more experiments, including the simulation compared with the case with a different number of environments, i.e, $m=3$ and $m=7$ (in Tab.1); comparisons with more related baselines such as DIVA and SDA (in Tab.2), under comparable model capacity and the number of parameters; and the experiments on intervened data, which is also another OOD scenario (in Tab.9).

-  We expand the experimental parts in terms of the implementation details (in Tab.13,14, supplement 7.9-7.13, section 5.2) and the meaning of every variable in each experiment to make our implementation clearer and more reproducible.

---

### Decision · Program_Chairs · 2021-01-07
**Final Decision**

**Decision:**

Reject

**Comment:**

This work introduces a method for supervised learning that takes a data-generating process into account. While the paper proposes an interesting approach to learning a causally invariant model, the reviewers had several concerns about the proposed method. I thank the authors for having the paper revised, addressing the reviewers' comments. However, there are still legitimate issues unresolved about the specific theoretical results and assumptions made throughout the work.  I share similar concerns, and, therefore, recommend rejection. Still, I would like to encourage the authors to address the reviewers' problems in the paper's next iteration.